# WHEN FEWER LAYERS BREAK MORE CHAINS: LAYER PRUNING HARMS TEST-TIME SCALING IN LLMS

## ABSTRACT

Layer pruning has emerged as a widely adopted technique for improving the efficiency of large language models (LLMs). Although existing methods demonstrate strong performance retention on general knowledge tasks, their effect on long-chain reasoning, a more brittle yet crucial capability, remains largely unexplored. In this work, we study the impact of layer pruning on long-chain reasoning through the lens of test-time scaling, a key mechanism in modern LLMs that enables strong reasoning capacity by allocating more computation at inference time. With extensive experiments, we demonstrate that pruning even one or two layers can severely impair test-time scaling, with performance collapsing drastically on long reasoning benchmarks even when performance on knowledge-intensive and shallow reasoning tasks remains stable. Furthermore, we find that standard supervised fine-tuning remedies fail to recover test-time scaling once it has deteriorated. Through in-depth analyses, we identify the mechanisms underlying this fragility of test-time scaling and highlight the fundamental risks of applying layer pruning to reasoning-intensive LLMs. These findings call for a rethinking of layer pruning strategies and provide insights for developing methods that preserve the robustness of reasoning.

## 1 INTRODUCTION

Recent studies have revealed that large language models exhibit considerable redundancy across their depth, with a substantial fraction of layers contributing minimally to overall functionality (Sun et al., 2025; Men et al., 2025). These findings have motivated growing interest in layer pruning, a family of techniques that remove layers to improve efficiency while preserving competitive accuracy (Muralidharan et al., 2024; Su et al., 2025). Recent advances highlight its promise: for instance, ShortGPT retains up to 85% of the original model accuracy after pruning 25% of layers (Men et al., 2025), and recovers over 90% with fine-tuning (Lu et al., 2024).

Despite these encouraging results, prior evaluations of layer pruning have relied primarily on general knowledge benchmarks such as HellaSwag (Zellers et al., 2019), which emphasize factual knowledge and shallow reasoning. However, such evaluations overlook long-chain reasoning, a capability central to many LLM applications, such as mathematical problem solving (Lightman et al., 2023; Mathematical Association of America, 2024), scientific reasoning (Rein et al., 2024) and multi-step logical inference (Suzgun et al., 2023). Importantly, long-chain reasoning is often more brittle under layer pruning within LLMs. As shown in Figure 1, on benchmarks with short reasoning chains such as MMLU (Hendrycks et al., 2021a), performance of Qwen3-8B degrades only gradually as pruning depth increases, whereas on tasks requiring long reasoning chains with thousands of tokens such as AIME24 (Mathematical Association of America, 2024), performance collapses after pruning a single layer and drops to near zero once 10% of layers are removed. These observations underscore the fragility of long-chain reasoning under layer pruning, which remains largely unexplored.

To study this fragility more systematically and mechanistically, we turn to test-time scaling, a recently emerging technique that provides a finer-grained understanding of how LLMs acquire and exercise long reasoning abilities (Muennighoff et al., 2025; Yeo et al., 2025). By allocating more computation at inference time, such as generating longer chains of thought or exploring multiple reasoning paths (Wei et al., 2022; Chen et al., 2024a), test-time scaling has emerged as a key mechanism for enabling human-level reasoning (Guo et al., 2025). Crucially, test-time scaling allows us to

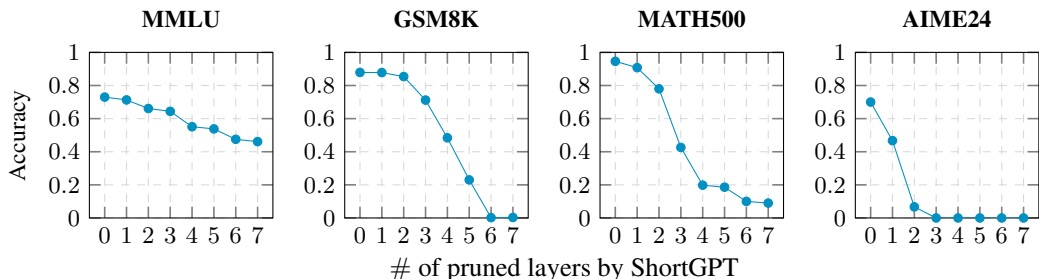

*Figure 1:* Accuracy of the pruned Qwen3-8B by ShortGPT across various pruning depths on MMLU, GSM8K, MATH500 and AIME24. *LLM-as-a-Judge* is applied to MATH500 and AIME24 as described in Section 3.

measure how reasoning ability depends on inference-time computation, thereby exposing degradations in terms of reasoning after layer pruning that standard knowledge or short-context benchmarks fail to capture. Thus, it provides a natural probe for assessing the vulnerability of complex and long reasoning under layer pruning.

In this work, we systematically investigate how layer pruning degrades test-time scaling, and whether supervised fine-tuning methods can recover it. Our findings reveal that pruning even one or two layers can severely impair test-time scaling, and that this degradation cannot be effectively recovered by LoRA fine-tuning (Hu et al., 2022) and full-parameter fine-tuning. We further provide in-depth analyses to uncover the underlying reasons, indicating that layer pruning causes recurring loops, reduced trajectory diversity and diminished self-reflection during long chain-of-thought and test-time scaling, pointing to structural damage rather than surface-level accuracy loss. This work brings to light critical yet previously overlooked vulnerabilities of layer pruning, underscoring the need for greater attention from the community to its risks on long reasoning and test-time scaling. In summary, our main contributions are as follows:

• We rigorously evaluate the impact of layer pruning on both sequential and parallel test-time scaling, demonstrating the fragility of test-time scaling under layer pruning.

• We comprehensively examine full-parameter fine-tuning and LoRA fine-tuning and demonstrate their limited ability to restore test-time scaling once degraded by layer pruning.

• We trace the root causes of this fragility through qualitative and quantitative analyses of model outputs, and further explore layer contributions and the impact of calibration data in reasoning-intensive settings, yielding insights for designing future methods that safeguard long-chain reasoning capacity within LLMs.

## 2  RELATED WORK

**LLM Layer Pruning.** Layer pruning aims to reduce the depth of LLMs to improve efficiency while retaining performance. For instance, Kim et al. (2024) compare pruning strategies based on criteria such as magnitude, perplexity, and Taylor expansion. ShortGPT (Men et al., 2025) highlights the redundancy of LLM layers and introduces Block Influence (BI) to estimate each layer's contribution and prune layers with low BI scores. Similarly, SLEB (Song et al., 2024) indicates LLMs' redundancy with high similarity between the outputs of neighboring blocks and applies this similarity to eliminate redundant blocks. Lu et al. (2024) further provide a comprehensive comparison of these layer pruning methods and shows that parameter-efficient fine-tuning can effectively restore pruned models. Additional analyses of these pruning strategies are presented in (Siddiqui et al., 2024; Muralidharan et al., 2024). There also exist structured pruning methods, such as LLM-Pruner (Ma et al., 2023), SliceGPT (Ashkboos et al., 2024), and FinerCut (Zhang et al., 2024), which operate at finer granularities such as channels, attention layers, and feed-forward networks (FFNs). In contrast, our work focuses on pruning at the Transformer layer level. Recently, another line of work proposes merging-based pruning, which fuses adjacent layers to decrease depth without retraining from scratch. Representative methods include MKA (Liu et al., 2024), which aligns layer manifolds

to guide merging, and LaCo (Yang et al., 2024b), which collapses subsequent layers into a prior layer while preserving their functional capacity.

**Test-time Scaling.** Test-time scaling, also called inference scaling, refers to techniques that enhance LLM performance by allocating additional computation at inference time. Recent studies have shown that LLMs benefit significantly from test-time scaling in long and complex reasoning (Wei et al., 2022; Chen et al., 2024a; Guo et al., 2025; Chen et al., 2024b; Team et al., 2025; Yeo et al., 2025). Current advances on test-time scaling mainly fall into two categories: sequential scaling and parallel scaling (Zuo et al., 2025). Sequential scaling focuses on extending LLM outputs into longer responses by increasing the length of reasoning chains, often through reflective or chain-of-thought processes. For example, s1 (Muennighoff et al., 2025) proposes a simple recipe combining a small curated reasoning set and budget forcing, showing consistent gains from longer inference. Similarly, T1 (Hou et al., 2025) advances reasoning via reinforcement learning and inference scaling, demonstrating reliable improvements with increased test-time budgets. Concurrently, parallel scaling involves producing multiple candidate responses during inference, either by increasing the number of sampled outputs (Chen et al., 2023) or expanding steps of search (Xie et al., 2024). The resulting candidates are then integrated through an aggregation strategy, often guided by reward or scoring models (Zhang et al., 2025).

# 3 Unveiling the Fragility of Test-Time Scaling under Layer Pruning

Recent advances in layer pruning have demonstrated that substantial reductions in model depth, e.g., removing up to 25% of layers, can be achieved with only marginal degradation on surface-level metrics such as perplexity or zero-shot accuracy (Men et al., 2025; Lu et al., 2024). However, these metrics fail to capture test-time scaling, a critical mechanism for eliciting reasoning capabilities (Wei et al., 2022; Muennighoff et al., 2025) which is far more brittle and may be sensitive to compression. In this section, we investigate how layer pruning disrupts this scaling dynamics.

**Problem formulation.** In this work, we focus on pruning Transformer layers. Following the notions in (Lu et al., 2024), we consider an LLM $\mathcal{M}$ composed of a sequence of Transformer layers $\mathcal{M} = l_1 \circ l_2 \circ \cdots \circ l_n$, where each layer $l_i$ includes a multi-head self-attention module and FFNs (Vaswani et al., 2017). The objective of layer pruning is to identify a set of layers $\{l'_1, l'_2, \ldots, l'_m\}$ where $m < n$, such that the pruned model $\mathcal{M}' = l'_1 \circ l'_2 \circ \cdots \circ l'_m$ retains sufficient performance while lowering computational burden.

**Models.** We study two reasoning models: s1.1-7B (Muennighoff et al., 2025) and Qwen3-8B (Yang et al., 2025). s1.1-7B is a 7-billion parameter language model finetuned from Qwen2.5-7B-Instruct (Yang et al., 2024a) on s1K-1.1 dataset (Muennighoff et al., 2025), which augments reasoning traces with budget forcing to elicit test-time scaling. Qwen3-8B is an 8-billion parameter instruction-tuned model which supports both thinking and non-thinking modes, where the former allocates additional test-time compute for multi-step reasoning while the latter produces concise responses. Both models have been shown to exhibit robust test-time scaling behavior, therefore providing an appropriate testbed for analyzing how layer pruning impacts such reasoning scalability.

**Layer pruning methods.** We focus on training-free pruning techniques that reduce model depth while preserving efficiency. For direct removal, we consider (i) **ShortGPT** (Men et al., 2025), which eliminates layers with low Block Influence scores, and (ii) **Reverse-order** (Kim et al., 2024), which removes deeper layers. For merging-based pruning, we adopt (iii) **LaCo** (Yang et al., 2024b), which linearly combines adjacent layers to preserve representational capacity. Appendix B provides their detailed explanations and implementation specifics in this work. We select these three methods for their strong empirical performance and complementary strategies, direct removal versus layer merging, providing diverse perspectives on how layer pruning influences test-time scaling.

**Evaluation dimensions.** We evaluate the effect of layer pruning on test-time scaling along two orthogonal dimensions: (1) **Sequential scaling.** We measure performance under reasoning with increasing thinking token budgets of $[512, 1024, 2048, 4096, 8192]$. We set the temperature to 1.0 and run experiments with three randomly selected seeds, reporting the average results. This setting reflects iterative reasoning, where later computations build upon earlier steps, allowing reflective and chain-of-thought steps into longer reasoning chains. (2) **Parallel scaling.** We report $pass@k$

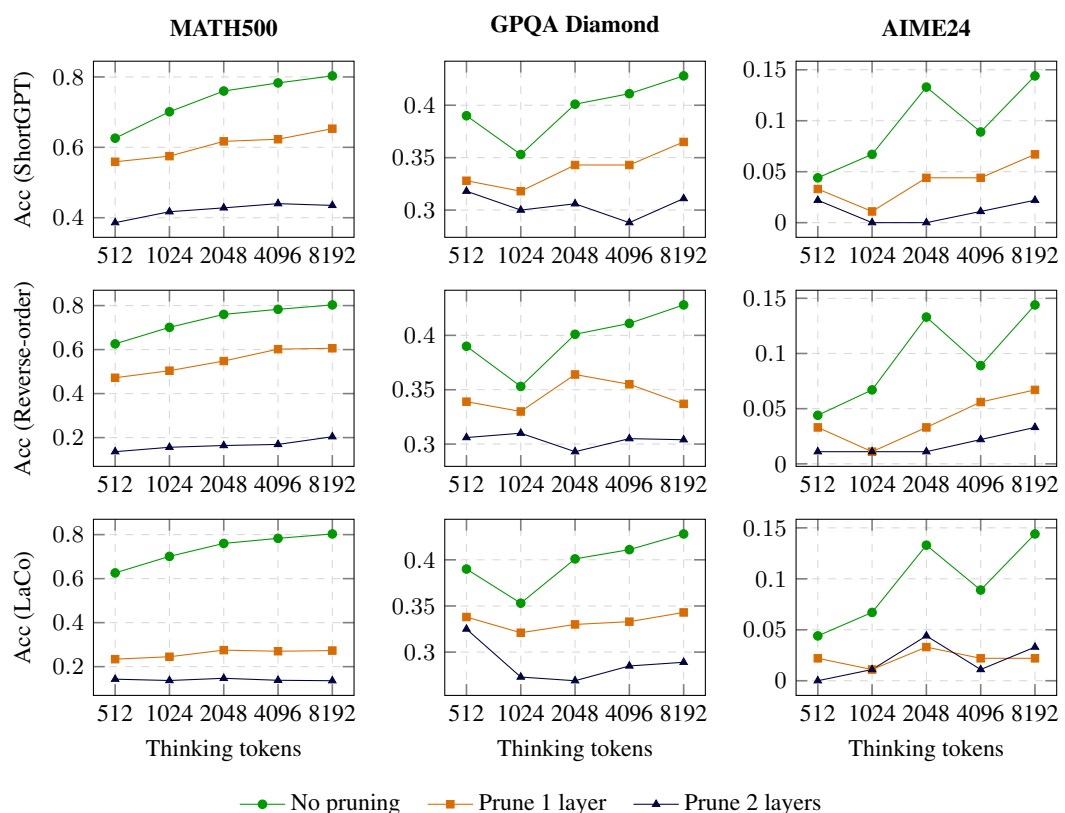

*Figure 2:* Sequential test-time scaling of s1.1-7B under different pruning depths.

with temperature $1.0$, which means the probability that at least one correct solution exists among $k$ randomly sampled outputs, with $k = [1, 2, 4, 8, 16]$. This setting captures parallel inference, where multiple reasoning trajectories are sampled independently and the results are aggregated.

**Evaluation datasets and metrics.** We evaluate on three standard reasoning benchmarks: (1) **MATH500** (Hendrycks et al., 2021b; Lightman et al., 2023), 500 competition-level math problems spanning diverse topics and difficulty levels; (2) **GPQA Diamond** (Rein et al., 2024), 198 graduate-level biology, physics, and chemistry questions targeting expert scientific reasoning; (3) **AIME24** (Mathematical Association of America, 2024), the 30 problems from the 2024 American Invitational Mathematics Examination, covering algebra, geometry, number theory, and probability. Our evaluation framework builds on *lm-evaluation-harness* (Gao et al., 2024). Following (Muennighoff et al., 2025), we adopt *LLM-as-a-Judge*, prompting a strong reference model, *GLM-4.5-Flash* (Zeng et al., 2025), to assess correctness, providing a more robust evaluation than exact string matching.

**Results.** For sequential scaling, results for s1.1-7B are reported in Figure 2, and those for Qwen3-8B are provided in Figure 7 in Appendix C. Overall, layer pruning severely disrupts sequential test-time scaling. For s1.1-7B, across ShortGPT, Reverse-order, and LaCo, pruning even a single layer substantially impairs scaling. With two layers pruned, the scaling ability on MATH500 and GPQA Diamond nearly vanishes, where performance remains flat or even degrades as thinking tokens increase. On the more challenging AIME24 benchmark, the accuracy of s1.1-7B drops close to zero. Qwen3-8B is relatively more resilient to pruning than s1.1-7B: after pruning one layer, it largely preserves sequential scaling, with performance continuing to improve as thinking tokens increase, and on GPQA Diamond it even approaches or slightly surpasses the original model. Nevertheless, pruning two layers completely collapses sequential scaling, leading to a sharp drop in performance. Taken together, these findings indicate that sequential test-time scaling is highly fragile under layer pruning, breaking down once pruning depth exceeds a very shallow threshold. For parallel scaling as shown in Figure 3, the results diverge. While ShortGPT and Reverse-order severely impair the model's parallel scaling ability, LaCo exhibits markedly different behavior. Even with two layers

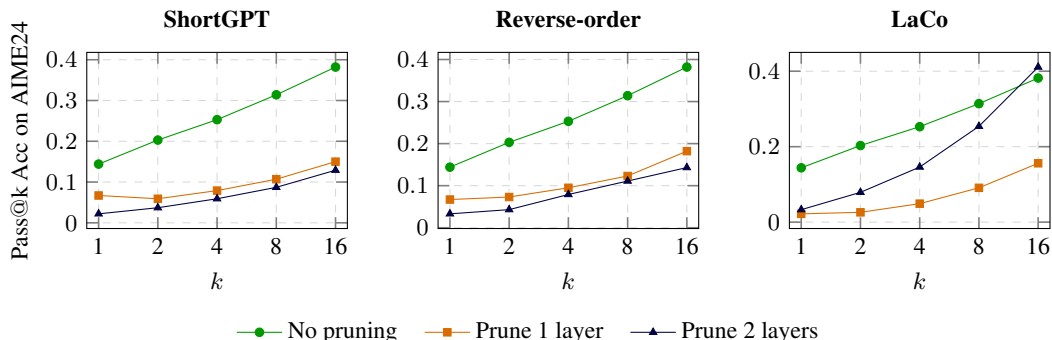

*Figure 3:* Parallel test-time scaling of s1.1-7B on AIME24 under different pruning depths.

removed, LaCo not only sustains robust parallel scaling but in some cases surpasses both the 1-layer pruned variant and the unpruned baseline. This contrast suggests that merging-based pruning strategies may provide a promising direction for preserving or even enhancing parallel test-time scaling, in contrast to direct removal approaches.

## 4 RECOVERING TEST-TIME SCALING: IS FINE-TUNING SUFFICIENT?

The fragility of test-time scaling under layer pruning, revealed in Section 3, raises a natural question: *Can supervised fine-tuning (SFT) restore test-time scaling after layer pruning?* While prior work has shown that SFT can substantially improve the performance of pruned LLMs, such evaluations have largely been restricted to relatively simple language and knowledge tasks such as MMLU and HellaSwag (Lu et al., 2024). Whether SFT can restore the disrupted test-time scaling remains a non-trivial open problem. In this section, we study two representative strategies, LoRA fine-tuning (LoRA FT) (Hu et al., 2022) and full-parameter fine-tuning (Full FT), to assess their effectiveness in recovering test-time scaling degraded by layer pruning.

**Experimental Settings.** Since ShortGPT, Reverse-order, and LaCo all induce more severe degradation of sequential scaling, we employ the same experimental configuration used for sequential scaling in Section 3. For SFT settings: (1) **LoRA Fine-tuning.** We fine-tune the pruned variants of s1.1-7B and Qwen3-8B on the s1K-1.1 dataset (Muennighoff et al., 2025) using LoRA FT with rank 16 and scaling factor 32. For each pruned model, we perform a grid search over learning rates $[1e-6, 4e-6, 1e-5, 4e-5, 1e-4]$, selecting the best value by validation on MATH500, GPQA Diamond and AIME24, respectively. Results are reported on the corresponding datasets, with detailed learning rate sweeps provided in Appendix D. (2) **Full-parameter Fine-tuning.** We adopt the same codebase, SFT dataset s1K-1.1 and configurations from s1 (Muennighoff et al., 2025), applying Full FT with a fixed learning rate of $1e-5$.

**Results.** Figure 4 presents the sequential scaling performance of pruned s1.1-7B after Full FT. The corresponding results for pruned Qwen3-8B after Full FT, pruned s1.1-7B after LoRA FT, and pruned Qwen3-8B after LoRA FT are provided in Figures 8, 9 and 10 in Appendix E, respectively. Overall, Full FT outperforms LoRA FT, but neither Full FT nor LoRA FT restores the sequential test-time scaling disrupted by layer pruning. For 1-layer pruned models, both SFT strategies yield negligible gains, and in some cases such as pruned Qwen3-8B by LaCo or Reverse-order, on GPQA Diamond, SFT even reduces performance. We attribute this to potential over-training on the small-scale dataset s1.1-1K with only 1k samples, which is insufficient to adapt a nearly intact Qwen3-8B model. For 2-layer pruned models, both Full FT and LoRA FT substantially improve accuracy (e.g., by up to 0.2 on MATH500). Nevertheless, the performance remains still significantly below the original model and sequential scaling is not recovered. These results indicate that small-scale SFT fails to repair the fundamental breakdown of sequential test-time scaling, highlighting its intrinsic limitations in recovering reasoning scalability after layer pruning.

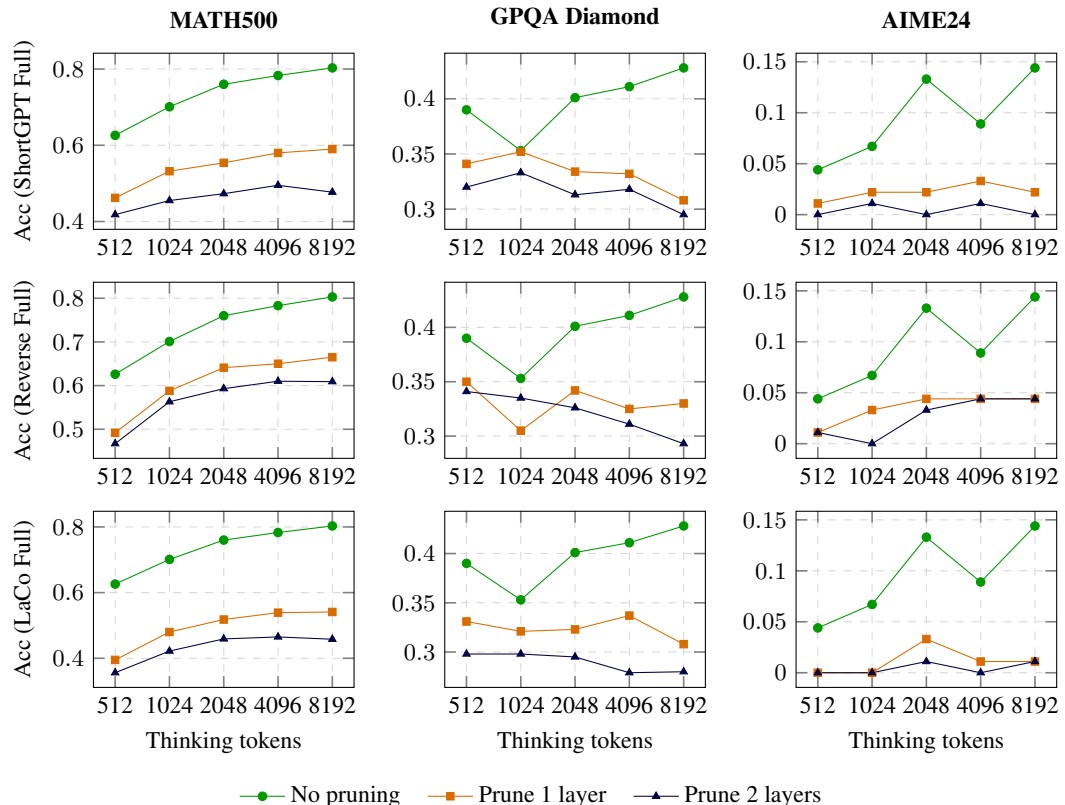

*Figure 4:* Sequential test-time scaling of s1.1-7B after Full FT under different pruning depths. We denote "Full" as Full FT and "Reverse" as Reverse-order in the figure for simplicity.

## 5 ANALYSIS

### 5.1 CASE STUDY: REPETITIVE REASONING LOOPS AFTER LAYER PRUNING

To better understand why pruning even a single layer severely disrupts test-time scaling, we conduct a detailed case study of model outputs. As illustrated in examples in Figure 5, layer pruning impairs the model's ability to sustain coherent reasoning trajectories. Instead, the pruned models often fall into **repetitive self-doubt** and **looping deduction**. For example, in the MATH500 case, the model initially identifies the correct candidate solution but repeatedly questions itself, rechecking the same invalid negative cases, e.g., *"But wait, let me check again..."*, without making progress. In many cases, the pruned model initially arrives at a promising intermediate step, but rather than building on it, the reasoning derails into repetitive checks of already invalid branches, or into circular speculation that never converges. Once stuck, the model fails to explore alternative directions and keeps generating redundant text without real progress. This suggests that layer pruning disrupts the model's ability to maintain diverse reasoning chains, forcing it into narrow and ultimately unproductive loops. These observations resonate with our central claim: *Layer Pruning Harms Test-Time Scaling*. By pruning layers, we inadvertently collapse model's reasoning diversity, causing breakdowns in long chain-of-thought and test-time scaling where longer thinking no longer leads to stronger performance. We provide the complete failure examples for more illustration in Appendix G.

To quantify this observation, we adopt Self-BLEU (Zhu et al., 2018), which measures the internal diversity of a corpus by computing the average similarity of each sentence to all others. Higher Self-BLEU score indicates greater textual similarity and lower diversity, suggesting potential increased repetition in reasoning. Table 1 reports average Self-BLEU scores computed from model outputs across AIME24 dataset. ShortGPT and LaCo consistently exhibit markedly higher Self-BLEU scores, indicating reduced output diversity. Moreover, Full FT fails to recover diversity effectively, with scores remaining elevated. Full FT even increases the Self-BLEU scores of Reverse-

| [MATH500] | [GPQA Diamond] | [AIME24] |
|---|---|---|
| Find the number of integer values of $k \in [-500, 500]$ for which $\log(kx) = 2\log(x+2)$ has exactly one real solution. | Two quantum states with lifetimes $10^{-9}$ s and $10^{-8}$ s need to be clearly resolved. What is a possible energy difference? | Let $p$ be the least prime for which $n^4 + 1 \equiv 0 \pmod{p^2}$. Find the least positive $m$ such that $m^4 + 1 \equiv 0 \pmod{p^2}$. |
| *...Therefore, the only possible $k$ is $k = 8$... But wait, let me check again... But wait... Let me think. If $k$ is negative... But wait, let me check if there are any other $k$ values... Wait, in Case 2, $k$ is negative. Let me check if for any negative $k$... Wait, in Case 2, $x$ is negative... But wait, let me check again... But wait, let me check if there are any other $k$ values... Thus, $k = 0$ or $k = 8$. But $k$ must be negative, so $k = 0$ is not valid. Therefore, there is no solution where $k$ is negative...* | The lifetime of a quantum state is related to its uncertainty in energy. The uncertainty principle tells us that $\Delta E \Delta t \geq \hbar/2$... *Wait, but how does the energy difference relate to the lifetimes?... Wait, maybe the question is referring to resolving the two states in time?... But the options are given as single values. Wait, maybe the question is not about the sum... Wait, no... Wait, maybe the question is about resolving the two states in energy... Wait, the uncertainty principle is...* | The smallest prime $p$ congruent to 1 modulo 8 is 17... 17 divides 83521-17=83504. Therefore, 17 divides 83521-17, so 17 divides 83521-17. Therefore, 17 divides 83521-17. Therefore, 17 divides 83521-17. Therefore, 17 divides 83521-17. Therefore, 17 divides 83521-17. Therefore, 17 divides 83521-17. Therefore, 17 divides 83521-17. Therefore, 17 divides 83521-17. Therefore, 17 divides 83521-17. Therefore, 17 divides 83521-17. ... |
| Setup is correct, but domain analysis for $k < 0$ causes repetitive self-doubt. | Correct principle cited, but application is inconsistent, leading to repetitive chains. | Finds $p = 17$ but fails to lift the root modulo $p^2$, leading to reasoning loops without progress. |

*Figure 5:* Examples of repetitive thinking from 1-layer pruned s1.1-7B model. Unlike the original model, the pruned version gets stuck in logic loops. The teal text denotes the correct reasoning trace while light blue text is the repetitive reasoning trace leading to failure, and the blue text is a brief summary of the failure reason.

*Table 1:* Self-BLEU scores for pruned models (lower is better). "1 layer" and "2 layer" indicate # of pruned layers, whereas "sft" refer to Full FT after pruning.

| Model/Method | Original | ShortGPT | | | | Reverse-order | | | | LaCo | | | |
|---|---|---|---|---|---|---|---|---|---|---|---|---|---|
| | original | 1 layer | 2 layer | 1 layer sft | 2 layer sft | 1 layer | 2 layer | 1 layer sft | 2 layer sft | 1 layer | 2 layer | 1 layer sft | 2 layer sft |
| s1.1-7B | 0.685 | 0.894 | 0.871 | 0.775 | 0.779 | 0.521 | 0.442 | 0.765 | 0.728 | 0.812 | 0.936 | 0.776 | 0.786 |
| Qwen3-8B | 0.688 | 0.805 | 0.961 | 0.745 | 0.739 | 0.601 | 0.606 | 0.749 | 0.725 | 0.586 | 0.674 | 0.765 | 0.754 |

order pruned models, which may help explain the performance degradation after SFT noted in Section 4. Overall, these results suggest that pruning induces a fundamental degradation of the model's capacity to generate diverse reasoning trajectories, which cannot be easily mitigated by supervised fine-tuning.

## 5.2 ASSESSING REASONING QUALITY BY SELF-REFLECTION HEURISTICS

To further assess reasoning quality, we use a rubric to quantify the frequency of three self-reflection heuristics (Gandhi et al., 2025) in s1.1-7B on AIME24, with *gpt-4o-mini* serving as an automated judge. The heuristics are: (i) **verification**, checking the validity of intermediate or final solutions, (ii) **backtracking**, revising earlier steps or pursuing alternatives after detecting an error, and (iii) **subgoal setting**, decomposing a complex task into manageable subproblems. These behaviors serve as established proxies for self-reflection, demonstrating a model's ability to monitor, evaluate, and regulate its own reasoning process.

After filtering out non-productive loops and repetitions, we observe a modest yet consistent decline in these heuristics for the pruned model as shown in Table 2. This indicates that pruning undermines more than surface-level diversity: it erodes the model's capacity for structured, self-correcting reasoning. Moreover, this impairment proves resistant to recovery: in most cases, Full FT fails to restore the heuristic frequency to the level of the unpruned baseline, suggesting a more fundamental degradation of capacity for complex and long reasoning.

*Table 2:* Evaluation results of pruned models across different strategies (higher is better). "1 layer" and "2 layer" indicate # of pruned layers, whereas "sft" refer to Full FT after pruning. The behaviors measured are: **Verification** (checks intermediate results), **Backtracking** (tries alternative approaches), and **Subgoal** (breaks main problems).

| | Original | ShortGPT | | | | Reverse-order | | | | LaCo | | | |
|---|---|---|---|---|---|---|---|---|---|---|---|---|---|
| Behavior/Models | Original | 1 layer | 2 layer | 1 layer sft | 2 layer sft | 1 layer | 2 layer | 1 layer sft | 2 layer sft | 1 layer | 2 layer | 1 layer sft | 2 layer sft |
| **Verification** | 2.167 | 0.867 | 0.467 | 2.333 | 2.033 | 0.433 | 0.000 | 1.967 | 2.000 | 0.867 | 0.900 | 2.033 | 2.700 |
| **Backtracking** | 2.067 | 0.733 | 0.533 | 2.300 | 2.333 | 1.733 | 0.567 | 1.967 | 2.767 | 1.667 | 1.000 | 1.833 | 2.467 |
| **Subgoal** | 4.967 | 2.367 | 1.233 | 4.100 | 3.400 | 4.200 | 2.067 | 3.833 | 2.967 | 3.100 | 0.167 | 4.067 | 3.167 |

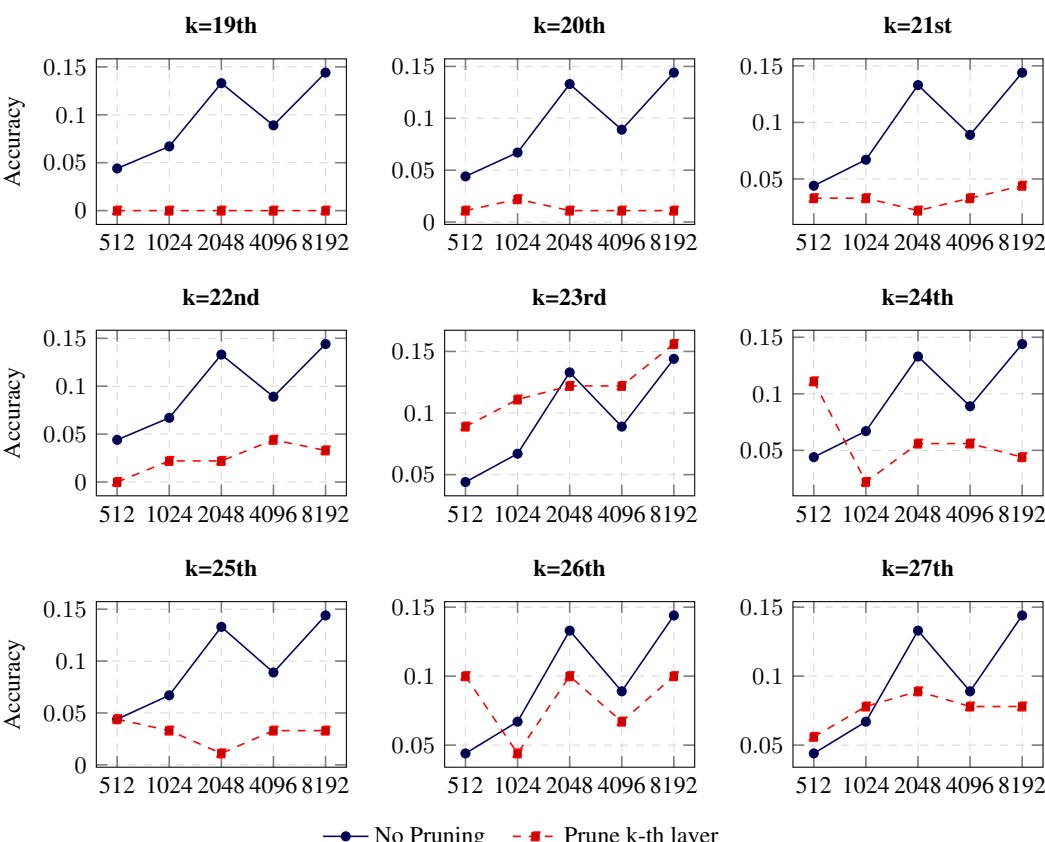

*Figure 6:* Brute-Force of s1.1-7B Layer Ablation for Sequential Test-time scaling (last 9 layers)

## 5.3 BRUTE-FORCE EXPLORATION OF LAYER ABLATION

To systematically assess the contribution of individual layers in both s1.1-7B and Qwen3-8B, we conduct a brute-force ablation study: pruning one layer at a time and evaluating the resulting models on sequential test-time scaling with AIME24. This setup allows us to directly measure the marginal role of each layer in sustaining test-time scaling. The performance of each ablated model is compared against that of the original model, providing a fine-grained view of how capacity of sequential test-time scaling is distributed across the layers. Comprehensive results are reported in Appendix F (Figures 11, 12, 13, and 14), while Figure 6 highlights the case of s1.1-7B, focusing on ablations of its last nine layers for illustrative purposes.

Our findings reveal that most layers play a non-trivial role in enabling test-time scaling. Even the removal of a single layer often leads to substantial degradation, indicating that reasoning relies on widely distributed contributions rather than being concentrated in a small subset of layers. Consequently, even modest layer pruning can disproportionately impair the capacity for test-time scaling, underscoring a fundamental tradeoff between pruning and reasoning fidelity.

*Table 3:* Effect of calibration data on ShortGPT's pruning order. Qwen3-8B and s1.1-7B show high sensitivity to calibration data, while the general-purpose LLaMA3.1-8B is stable.

| Model | Calibration Dataset | Pruning Order (Low Layer Importance First) |
|---|---|---|
| **Qwen3-8B** | PG19 | [20, 17, 21, 18, 2, 16, 19, 15, 23, 14] |
| | MATH500 | [2, 3, 1, 5, 17, 4, 20, 16, 19, 21] |
| | AIME24 | [2, 3, 1, 20, 17, 5, 16, 21, 24, 19] |
| **s1.1-7B** | PG19 | [16, 17, 15, 14, 18, 13, 12, 11, 20, 23] |
| | MATH500 | [2, 16, 15, 5, 17, 14, 13, 12, 4, 1] |
| | AIME24 | [16, 2, 15, 17, 14, 5, 13, 4, 12, 1] |
| **LlaMA3.1-8B** | PG19 | [25, 27, 26, 24, 28, 23, 22, 29, 20, 21] |
| | MATH500 | [25, 24, 26, 23, 27, 28, 22, 21, 29, 20] |
| | AIME24 | [25, 23, 24, 26, 27, 22, 28, 21, 20, 19] |

*Table 4:* Effect of calibration data on LaCo's merge order. The table shows the sequence of layers to be absorbed. The results show that LaCo's layer selection is notably more stable against changes in calibration data compared to ShortGPT.

| Model | Calibration Dataset | Merge 1 Layer | Merge 2 Layers | Merge 3 Layers |
|---|---|---|---|---|
| **Qwen3-8B** | Wikitext-2 | [34] | [34, 29] | [34, 33, 29] |
| | MATH500 | [34] | [34, 33] | [34, 33, 32] |
| **s1.1-7B** | Wikitext-2 | [22] | [22, 18] | [25, 18, 17] |
| | MATH500 | [21] | [26, 18] | [26, 25, 16] |
| **LlaMA3.1-8B** | Wikitext-2 | [30] | [30, 15] | [30, 29, 15] |
| | MATH500 | [30] | [30, 27] | [30, 29, 26] |

## 5.4 IMPACT OF CALIBRATION DATA ON PRUNING METHODOLOGIES

In this section, we discuss the sensitivity of ShortGPT and LaCo to calibration data. For ShortGPT, calibration data has a significant effect. As shown in Table 3, reasoning-focused datasets MATH500 and AIME24 tend to lead ShortGPT to prune shallow layers first (e.g., 1–5), while general-purpose text PG19 shifts pruning mainly to middle or deeper layers (14–23). In contrast, the general-purpose LlaMA3.1-8B yields a stable pruning order regardless of calibration data. Our findings indicate that calibration data can affect identifying layer importance notably in specialized models and tasks. Concurrently, LaCo proves more robust. Table 4 shows that LaCo selects nearly identical sets of layers across domains for LlaMA3.1-8B and Qwen3-8B, with only modest variation in s1.1-7B. Unlike ShortGPT, it avoids drastic variations in identifying layer importance, underscoring that direct-removal pruning method may demand in-domain calibration data, while merging-based pruning method may be comparatively insensitive to calibration choice.

## 6 CONCLUSION

This work provides new insights into how layer pruning reshapes long-chain reasoning in LLMs through the lens of test-time scaling. We find that even minimal layer removal can collapse test-time scaling, especially sequential test-time scaling, causing severe failures on reasoning-intensive benchmarks despite stability on knowledge tasks. Moreover, supervised fine-tuning has limited effect on recovering the lost test-time scaling. Our in-depth analyses reveal recurring loops, reduced trajectory diversity and diminished self-reflection of model generation after layer pruning, pointing to structural damage rather than surface-level accuracy loss.

Overall, our results highlight a fundamental trade-off: efficiency gains from layer pruning often undermine the very mechanisms that enable strong reasoning. We call for caution in applying layer pruning to reasoning-centric settings and advocate for layer pruning methods that explicitly safeguard test-time scaling. Future work could explore hybrid strategies that balance efficiency with robustness, ensuring that pruning preserves both performance and reasoning depth.

## 7 ETHICS STATEMENT

In this work, we carefully ensure that all methods and experimental protocols conform to established ethical guidelines. Our investigation centers on layer pruning as a strategy to improve the efficiency of LLMs and to lower computational demands, contributing to more sustainable AI practices. In addition, every model and dataset employed in this research is obtained from openly accessible sources, guaranteeing respect for intellectual property and protection of personal privacy. Except for the the models used as experimental subjects (Qwen3-8B and s1.1-7B) and the APIs employed for evaluation (GLM-4.5-Flash and GPT-4o-mini), we only used LLMs as writing assistants, as detailed in Section A. All uses of LLMs in this work comply with the ICLR Code of Ethics.

## 8 REPRODUCIBILITY STATEMENT

We made several efforts to ensure reproducibility. First, we provide detailed pruning method settings used throughout this paper in Appendix B. Second, we controlled for stochasticity by running experiments with three fixed random seeds to ensure result stability. Finally, the code and detailed instructions are included in the supplementary materials to facilitate full reproduction of our results.

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

## A  THE USE OF LARGE LANGUAGE MODELS

We used large language models solely as a general-purpose writing aid to help improve the clarity and readability of the text, and to suggest minor wording improvements. The LLMs did not contribute to the research ideation, experimental design, analysis, or interpretation of results. All technical content, experiments, and conclusions presented in this paper are entirely the work of the authors.

*Table 5:* Layer pruning order used for pruning and merging experiments.

| Model | ShortGPT | Reverse-order | LaCo |
|-------|----------|---------------|------|
| **Qwen3-8B** | [20, 17] | [34, 33] | [34, 29] |
| **s1.1-7B** | [16, 17] | [26, 25] | [22, 18] |

# B   DETAILS ABOUT SHORTGPT, REVERSE-ORDER, LACO, AND THEIR IMPLEMENTATIONS

We select three representative layer pruning strategies, ShortGPT, Reverse-order, and LaCo, due to their superior empirical performance.

**ShortGPT.** ShortGPT proposes a layer pruning strategy using Block Influence (BI) to indicate layer importance. The BI score of the $i$-th layer is defined as

$$\text{BI}_i = 1 - \mathbb{E}_{X,t} \frac{X_{i,t}^\top X_{i+1,t}}{\|X_{i,t}\|_2 \, \|X_{i+1,t}\|_2},$$

where $X_i$ denotes the input to the $i$-th layer, and $X_{i,t}$ is the $t$-th row of $X_i$. BI measures the degree of alignment between consecutive layer activations: a lower BI indicates higher inter-layer correlation, suggesting that the corresponding block contributes less to information propagation and can be pruned with limited performance degradation.

**Reverse-order.** Reverse-order pruning assigns lower importance scores to deeper layers, assuming that tail layers contribute less to the overall performance of LLMs. Because the final output layer often plays a critical role in generation and prediction and pruning it causes a sharp performance drop, in this work, we retain the final layer and begin pruning backward from the penultimate layer.

**LaCo.** LaCo is a layer-collapsing algorithm merges parameters that are highly similar, thereby reducing redundancy across depth. Specifically, given a target layer $l$ and $m$ subsequent layers to be merged, the merged parameter $\theta_l^*$ is computed as

$$\theta_l^* = \theta_l + (\theta_{l+1} - \theta_l) + \cdots + (\theta_{l+m} - \theta_l) = \theta_l + \sum_{k=1}^{m} (\theta_{l+k} - \theta_l).$$

During training-free merging, the LaCo algorithm scans all pairs of consecutive layers and evaluates their similarity. Layers whose similarity exceeds a predefined threshold are collapsed, allowing us to control the number of merged layers by adjusting this threshold. We apply this method using both the default dataset from the LaCo study and a dataset sampled from the MATH500 training set.

We also provide the specific layer pruning order used for the pruning and merging experiments throughout this paper, as detailed in Table 5. To generate these sequences, we followed the calibration methodologies of the original publications. For the ShortGPT method, we used a sample of 10,000 instances from the PG19 dataset. For the LaCo method, we used 5 data samples from wikitext, consistent with its official repository.

# C   ADDITIONAL RESULTS FOR SEQUENTIAL SCALING OF PRUNED QWEN3-8B IN SECTION 3

Figure 7 presents the additional results for sequential test-time scaling of the pruned Qwen3-8B models, complementing Section 3. The conclusions mirror those for pruned s1.1-7B: layer pruning fundamentally disrupts sequential test-time scaling. Compared with s1.1-7B, Qwen3-8B achieves stronger performance across all three datasets and exhibits greater robustness to light pruning. For example, pruning a single layer with Reverse-order or LaCo still allows the model to sustain sequential scaling on GPQA Diamond at a level comparable to the original. This indicates that Qwen3-8B, being larger and stronger, can partially absorb the damage of mild pruning. However, this robustness disappears once two layers are removed: accuracy drops sharply and sequential scaling collapses entirely. Overall, these results provide converging evidence that, even for stronger base models, layer

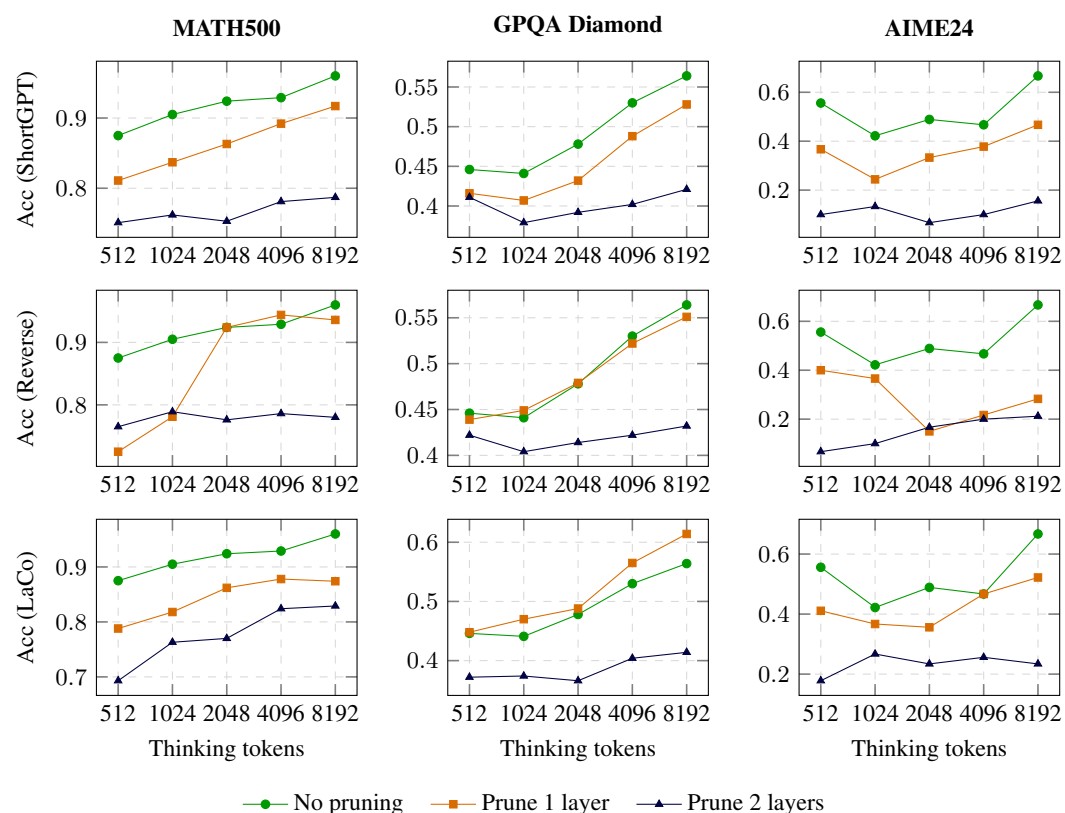

*Figure 7:* Sequential test-time scaling of Qwen3-8B under different pruning depths.

pruning consistently undermines test-time scaling once the pruning depth surpasses a very shallow threshold.

## D LEARNING RATE SEARCH IN LORA FINE-TUNING

Table 6 presents the results of grid search over learning rates of LoRA fine-tuning. We fine-tune the pruned variants of s1.1-7B and Qwen3-8B on the s1.1-1K dataset (Muennighoff et al., 2025) using LoRA with rank 16 and scaling factor 32. For each pruned configuration, we conduct a grid search over learning rates $[1 \times 10^{-6}, 4 \times 10^{-6}, 1 \times 10^{-5}, 4 \times 10^{-5}, 1 \times 10^{-4}]$ and select the best-performing value based on validation accuracy on MATH500, GPQA Diamond, and AIME24, respectively. The final results are reported on the corresponding evaluation datasets.

## E ADDITIONAL RESULTS FOR SEQUENTIAL SCALING OF THE FINE-TUNED MODELS IN SECTION 3

Figures 8, 9, and 10 provide results for sequential scaling of pruned Qwen3-8B after Full FT, pruned s1.1-7B after LoRA FT and pruned Qwen3-8B after LoRA FT. Overall, we find that both SFT methods fail to restore sequential test-time scaling effectively and Full FT consistently outperforms LoRA FT. For 2-layer pruned models, both methods provide noticeable improvements; in particular, Full FT on the 2-layer pruned Qwen3-8B can nearly restore performance to the level of its 1-layer pruned counterpart. In contrast, for 1-layer pruned models, the benefits of fine-tuning are much more limited and in some cases even negative. For example, on GPQA Diamond, fine-tuning the 1-layer pruned Qwen3-8B by LaCo or Reverse-order pruning actually reduces performance as shown in Figure 8 and Figure 10. We attribute this degradation to potential over-training on the small-scale s1.1-1K dataset, which contains only 1k samples. Importantly, regardless of the pruning depth

*Table 6:* Results of LoRA FT across different learning rates on MATH500, GPQA Diamond, and AIME24. Best results per row are in **bold**. "xxx_k_layer" means "pruned k layers by xxx" and "Reverse" denotes Reverse-order pruning for simplicity.

| Model/Method | MATH500 | | | | | GPQA Diamond | | | | | AIME24 | | | | |
|---|---|---|---|---|---|---|---|---|---|---|---|---|---|---|---|
| | 1e-6 | 4e-6 | 1e-5 | 4e-5 | 1e-4 | 1e-6 | 4e-6 | 1e-5 | 4e-5 | 1e-4 | 1e-6 | 4e-6 | 1e-5 | 4e-5 | 1e-4 |
| s1.1-7B ShortGPT_1_layer | 0.478 | 0.410 | 0.510 | 0.578 | **0.608** | 0.273 | 0.267 | 0.252 | 0.313 | **0.338** | 0.033 | 0.033 | 0.067 | **0.100** | 0.033 |
| s1.1-7B Reverse_1_layer | 0.377 | 0.403 | 0.382 | **0.446** | 0.400 | **0.292** | 0.257 | 0.247 | 0.242 | 0.247 | 0.000 | 0.033 | **0.033** | 0.033 | 0.033 |
| s1.1-7B LaCo_1_layer | 0.268 | 0.324 | 0.434 | **0.532** | 0.590 | 0.303 | 0.298 | **0.303** | 0.303 | 0.293 | 0.033 | 0.033 | 0.066 | **0.100** | 0.033 |
| s1.1-7B ShortGPT_2_layer | 0.282 | 0.374 | 0.360 | 0.454 | **0.512** | 0.253 | 0.268 | **0.308** | 0.262 | 0.293 | 0.000 | 0.000 | 0.000 | **0.033** | 0.033 |
| s1.1-7B Reverse_2_layer | 0.098 | 0.073 | 0.082 | **0.132** | 0.104 | 0.247 | 0.249 | 0.268 | **0.273** | 0.257 | 0.000 | 0.000 | **0.033** | 0.000 | 0.000 |
| s1.1-7B LaCo_2_layer | 0.198 | 0.192 | 0.302 | 0.386 | **0.454** | 0.222 | 0.273 | **0.298** | 0.283 | 0.268 | 0.067 | **0.067** | 0.033 | 0.033 | 0.000 |
| Qwen3-8B ShortGPT_1_layer | 0.876 | **0.880** | 0.800 | 0.758 | 0.760 | 0.444 | **0.444** | 0.399 | 0.399 | 0.444 | **0.433** | 0.400 | 0.300 | 0.133 | 0.167 |
| Qwen3-8B Reverse_1_layer | 0.886 | 0.918 | **0.926** | 0.822 | 0.828 | 0.477 | **0.505** | 0.419 | 0.404 | 0.449 | 0.567 | **0.600** | 0.567 | 0.233 | 0.266 |
| Qwen3-8B LaCo_1_layer | **0.934** | 0.930 | 0.884 | 0.081 | 0.844 | 0.423 | 0.485 | 0.414 | 0.429 | **0.510** | 0.167 | 0.167 | 0.133 | **0.267** | 0.200 |
| Qwen3-8B ShortGPT_2_layer | 0.282 | 0.374 | 0.360 | 0.454 | **0.512** | 0.338 | 0.328 | 0.348 | **0.374** | 0.343 | **0.200** | 0.167 | 0.100 | 0.067 | 0.100 |
| Qwen3-8B Reverse_2_layer | 0.808 | 0.918 | **0.934** | 0.776 | 0.804 | 0.409 | 0.455 | **0.484** | 0.399 | 0.414 | 0.367 | 0.267 | **0.567** | 0.467 | 0.200 |
| Qwen3-8B LaCo_2_layer | 0.686 | 0.708 | 0.572 | **0.718** | 0.710 | 0.342 | 0.378 | 0.343 | 0.369 | **0.404** | 0.067 | 0.067 | 0.100 | 0.067 | **0.167** |

or fine-tuning method, neither LoRA nor Full FT is able to recover the disrupted sequential test-time scaling, highlighting the fundamental limitations of supervised fine-tuning in restoring scaling behaviors.

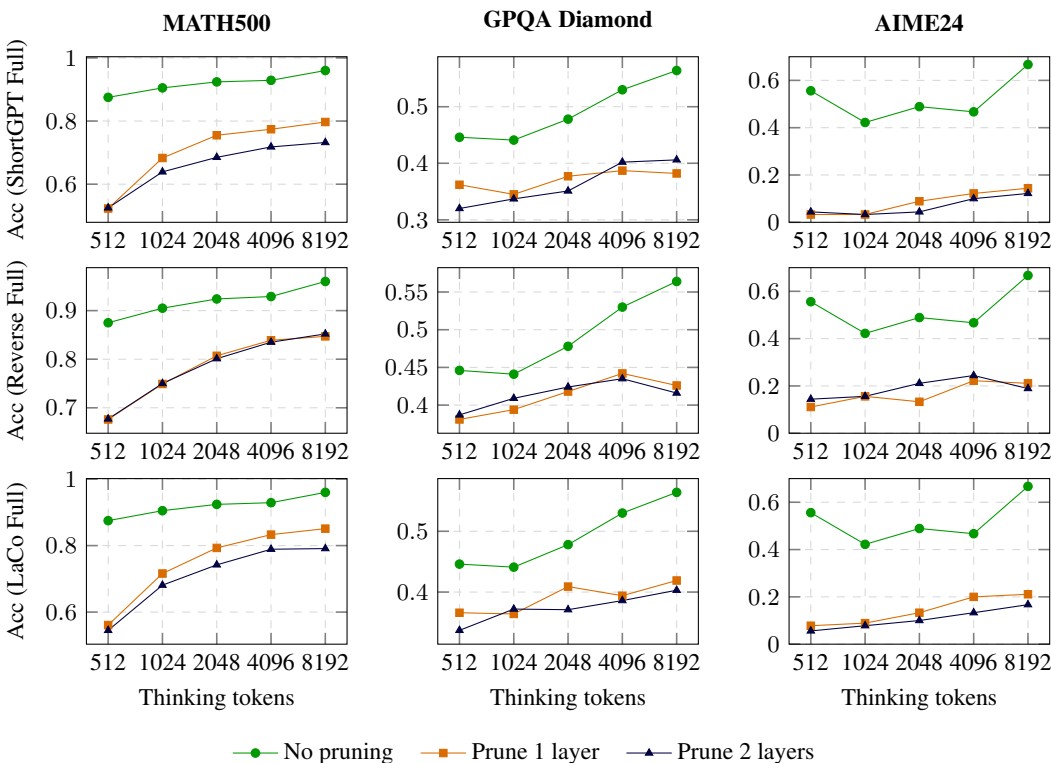

*Figure 8:* Sequential test-time scaling of Qwen3-8B after Full FT under different pruning depths. "Full" means Full FT and "Reverse" means Reverse-order in the figure for simplicity.

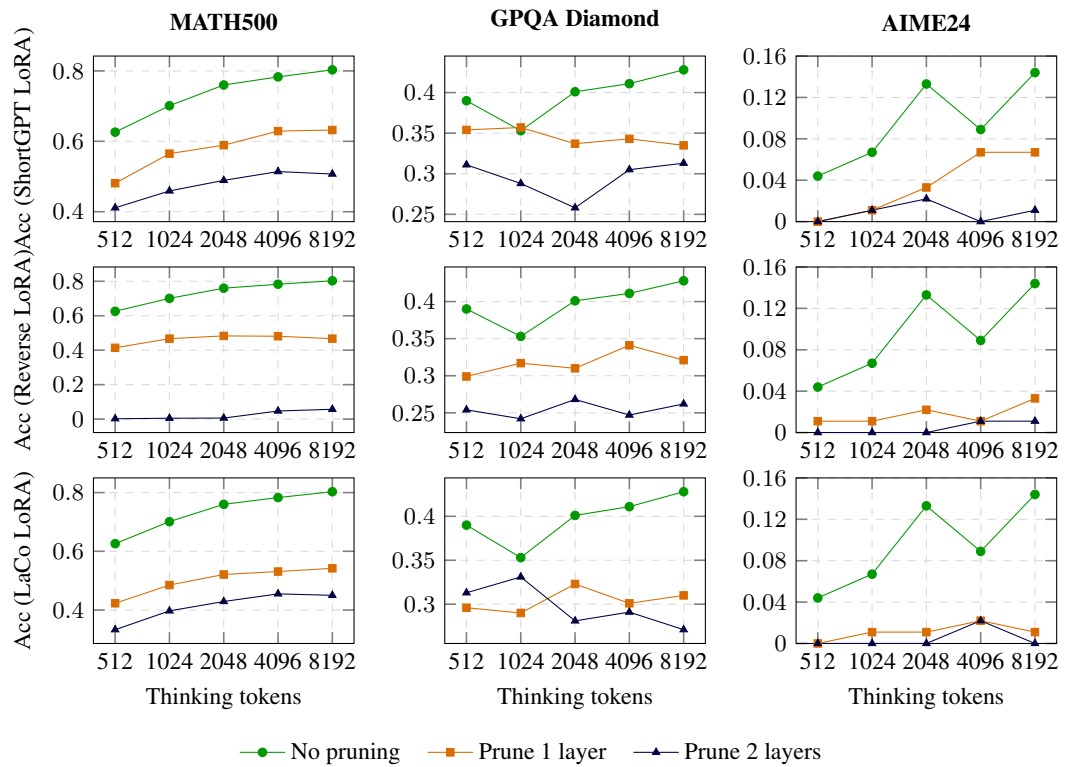

Figure 9: Sequential test-time scaling of s1.1-7B after LoRA FT under different pruning depths. "LoRA" means LoRA FT and "Reverse" means Reverse-order in the figure for simplicity.

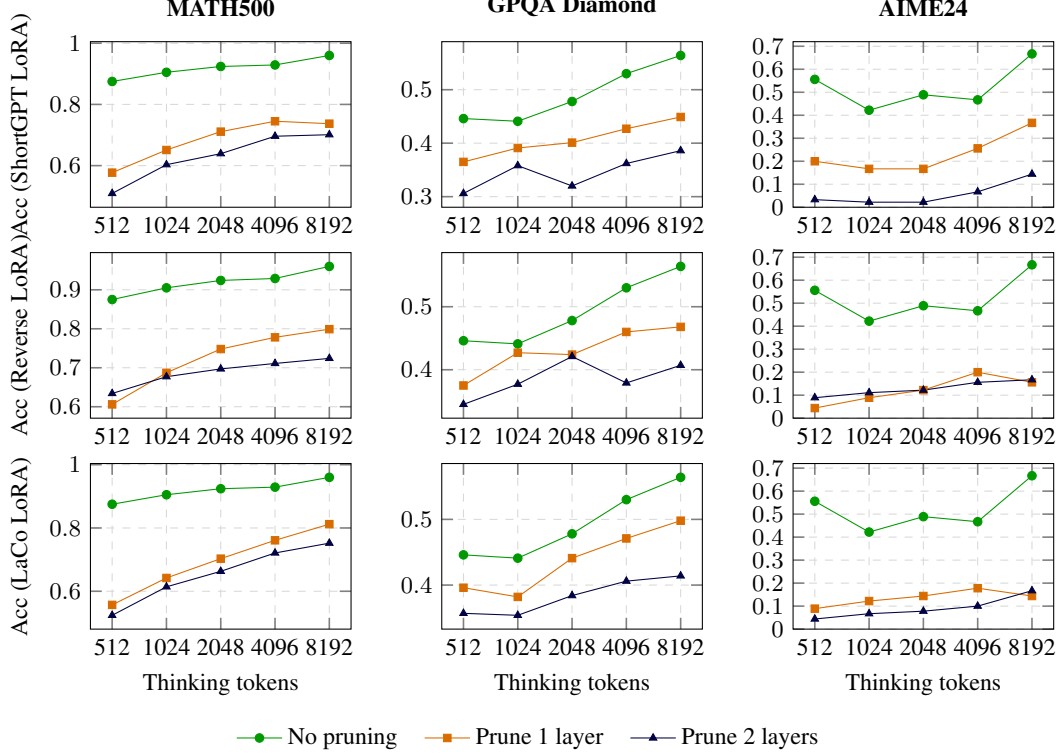

Figure 10: Sequential test-time scaling of Qwen3-8B after LoRA FT under different pruning depths. "LoRA" means LoRA FT and "Reverse" means Reverse-order in the figure for simplicity.

# F    BRUTE-FORCE LAYER ABLATION FOR SEQUENTIAL TEST-TIME SCALING

We prune one layer at a time and evaluate the sequential test-time scaling of the corresponding variants, as shown in Figures 11, 12, 13 and 14. Most layers make significant contributions to test-time scaling, and removing even a single layer can substantially harm this ability.

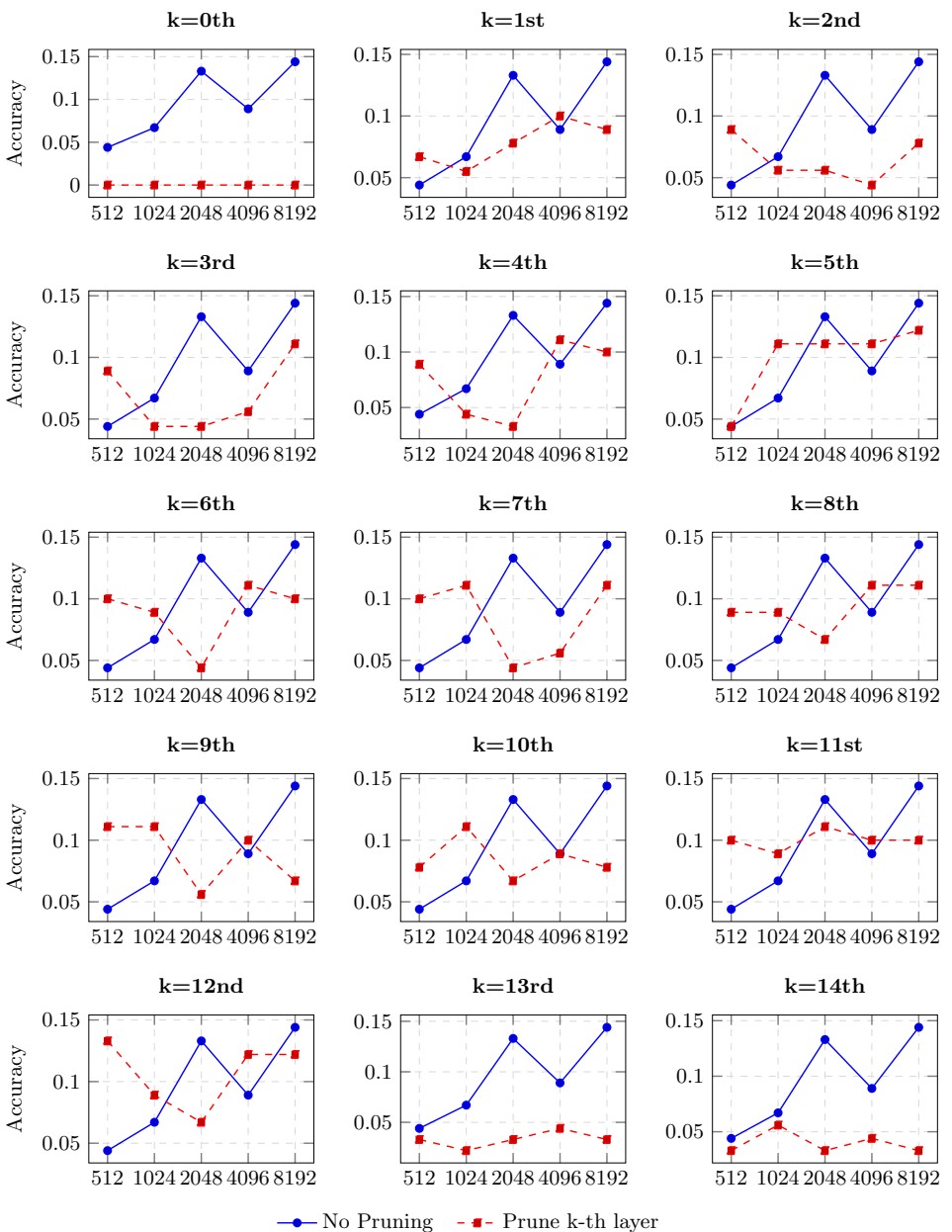

*Figure 11:* Brute-force of s1.1-7B layer ablation for sequential test-time scaling (Part 1)

# G    ADDITIONAL QUALITATIVE EXAMPLES OF REPETITIVE REASONING

We provide additional qualitative examples of the repetitive reasoning phenomenon in pruned models, as discussed in Section 5.1.

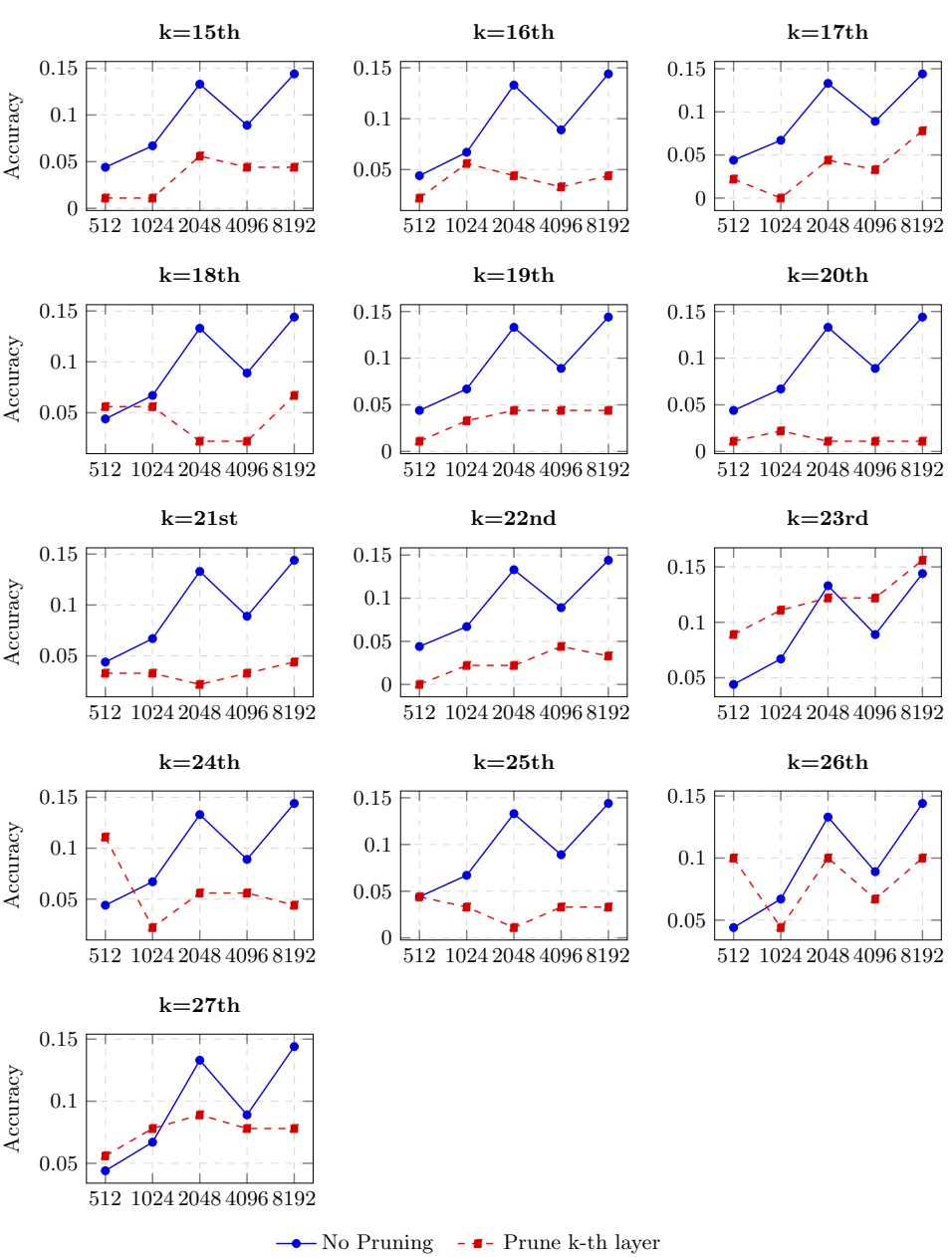

*Figure 12:* Brute-force of s1.1-7B layer ablation for sequential test-time scaling (Part 2)

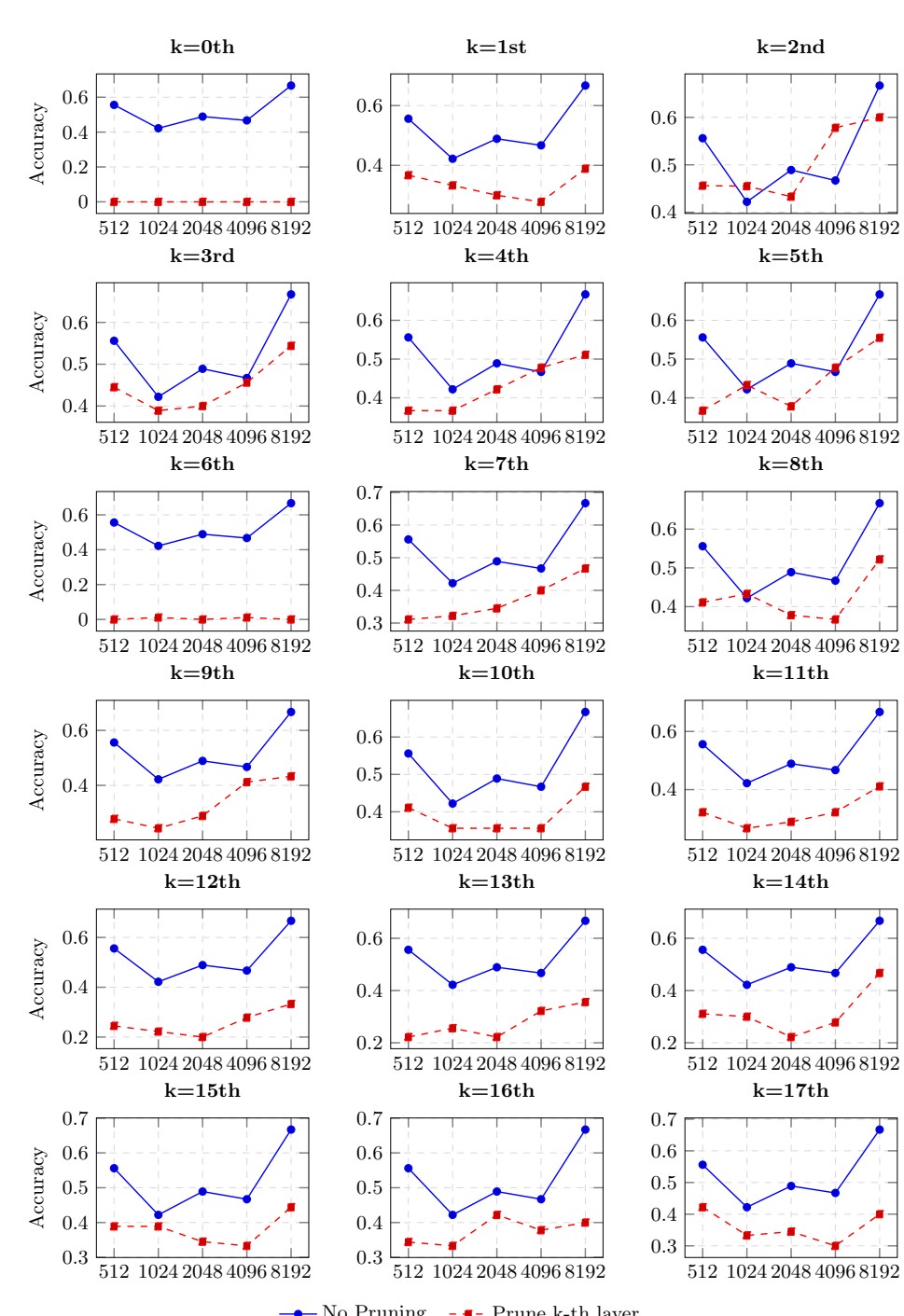

*Figure 13:* Brute-force of Qwen3-8B layer ablation for sequential test-time scaling (Part 1)

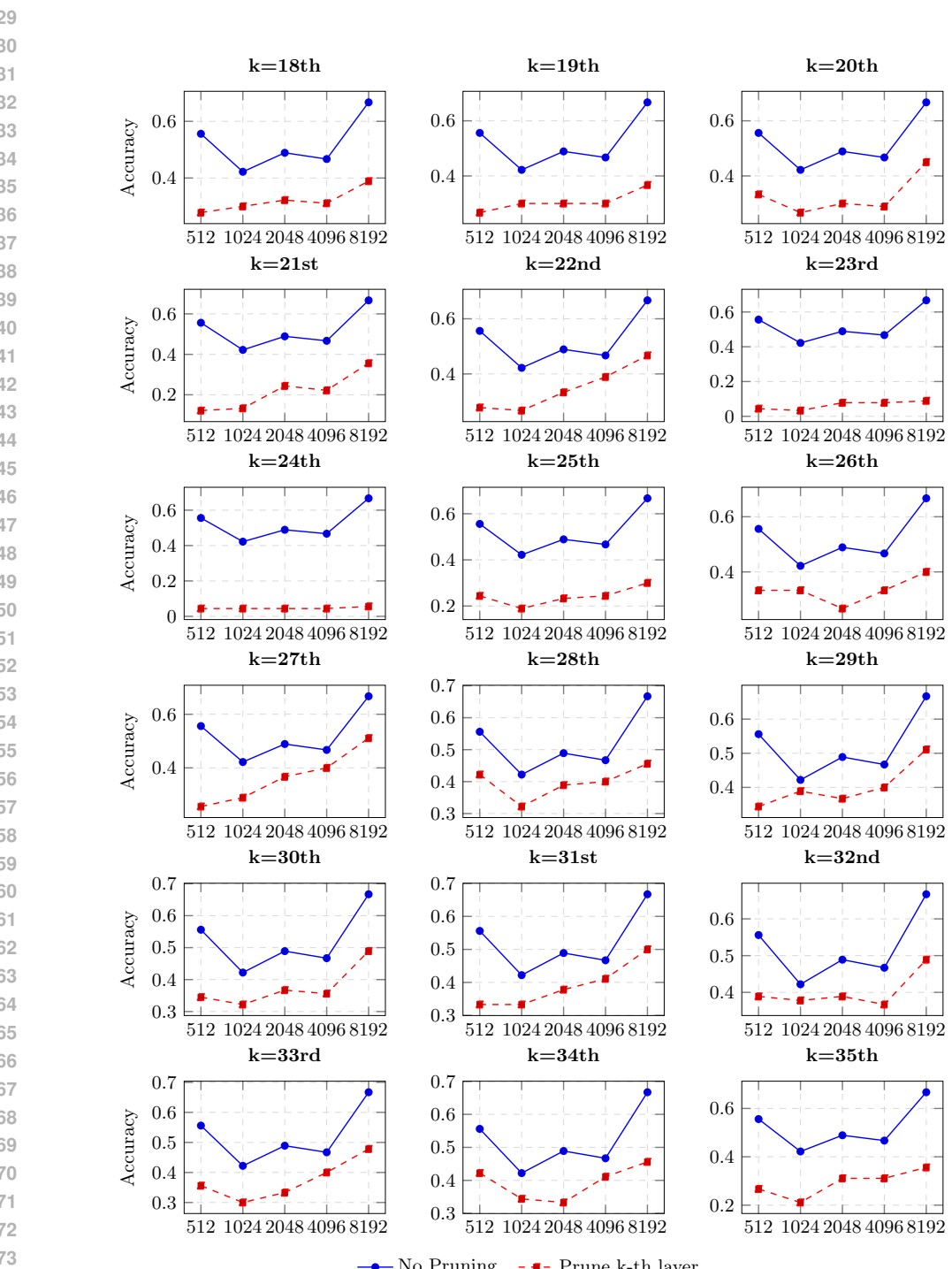

*Figure 14:* Brute-force of Qwen3-8B layer ablation for sequential test-time scaling (Part 2)

## H   COMPARISON CHARTS BEFORE AND AFTER FINE-TUNING

**Introduction to Our SFT Datasets** .  **s1K**[1]. The s1K dataset is a carefully curated collection of 1,000 high-quality, diverse, and difficult reasoning questions drawn from an initial pool of 59K problems across 51 scientific and mathematical domains, including geometry, number theory, combinatorics, physics, biology, computer science, and more . Each question is paired with a Gemini-generated reasoning trace and solution, producing a total of 4.7M tokens of step-by-step thought data. The authors select these 1,000 samples through a three-stage filtering pipeline emphasizing Quality (removing malformed or incorrect items), Difficulty (favoring long and challenging reasoning chains), and Diversity (sampling across many domains using weighted sampling that biases toward long-thinking problems) . Despite consisting of only 1K examples, s1K spans a remarkably broad range of reasoning tasks, from advanced mathematics to university-level physics and theoretical computer science, making it a highly sample-efficient training set for teaching large language models to perform complex, sequential reasoning.

**OpenR1-Math-220K**[2]. OpenR1-Math-220k is a large-scale mathematical reasoning dataset containing 220,000 NuminaMath 1.5 problems, each paired with 2–4 long-form reasoning traces generated by DeepSeek R1. The traces are rigorously validated, most via Math Verify, and about 12% judged by Llama-3.3-70B-Instruct, ensuring that every problem includes at least one correct chain-of-thought solution. The dataset provides two splits: a 94k "default" subset optimized for SFT performance, and an extended 131k subset with additional sources like cn_k12, which increases coverage but slightly lowers downstream performance due to easier questions. To construct the dataset, the authors generated step-by-step solutions for 400k problems (up to 16k tokens per trace), producing up to 300k problem solutions per day on 512 H100 GPUs, with multiple solutions enabling rejection sampling and preference-optimization training such as DPO. We randomly sampled 10k samples from the OpenR1-Math-220k dataset for Full FT on pruned s1.1-7B.

Here we provide a detailed analysis of the models' performance before and after full finetuning. We evaluate s1.1-7B using a small dataset (1K) in Figure 15 using the settings described in Section 4. Additionally, to investigate the impact of dataset scaling, we report results for s1.1-7B finetuned on a larger dataset of 10K samples extracted from OpenR1-Math-220K in Figure 16.

**Overfitting Risks on Small Datasets.**   The results on s1.1-7B with the 1K dataset highlight the risks of finetuning pruned models on limited data. As seen in Figure 15, particularly within the ShortGPT strategy, applying SFT resulted in a performance drop compared to the standard pruning baseline (solid lines falling below or barely matching dashed lines). This indicates that on smaller datasets, the pruned models are prone to overfitting; rather than "healing" the damage caused by layer removal, the model memorizes the limited training data, degrading the robust zero-shot capabilities inherent in the base model.

**Limited Recovery of Test-Time Scaling via Data Scaling.**   Scaling the finetuning data size proves essential for general accuracy recovery, yet it remains insufficient for fully restoring test-time scaling capabilities. Figure 16 demonstrates that when the dataset size is increased to 10K, the s1.1-7B model significantly recovers accuracy across most tasks, narrowing the gap between pruned and unpruned models compared to the 1K setting. However, a critical limitation persists: while absolute accuracy improves, the pruned models—particularly the **Prune 2 layer** cases—fail to fully retain the sequential test-time scaling properties of the original model. As illustrated in the "Prune 2 layer sft" curves (solid red lines) in Figure 16, the positive slope typically observed as thinking tokens increase is often flattened or erratic. This suggests that while larger-scale SFT can recover specific knowledge or formatting, it struggles to repair the model's underlying mechanism for iterative reasoning scaling after aggressive layer removal.

---

[1]https://huggingface.co/datasets/simplescaling/s1K-1.1
[2]https://huggingface.co/datasets/open-r1/OpenR1-Math-220k

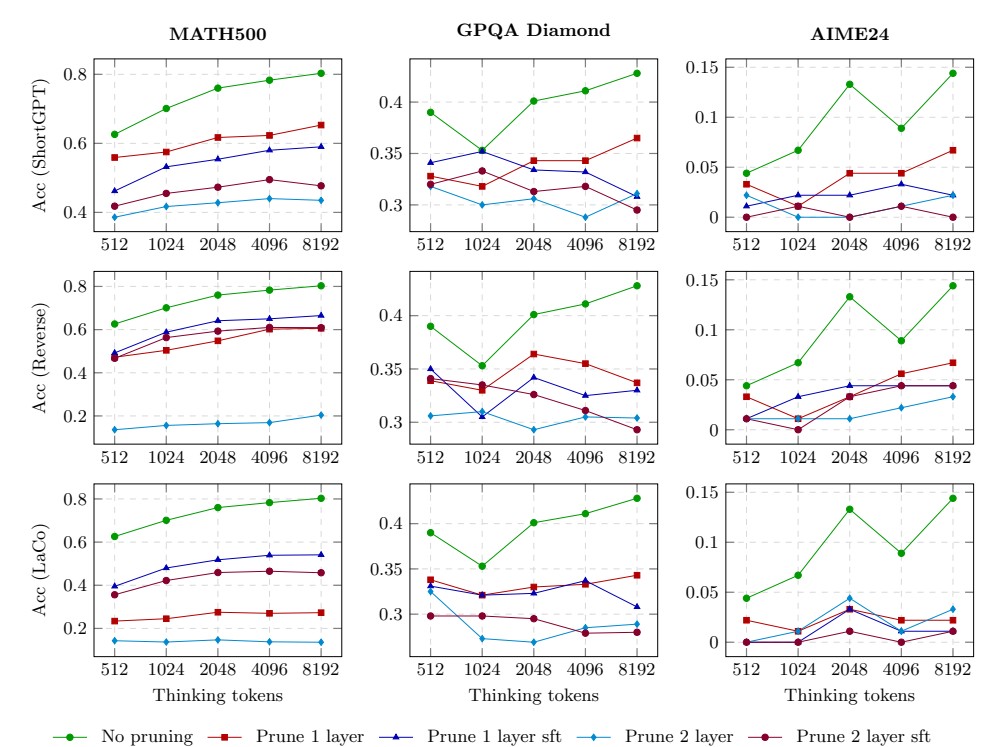

*Figure 15:* Comparison of Sequential test-time scaling of s1.1-7B under different pruning strategies: Standard Pruning vs. SFT Pruning (1K Data).

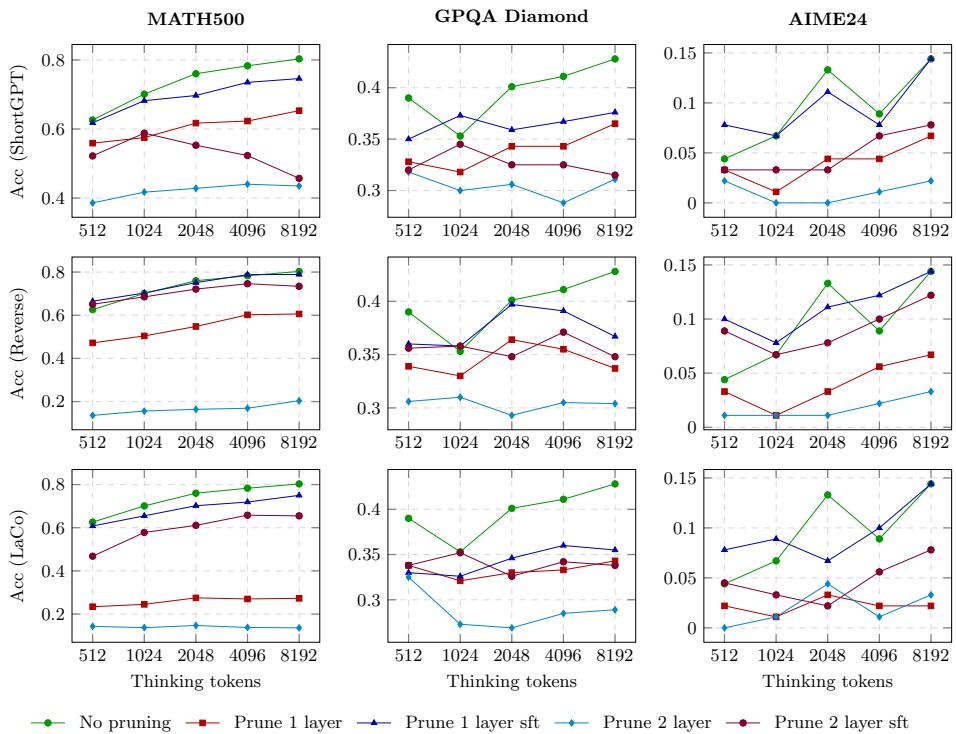

*Figure 16:* Comparison of Sequential test-time scaling of s1.1-7B under different pruning strategies: Standard Pruning vs. SFT Pruning on larger dataset (10K Data).

## I NON-REASONING MODELS

We apply non-thinking modes of s1.1-7B and qwen3-8B by setting the budgt of thinking tokens to 0. Additionally, we use Llama3.1-8B which doesn't have intermediate reasoning traces. Other settings are the same with Section 3. Figure 17 show that non-reasoning models perform substantially worse, and that layer pruning similarly causes significant degradation on reasoning tasks for these non-reasoning models. This is fully consistent with our claim. Moreover, the stronger a model's inherent reasoning ability is, the more severely it is affected by layer pruning. In contrast, Llama3.1-8B, which lacks strong reasoning capabilities to begin with, experiences the smallest performance drop after pruning.

## J STUDIES ON SAMPLING STRATEGIES: TEMPERATURES AND PENALTY-BASED SAMPLING

We conducted extensive additional experiments on temperature scaling (Figure 18 and Figure 19 ), repetition penalty (Figure 20 and Figure 21 ), and frequency penalty (Figure 22 and Figure 23 ), showing that all of these factors fail to mitigate the reasoning degradation caused by layer pruning.

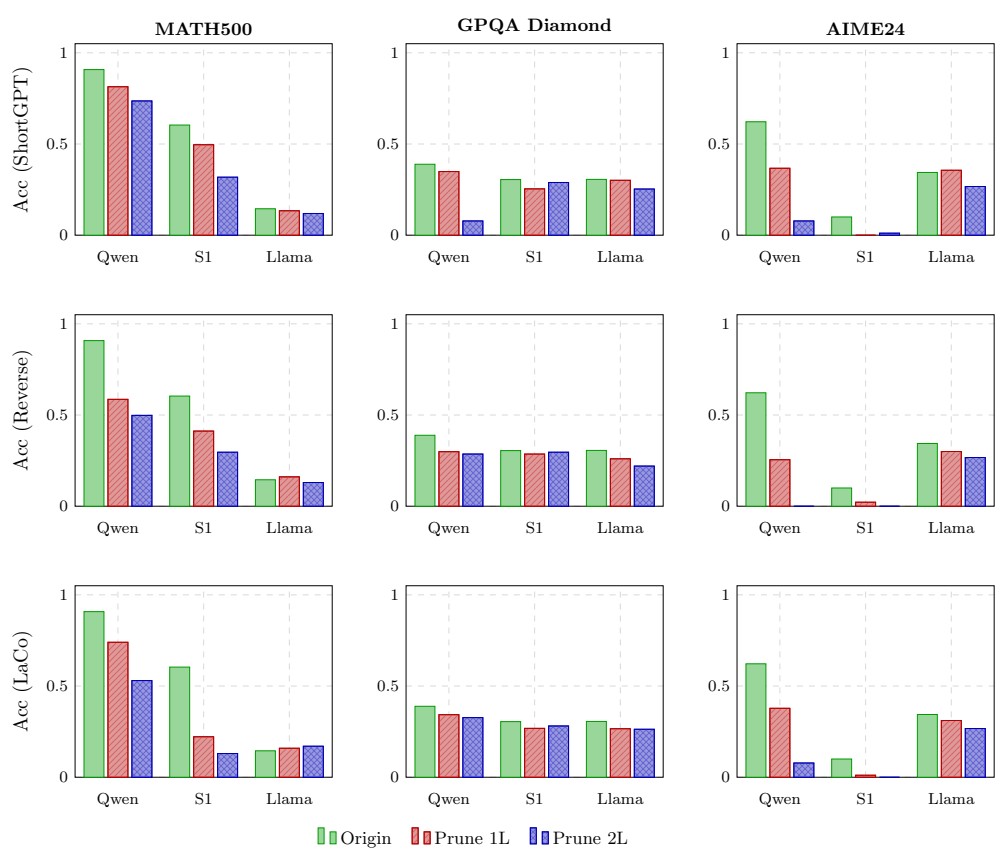

*Figure 17:* Comparison of non-reasoning model accuracy.

## K TRAINING-BASED PRUNING ANALYSIS

In this section, we examine whether training-based pruning can mitigate the loss of reasoning ability associated with layer removal. We focus on a method we term *Iterative LoRA*, wherein LoRA fine-tuning is applied using the s1K-1.1 dataset immediately following each pruning step. This experiment aims to decouple the loss of static knowledge from the loss of reasoning capabilities.

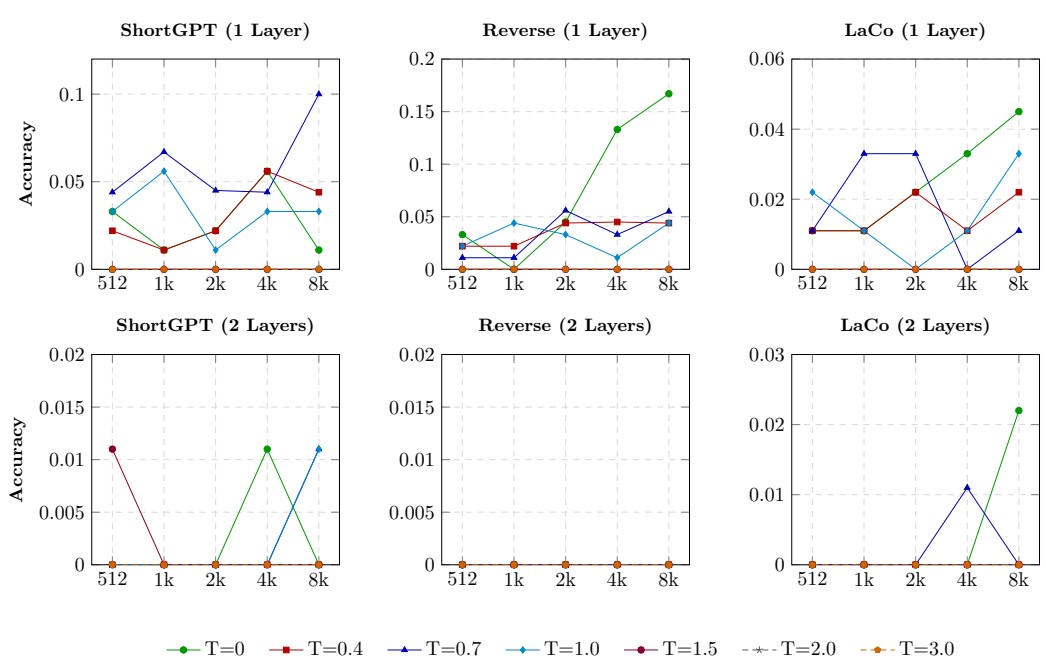

*Figure 18:* Sequential test-time scaling of non-reasoning s1.1-7B under different pruning depths and temperature.

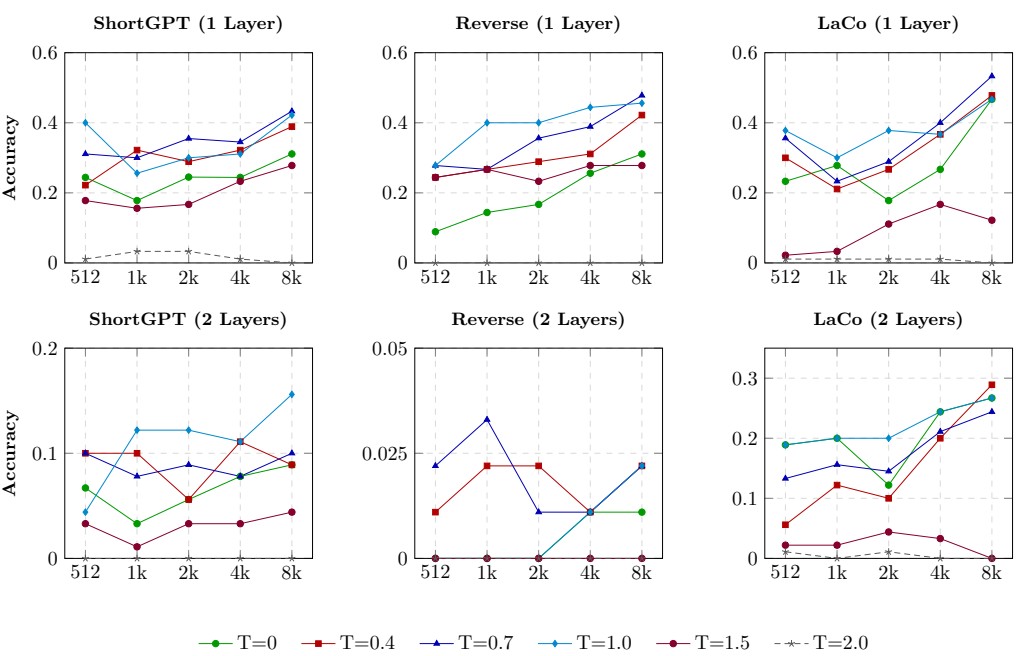

*Figure 19:* Sequential test-time scaling of non-reasoning Qwen3-8B under different pruning depths and temperature.

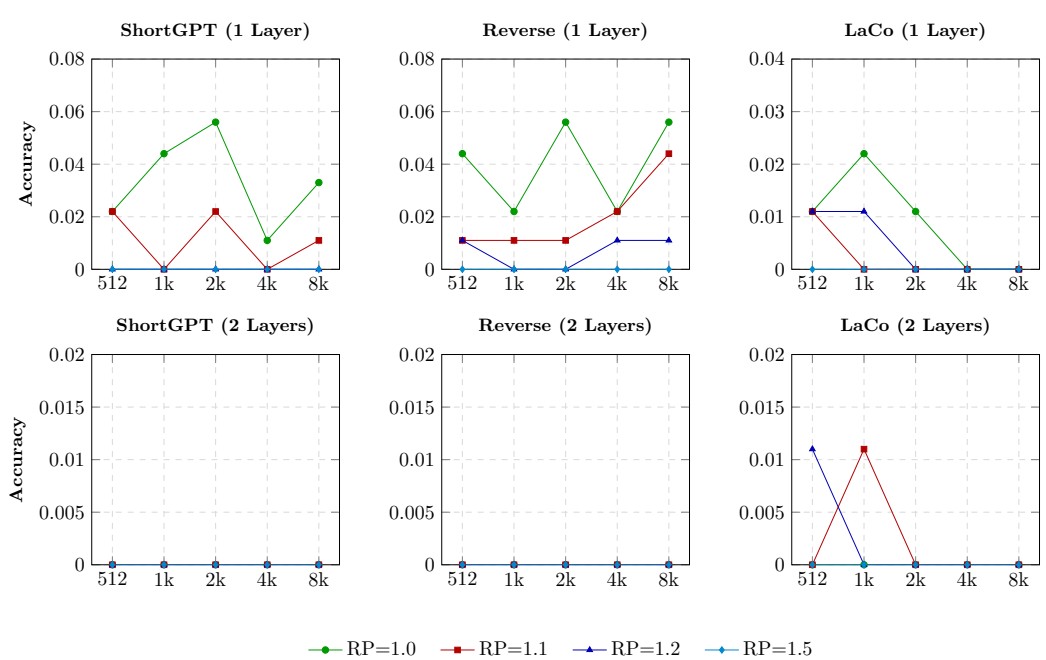

*Figure 20:* Sequential test-time scaling of non-reasoning s1.1-7B under different pruning depths and repetition panelty.

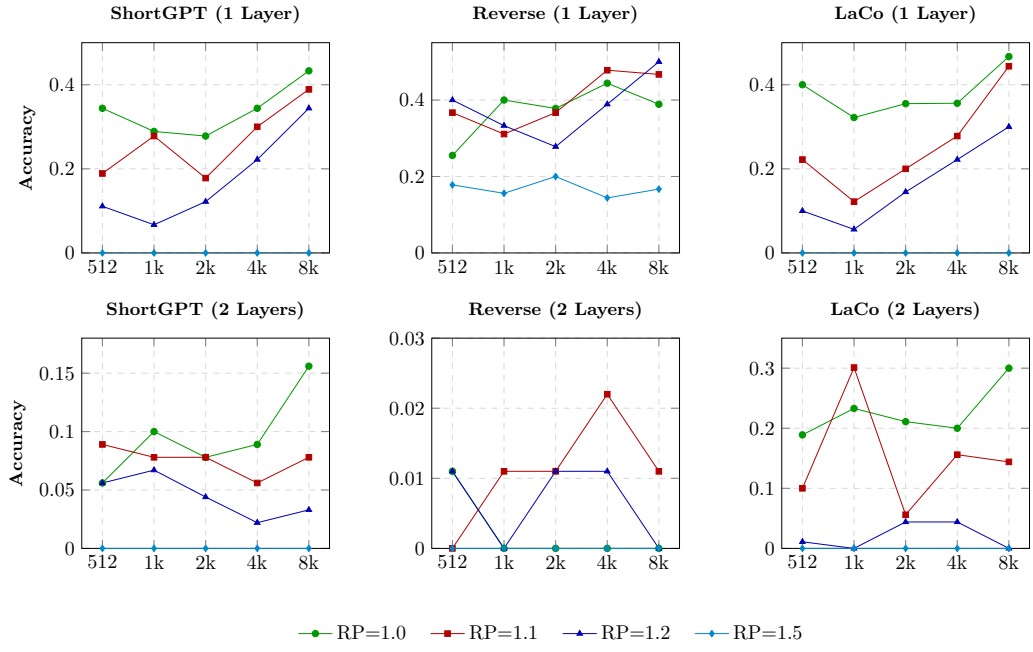

*Figure 21:* Sequential test-time scaling of non-reasoning Qwen3-8B under different pruning depths and repetition panelty.

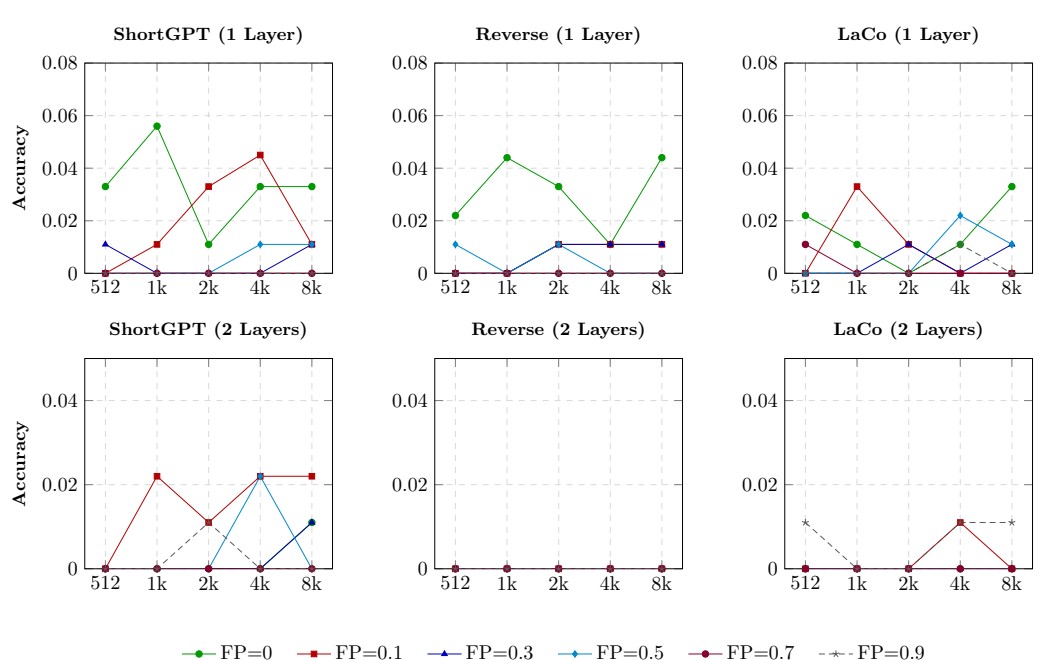

*Figure 22:* Sequential test-time scaling of non-reasoning s1.1-7B under different pruning depths and frequency panelty.

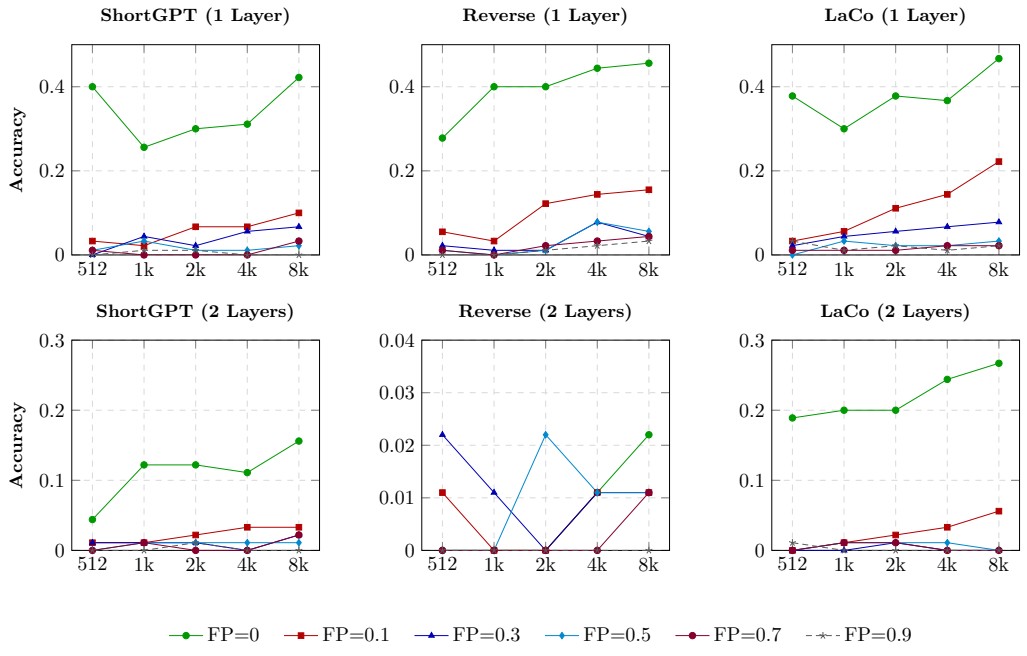

*Figure 23:* Sequential test-time scaling of non-reasoning Qwen3-8B under different pruning depths and frequency panelty.

Our empirical results indicate that while retraining can partially recover standard inference performance, it fails to restore the model's sequential scaling properties.

Figure 24 presents the scaling trajectories for the s1.1-7B model. The unpruned baseline (green line) exhibits strong positive scaling on MATH500 and GPQA Diamond as test-time compute increases. Conversely, the pruned variants display significant scaling saturation. Although Iterative LoRA improves the base accuracy (y-intercept) compared to raw pruning, the slopes of the performance curves remain flattened. This suggests that the pruned models struggle to effectively leverage additional search steps, regardless of parameter recovery via fine-tuning.

This limitation is further corroborated by the Qwen3-8B results shown in Figure 25. Across the AIME24 and MATH500 benchmarks, the performance gap between the unpruned baseline and the pruned models widens as the token budget increases. The pruned models exhibit diminished returns from test-time scaling, indicating that layer pruning likely induces a structural degradation that prevents the model from engaging in deep reasoning, a deficit that cannot be rectified through standard post-pruning fine-tuning.

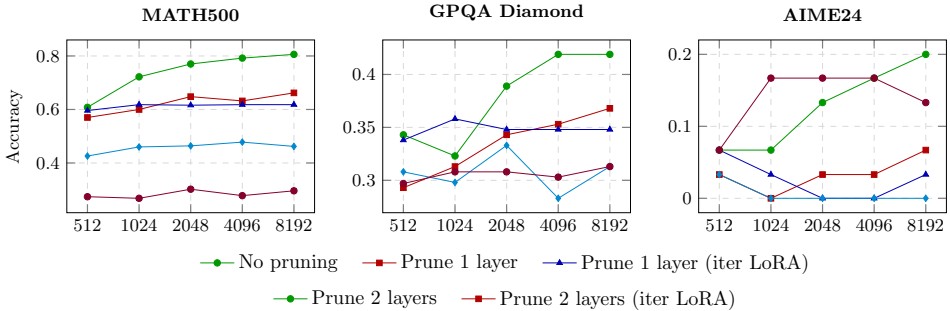

*Figure 24:* Sequential test-time scaling of iteratively tuned s1.1-7B. While unpruned models scale effectively with more compute, pruned variants show saturated performance curves despite fine-tuning.

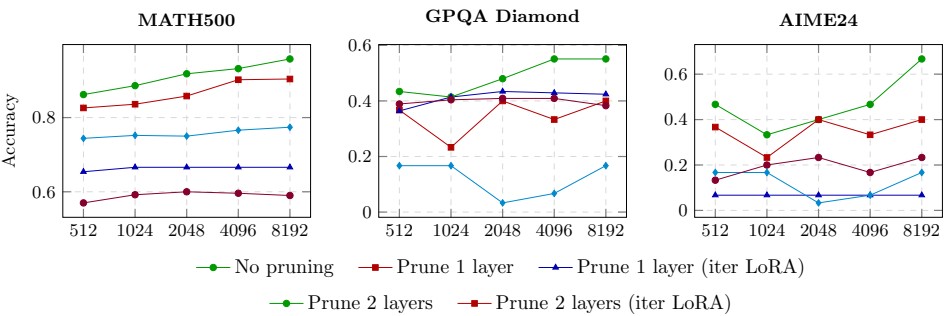

*Figure 25:* Sequential test-time scaling of iteratively tuned Qwen3-8B. The results demonstrate that pruning hinders the model's capacity for test-time scaling, resulting in widening performance gaps at higher token budgets.

## L  IMPACT OF PRUNING ON TEST-TIME SCALING IN LARGER MODELS

To verify whether our findings generalize to larger architectures, we extended our analysis to Qwen3-14B and GPT-OSS-20B. Our results confirm that pruning detrimentally affects test-time scaling capabilities, manifested either as consistent performance degradation or complete generation collapse.

## QUANTITATIVE DEGRADATION ON QWEN3-14B

Figure 26 presents the sequential test-time scaling of Qwen3-14B under varying pruning depths. Across all datasets and evaluation protocols, pruning one or more layers consistently degrades performance compared to the unpruned model. These results demonstrate that pruning inevitably impairs the model's ability to leverage longer thinking sequences, confirming that the trade-off between sparsity and inference scaling observed in smaller models extends to large-scale LLMs.

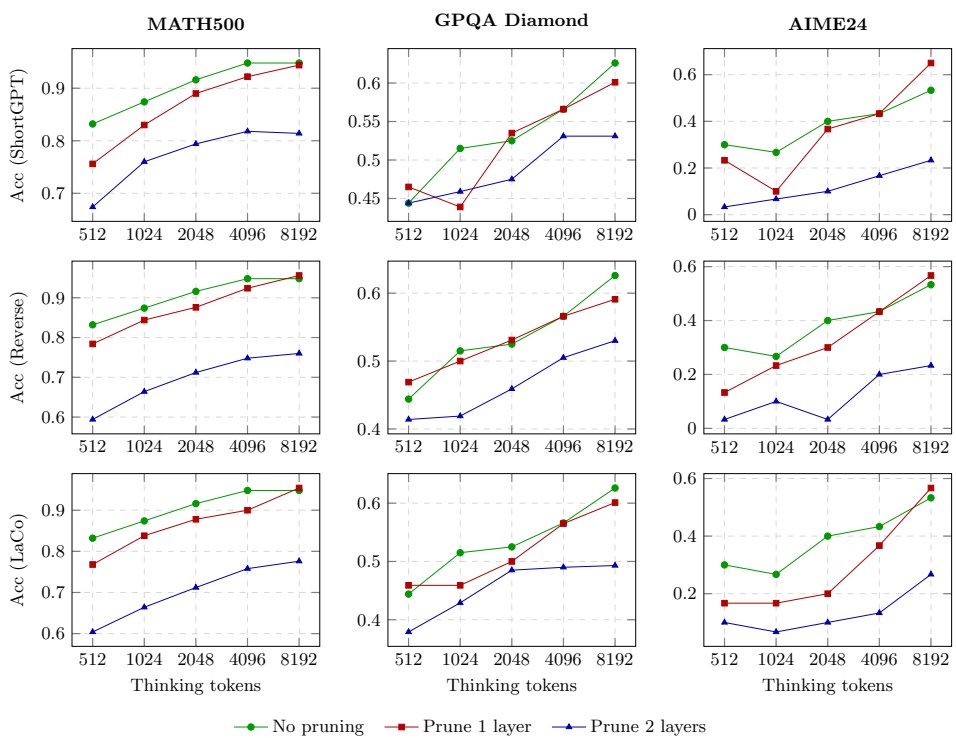

*Figure 26:* Sequential test-time scaling of Qwen3-14B under different pruning depths.

## GENERATION COLLAPSE ON GPT-OSS-20B

We further observed significantly more severe degradation when applying ShortGPT to the GPT-OSS-20B model. While the failure modes discussed in Section 5.1 were characterized by recursive logic loops, the pruned GPT-OSS-20B model demonstrates complete generation collapse. As illustrated in Figure 27, the output is devoid of semantic meaning, consisting largely of repetitive control tokens (e.g., `<|im_start|>`) and incoherent sequences rather than valid reasoning traces. This indicates that for certain architectures, pruning destroys the fundamental mechanisms required for test-time compute scaling.

| [MATH500] | [GPQA Diamond] | [AIME24] |
|---|---|---|
| Find the number of integer values of $k \in [-500, 500]$ for which $\log(kx) = 2\log(x+2)$ has exactly one real solution. | Two quantum states with lifetimes $10^{-9}$ s and $10^{-8}$ s need to be clearly resolved. What is a possible energy difference? | Let $p$ be the least prime for which $n^4 + 1 \equiv 0 \pmod{p^2}$. Find the least positive $m$ such that $m^4 + 1 \equiv 0 \pmod{p^2}$. |

| | | |
|---|---|---|
| *<\|im_start\|>think*
*<\|im_start\|>answer*
*<\|im_start\|>im_start\|>*
*kim_start\|>*
*im_start\|>kim_start\|>*
*im_start\|>kim_start\|>*
*im_start\|>kim_start\|>*
*im_start\|>kim_start\|>*
*im_start\|>kim_start\|>*
*im_start\|>...* | *<\|im_start\|>think*
*<\|>assistant*
*<\|im_start\|>answer*
*<\|im_start\|>assistant*
*<\|im_start\|>answer*
*<\|im_start\|>assistant*
*<\|im_start\|>answer*
*<\|im_start\|>assistant*
*<\|im_start\|>answer...* | *<\|im_start\|>think,*
*<\|im_start\|>answer*
*<\|im_start\|>assistant*
*<\|im_start\|>assistant*
*<\|im_start\|>assistant*
*<\|im_start\|>assistant*
*<\|im_start\|>im_start\|>*
*assistant<\|im_start\|>*
*im_start\|>assistant...* |

*Figure 27:* Qualitative examples of generation collapse in the 1-layer pruned GPT-OSS-20B model. Unlike the unpruned baseline, the pruned model exhibits severe degeneration, frequently entering repetitive loops of control tokens (e.g., `<|im_start|>`) instead of generating coherent reasoning traces.

*Table 7:* **Failure Analysis of a Pruned s1.1-7B Model on an AIME24 Problem.** The table shows the reference solution (yellow) against the model's generated reasoning (light blue) and answer (blue). The reasoning trace highlights a failure where the model makes a calculation error and enters a self-contradictory loop.

Let $p$ be the least prime number for which there exists a positive integer $n$ such that $n^4 + 1$ is divisible by $p^2$. Find the least positive integer $m$ such that $m^4 + 1$ is divisible by $p^2$.

If $p = 2$, then $4 \mid n^4 + 1$ for some integer $n$. But $\left(n^2\right)^2 \equiv 0$ or $1 \pmod 4$, so it is impossible. Thus $p$ is an odd prime. For integer $n$ such that $p^2 \mid n^4 + 1$, we have $p \mid n^4 + 1$, hence $p \nmid n^4 - 1$, but $p \mid n^8 - 1$. By [Fermat's Little Theorem](https://artofproblemsolving.com/wiki/index.php/Fermat

$$p \mid \gcd\left(n^{p-1} - 1, n^8 - 1\right) = n^{\gcd(p-1,8)} - 1.$$

Here, $\gcd(p - 1, 8)$ mustn't be divide into $4$ or otherwise $p \mid n^{\gcd(p-1,8)} - 1 \mid n^4 - 1$, which contradicts. So $\gcd(p - 1, 8) = 8$, and so $8 \mid p - 1$. The smallest such prime is clearly $p = 17 = 2 \times 8 + 1$. So we have to find the smallest positive integer $m$ such that $17 \mid m^4 + 1$. We first find the remainder of $m$ divided by $17$ by doing

| $x \pmod{17}$ | 1 | 2 | 3 | 4 | 5 | 6 | 7 | 8 | 9 | 10 | 11 | 12 | 13 | 14 | 15 | 16 |
|---|---|---|---|---|---|---|---|---|---|---|---|---|---|---|---|---|
| $x^4 + 1 \pmod{17}$ | 2 | 0 | 14 | 2 | 14 | 5 | 5 | 0 | 0 | 5 | 5 | 14 | 2 | 14 | 0 | 2 |

So $m \equiv \pm 2, \pm 8 \pmod{17}$. If $m \equiv 2 \pmod{17}$, let $m = 17k + 2$, by the binomial theorem,

$$0 \equiv (17k + 2)^4 + 1 \equiv \binom{4}{1}(17k)(2)^3 + 2^4 + 1 = 17(1 + 32k) \pmod{17^2}$$

$$\implies 0 \equiv 1 + 32k \equiv 1 - 2k \pmod{17}.$$

So the smallest possible $k = 9$, and $m = 155$. If $m \equiv -2 \pmod{17}$, let $m = 17k - 2$, by the binomial theorem,

$$0 \equiv (17k - 2)^4 + 1 \equiv \binom{4}{1}(17k)(-2)^3 + 2^4 + 1 = 17(1 - 32k) \pmod{17^2}$$

$$\implies 0 \equiv 1 - 32k \equiv 1 + 2k \pmod{17}.$$

So the smallest possible $k = 8$, and $m = 134$. If $m \equiv 8 \pmod{17}$, let $m = 17k + 8$, by the binomial theorem,

$$0 \equiv (17k + 8)^4 + 1 \equiv \binom{4}{1}(17k)(8)^3 + 8^4 + 1 = 17(241 + 2048k) \pmod{17^2}$$

$$\implies 0 \equiv 241 + 2048k \equiv 3 + 8k \pmod{17}.$$

So the smallest possible $k = 6$, and $m = 110$. If $m \equiv -8 \pmod{17}$, let $m = 17k - 8$, by the binomial theorem,

$$0 \equiv (17k - 8)^4 + 1 \equiv \binom{4}{1}(17k)(-8)^3 + 8^4 + 1 = 17(241 - 2048k) \pmod{17^2}$$

$$\implies 0 \equiv 241 + 2048k \equiv 3 + 9k \pmod{17}.$$

So the smallest possible $k = 11$, and $m = 179$. In conclusion, the smallest possible $m$ is $\boxed{110}$. Solution by Quantum-Phantom We work in the ring $\mathbb{Z}/289\mathbb{Z}$ and use the formula

$$\sqrt[4]{-1} = \pm\sqrt{\frac{1}{2}} \pm \sqrt{-\frac{1}{2}}.$$

Since $-\frac{1}{2} = 144$, the expression becomes $\pm 12 \pm 12i$, and it is easily calculated via Hensel that $i = 38$, thus giving an answer of $\boxed{110}$. Note that $n^4 + 1 \equiv 0 \pmod p$ means $\operatorname{ord}_p(n) = 8 \mid p - 1$. The smallest prime that does this is $17$ and $2^4 + 1 = 17$ for example. Now let $g$ be a primitive root of $17^2$. The satisfying $n$ are of the form, $g^{\frac{p(p-1)}{8}}, g^{3\frac{p(p-1)}{8}}, g^{5\frac{p(p-1)}{8}}, g^{7\frac{p(p-1)}{8}}$. So if we find one such $n$, then all $n$ are $n, n^3, n^5, n^7$. Consider the $2$ from before. Note $17^2 \mid 2^{4 \cdot 17} + 1$ by LTE. Hence the possible $n$ are, $2^{17}, 2^{51}, 2^{85}, 2^{119}$. Some modular arithmetic yields that $2^{51} \equiv \boxed{110}$ is the least value.

**[Repetitive generation with 8192 thinking tokens]**

Okay, so I need to find the least prime number $p$ such that there exists a positive integer $n$ where $p^2$ divides $n^4 + 1$. Then, once I find that $p$, I have to find the smallest positive integer $m$ such that $m^4 + 1$ is divisible by $p^2$.

First, let me recall some number theory concepts that might help here. I remember that for primes $p$, if $p$ divides $n^4 + 1$, then $n^4 \equiv -1 \pmod{p}$. That means $n^8 \equiv 1 \pmod{p}$, right? Because if $n^4 \equiv -1 \pmod{p}$, then squaring both sides gives $n^8 \equiv 1 \pmod{p}$. So the order of $n$ modulo $p$ divides 8. But primes $p$ for which there exists an integer $n$ with $n^4 \equiv -1 \pmod{p}$ must have that 8 is the order. Wait, not necessarily 8. Wait, maybe I need to think again.

Wait, if $n^4 \equiv -1 \pmod{p}$, then $n^8 \equiv 1 \pmod{p}$. So the order of $n$ modulo $p$ must divide 8. So possible orders are divisors of 8: 1, 2, 4, 8. But if $n^4 \equiv -1 \pmod{p}$, then the order can't be 1 or 2, because if the order is 1, then $n^4 \equiv 1$, but then $n^4 \equiv -1 \pmod{p}$ would imply $1 \equiv -1 \pmod{p}$, which would mean $p = 2$, but $p$ is prime and $p = 2$ would make $n^4 + 1 \equiv 0 + 1 \equiv 1 \pmod{2}$, which is not divisible by $p^2$. So $p = 2$ is not possible. Similarly, if order is 2, then $n^2 \equiv 1 \pmod{p}$, which would imply $n^4 \equiv 1 \pmod{p}$, contradicting $n^4 \equiv -1 \pmod{p}$. So order can't be 1 or 2. If order is 4, then $n^4 \equiv 1 \pmod{p}$, which again would imply $n^4 \equiv 1 \pmod{p}$, so same problem. Therefore, order must be 8. So primes $p$ for which 8 divides the order of $n$ modulo $p$. So $p$ must be such that 8 divides the multiplicative order of $n$ modulo $p$. Wait, maybe not. Wait, if $n$ has order 8 modulo $p$, then the multiplicative group modulo $p$ is cyclic of order $p - 1$, so 8 must divide $p - 1$. So primes $p$ where 8 divides $p - 1$. So $p \equiv 1 \pmod 8$. But maybe not all primes $p \equiv 1 \pmod 8$ have this property. Wait, but the problem says "there exists a positive integer $n$ such that $p^2$ divides $n^4 + 1$". So perhaps the minimal prime $p$ is 17, since 17 is the smallest prime $\equiv 1 \pmod 8$. Wait, but 17 is a prime, so maybe 17 is the minimal $p$. Wait, but let me check.

Wait, let's see. Let's check for small primes. Let's start with $p = 2$. For $p = 2$, $n^4 + 1 \pmod 2$. If $n$ is even, then $n^4$ is 0 $\pmod 2$, so $n^4 + 1 \equiv 1 \pmod 2$, which is not divisible by 4. If $p = 2$, then $n^4 + 1 \equiv 0 \pmod 2$ would require $n$ even, so $n = 2k$. Then $(2k)^4 + 1 = 16k^4 + 1$, which is odd, so it's 1 $\pmod{16}$. So 16 divides $n^4 + 1$? Let's check $n = 1$: $1^4 + 1 = 2$, which is not divisible by 16. $n = 2$: $16 + 1 = 17$, not divisible by 16. $n = 3$: $81 + 1 = 82$, not divisible by 16. $n = 4$: $256 + 1 = 257$, which is 1 $\pmod{16}$. So $p = 2$ is not a prime. Wait, maybe $p = 2$ is not the case. Wait, but the problem says "positive integer n" so n=1,2,3,..., so $n = 1$: $1^4 + 1 = 2$, not divisible by $p = 2$ squared=4. $n = 2$: 17, not divisible by 4. $n = 3$: 82, not divisible by 4. $n = 4$: 257, which is 1 $\pmod 4$. So $p = 2$ is not a prime. So maybe the first prime is $p = 5$. Let's check for $p = 5$. If $p = 5$, then $n^4 + 1 \equiv 0 \pmod{25}$. So $n^4 \equiv -1 \pmod 5$. Let's check $n = 1$: $1^4 = 1 \equiv 1 \pmod 5$. $n = 2$: $16 \equiv 1 \pmod 5$. $n = 3$: $81 \equiv 1 \pmod 5$. $n = 4$: $256 \equiv 1 \pmod 5$. So none of them are $\equiv -1 \pmod 5$. So $p = 5$ is not a prime. Wait, maybe $p = 13$. Let's check for $p = 13$. Let's see if there exists $n$ such that $n^4 \equiv -1 \pmod{13}$. Let's check n=1:1, n=2:16 $\equiv 3$, n=3:81 $\equiv 3 \pmod{13}$, n=4:256 $\equiv -1 \pmod{13}$. Wait, 256 divided by 13 is 19*13=247, remainder 256-247=9. Wait, 256-13*19=256-247=9. So $256 \equiv 9 \pmod{13}$. Wait, 256 divided by 13 is 19*13=247, so 256-247=9. So $256 \equiv 9 \pmod{13}$. So n=4: $4^4 = 256 \equiv 9 \pmod{13}$. So $9 \equiv -1 \pmod{13}$? $9 \equiv -1 \pmod{13}$? 9 $\pmod{13}$ is 9, so no. Wait, 9 is not $\equiv -1 \pmod{13}$. So n=4: $256 \equiv 9 \pmod{13}$. So n=4: $256 \equiv 9 \pmod{13}$, which is 9. So $n^4 \equiv 9 \pmod{13}$, which is not -1. So $p = 13$ is not a prime. Wait, maybe $p = 17$. Let's check for $p = 17$. Let's see if there exists $n$ such that $n^4 \equiv -1 \pmod{17}$.

**[The logic goes wrong here: Calculation error. $256 \equiv 1 \pmod{17}$, not $-1$. The model then contradicts itself and gets stuck instead of testing other values like $n = 8$.]**

Let's check n=1:1, n=2:16, n=3:81 $\equiv 13 \pmod{17}$, n=4:256 $\equiv -1 \pmod{17}$. Wait, 256 divided by 17 is 15*17=255, so 256-255=1, so $256 \equiv 1 \pmod{17}$? Wait, 256-15*17=255 is 15*17=255, so 256-255=1. So $256 \equiv 1 \pmod{17}$.

So n=4: $4^4 = 256 \equiv 1 \pmod{17}$. So $1 \equiv -1 \pmod{17}$? No, $1 \equiv 1 \pmod{17}$. So n=4: $256 \equiv 1 \pmod{17}$. So n=4: $256 \equiv 1 \pmod{17}$. So n=4: $256 \equiv 1 \pmod{17}$.

...

So n=4: $256 \equiv 1 \pmod{17}$. So n=4: 256.

The least prime number $p$ for which there exists a positive integer $n$ such that $p^2$ divides $n^4 + 1$ is $\boxed{17}$. The least positive integer $m$ such that $m^4 + 1$ is divisible by $17^2$ is $\boxed{17}$.

Okay, so the problem is asking for the least prime number $p$ such that there exists a positive integer $n$ where $p^2$ divides $n^4 + 1$. Then, once we find that prime $p$, we need to find the smallest positive integer $m$ such that $m^4 + 1$ is divisible by $p^2$.

First, let's recall that if $p$ divides $n^4 + 1$, then $n^4 \equiv -1 \pmod{p}$. Squaring both sides gives $n^8 \equiv 1 \pmod{p}$. Therefore, the order of $n$ modulo $p$ must divide 8. So, the possible orders are divisors of 8: 1, 2, 4, or 8. For $n^4 \equiv -1 \pmod{p}$, the order must be exactly 8. Therefore, 8 must divide the multiplicative order of $n$ modulo $p$. This implies that the multiplicative group modulo $p$ has an element of order 8, so 8 divides $p - 1$. Therefore, $p \equiv 1 \pmod{8}$.

The smallest prime $p$ congruent to 1 modulo 8 is 17. Let's check if there exists an $n$ such that $17^2$ divides $n^4 + 1$.

We need to find the smallest $m$ such that $17^2$ divides $m^4 + 1$. Let's check small values of $m$:

For $m = 1$: $1^4 + 1 = 2$, not divisible by 17.

For $m = 2$: $16 + 1 = 17$, not divisible by 172.

For $m = 3$: $81 + 1 = 82$, not divisible by 172.

For $m = 4$: $256 + 1 = 257$, which is 1 $\pmod{17}$. Not divisible by 172.

For $m = 5$: $625 + 1 = 626$, which is 14 $\pmod{17}$. Not divisible by 172.

For $m = 6$: $1296 + 1 = 1297$, which is 1 $\pmod{17}$. Not divisible by 172.

For $m = 7$: $2401 + 1 = 2402$, which is 5 $\pmod{17}$. Not divisible by 172.

For $m = 8$: $4096 + 1 = 4097$, which is 0 $\pmod{17}$. Wait, $17 \times 241 = 4097$. 4097 divided by 17 is 241. So 4097 is 0 $\pmod{17}$. Therefore, 17 divides $4096 + 1$, but we need $17^2$. So 17 divides 4097, but $17^2 = 289$. Let's check 4097 divided by 289: $289 \times 14 = 4046$, so $4097 - 4046 = 51$. 51 is $3 \times 17$, so $4097 = 17 \times 241$, which is $17 \times 17 \times 14$. Wait, $17 \times 241$ is 4097. So 17 divides 4097, but $17^2$ does not. Therefore, $m = 8$: 4097 is $17 \times 241$, which is $17 \times 17 \times 14$, so 17 divides it once, but not squared. Therefore, $m = 8$ is not sufficient.

Wait, but 17 divides 4097, which is $8^4 + 1$. So $8^4 + 1 = 4097$, which is $17 \times 241$. So 17 divides 4097, but $17^2$ does not. Therefore, $m = 8$ is not the answer. Wait, but the problem says "divisible by $p^2$", so we need $m^4 + 1$ to be divisible by $17^2$. So $m = 8$: 4097 is $17 \times 241$, which is $17 \times 17 \times 14$, so 17 divides it once. Therefore, $m = 8$ is not sufficient. Therefore, $m = 8$ is not the answer.

Wait, but maybe I need to check higher $m$. Let's check $m = 17$: $17^4 + 1$. Let's compute $17^4$: $17^2 = 289$, so $17^4 = 289^2 = (289)^2$. $289^2 = 83521$. 83521 $\pmod{17}$: $17 \times 4912 = 83504$, so $83521 - 83504 = 17$. Therefore, 17 divides $83521 - 17 = 83504$. Therefore, 17 divides $17^4 - 17$, so 17 divides $17^4 - 17$. Therefore, 17 divides $17^4 + 1$? Wait, $17^4 + 1$. Let's compute $17^4$: $17^2 = 289$, so $17^4 = 289^2 = 83521$. $83521 + 1 = 83522$. 83522 divided by 17: $17 \times 4912 = 83504$. $83522 - 83504 = 18$. So $17 \times 4912 = 83504$, so $83522 - 83504 = 18$. Therefore, 17 divides $83521 - 17 = 83504$. Therefore, 17 divides $83521 - 17$, so 17 divides $83521 - 17$. Therefore, 17 divides $83521 - 17$. Therefore, 17 divides $83521 - 17$. Therefore, 17 divides $83521 - 17$. Therefore, 17 divides $83521 - 17$. Therefore, 17 divides $83521 - 17$. Therefore, 17 divides $83521 - 17$. Therefore, 17 divides $83521 - 17$. Therefore, 17 divides $83521 - 17$. Therefore, 17 divides $83521 - 17$.

...

Therefore, 17 divides $83521 - 17$. Therefore, 17 divides $83521 - 17$. Therefore, 17 divides $83521 - 17$. Therefore, 17 divides $83521 - 17$. Therefore, 17 divides $83521 - 17$. Therefore, 17 divides $83521 - 17$. Therefore, 17 divides $83521 - 1$

*Table 8:* **Failure Analysis of a Pruned Qwen3-8B Model on an AIME24 Problem.** The table shows the reference solution (yellow) against the model's generated reasoning (light blue) and answer (blue). The reasoning trace highlights a failure where the model makes a calculation error and enters a self-contradictory loop.

Let $p$ be the least prime number for which there exists a positive integer $n$ such that $n^4 + 1$ is divisible by $p^2$. Find the least positive integer $m$ such that $m^4 + 1$ is divisible by $p^2$.

If $p = 2$, then $4 \mid n^4 + 1$ for some integer $n$. But $\left(n^2\right)^2 \equiv 0$ or $1 \pmod{4}$, so it is impossible. Thus $p$ is an odd prime. For integer $n$ such that $p^2 \mid n^4 + 1$, we have $p \mid n^4 + 1$, hence $p \nmid n^4 - 1$, but $p \mid n^8 - 1$. By [Fermat's Little Theorem](https://artofproblemsolving.com/wiki/index.php/Fermat

$$p \mid \gcd\left(n^{p-1} - 1, n^8 - 1\right) = n^{\gcd(p-1,8)} - 1.$$

Here, $\gcd(p - 1, 8)$ mustn't be divide into $4$ or otherwise $p \mid n^{\gcd(p-1,8)} - 1 \mid n^4 - 1$, which contradicts. So $\gcd(p - 1, 8) = 8$, and so $8 \mid p - 1$. The smallest such prime is clearly $p = 17 = 2 \times 8 + 1$. So we have to find the smallest positive integer $m$ such that $17 \mid m^4 + 1$. We first find the remainder of $m$ divided by 17 by doing

| $x \bmod 17$ | 1 | 2 | 3 | 4 | 5 | 6 | 7 | 8 | 9 | 10 | 11 | 12 | 13 | 14 | 15 | 16 |
|---|---|---|---|---|---|---|---|---|---|---|---|---|---|---|---|---|
| $x^4 + 1 \bmod 17$ | 2 | 0 | 14 | 2 | 14 | 5 | 5 | 0 | 0 | 5 | 5 | 14 | 2 | 14 | 0 | 2 |

So $m \equiv \pm 2, \pm 8 \pmod{17}$. If $m \equiv 2 \pmod{17}$, let $m = 17k + 2$, by the binomial theorem,

$$0 \equiv (17k + 2)^4 + 1 \equiv \binom{4}{1}(17k)(2)^3 + 2^4 + 1 = 17(1 + 32k) \pmod{17^2}$$

$$\implies 0 \equiv 1 + 32k \equiv 1 - 2k \pmod{17}.$$

So the smallest possible $k = 9$, and $m = 155$. If $m \equiv -2 \pmod{17}$, let $m = 17k - 2$, by the binomial theorem,

$$0 \equiv (17k - 2)^4 + 1 \equiv \binom{4}{1}(17k)(-2)^3 + 2^4 + 1 = 17(1 - 32k) \pmod{17^2}$$

$$\implies 0 \equiv 1 - 32k \equiv 1 + 2k \pmod{17}.$$

So the smallest possible $k = 8$, and $m = 134$. If $m \equiv 8 \pmod{17}$, let $m = 17k + 8$, by the binomial theorem,

$$0 \equiv (17k + 8)^4 + 1 \equiv \binom{4}{1}(17k)(8)^3 + 8^4 + 1 = 17(241 + 2048k) \pmod{17^2}$$

$$\implies 0 \equiv 241 + 2048k \equiv 3 + 8k \pmod{17}.$$

So the smallest possible $k = 6$, and $m = 110$. If $m \equiv -8 \pmod{17}$, let $m = 17k - 8$, by the binomial theorem,

$$0 \equiv (17k - 8)^4 + 1 \equiv \binom{4}{1}(17k)(-8)^3 + 8^4 + 1 = 17(241 - 2048k) \pmod{17^2}$$

$$\implies 0 \equiv 241 + 2048k \equiv 3 + 9k \pmod{17}.$$

So the smallest possible $k = 11$, and $m = 179$. In conclusion, the smallest possible $m$ is $\boxed{110}$. Solution by Quantum-Phantom We work in the ring $\mathbb{Z}/289\mathbb{Z}$ and use the formula

$$\sqrt[4]{-1} = \pm\sqrt{\frac{1}{2}} \pm \sqrt{-\frac{1}{2}}.$$

Since $-\frac{1}{2} = 144$, the expression becomes $\pm 12 \pm 12i$, and it is easily calculated via Hensel that $i = 38$, thus giving an answer of $\boxed{110}$. Note that $n^4 + 1 \equiv 0 \pmod{p}$ means $\text{ord}_p(n) = 8 \mid p - 1$. The smallest prime that does this is 17 and $2^4 + 1 = 17$ for example. Now let $g$ be a primitive root of $17^2$. The satisfying $n$ are of the form, $g^{\frac{p(p-1)}{8}}, g^{3\frac{p(p-1)}{8}}, g^{5\frac{p(p-1)}{8}}, g^{7\frac{p(p-1)}{8}}$. So if we find one such $n$, then all $n$ are $n, n^3, n^5, n^7$. Consider the 2 from before. Note $17^2 \mid 2^{4 \cdot 17} + 1$ by LTE. Hence the possible $n$ are, $2^{17}, 2^{51}, 2^{85}, 2^{119}$. Some modular arithmetic yields that $2^{51} \equiv \boxed{110}$ is the least value.

**[Repetitive generation with 8192 thinking tokens]**

Okay, so I need to find the smallest prime number $p$ where there's a positive integer $n$ such that $p^2$ divides $n^4 + 1$. Then, once I find that $p$, I have to find the smallest positive integer $m$ such that $m^4 + 1$ is divisible by $p^2$. Alright, let me start by understanding the problem step by step.

First, let me recall that if $p^2$ divides $n^4 + 1$, then $n^4 \equiv -1 \pmod{p^2}$. That means that $-1$ is a quartic residue modulo $p^2$. So, I need primes $p$ where $-1$ is a quartic residue modulo $p^2$. Hmm, quartic residues... I remember that for primes, the existence of solutions to certain congruences depends on the prime's properties, like whether it's congruent to 1 $\pmod{\cdot}$ or not.

Let me think. For a prime $p$, if there exists an integer $n$ such that $n^4 \equiv -1 \pmod{p^2}$, then $n^4 \equiv -1 \pmod{p}$. So first, maybe I should check for primes $p$ where $-1$ is a quartic residue modulo $p$. Then, maybe check if such primes exist and find the smallest one.

But how do I check if $-1$ is a quartic residue modulo $p$? Let me recall that for a prime $p$, the multiplicative group modulo $p$ is cyclic of order $p - 1$. So, the quartic residues modulo $p$ are the elements of the group raised to the 4th power. Therefore, the number of quartic residues modulo $p$ is $(p - 1)/\gcd(4, p - 1)$. So, for $-1$ to be a quartic residue modulo $p$, $-1$ must be in the subgroup generated by the quartic residues. Therefore, the order of $-1$ in the multiplicative group must divide $(p - 1)/\gcd(4, p - 1)$. Wait, maybe I need to think differently.

Alternatively, since the multiplicative group mod $p$ is cyclic, then the equation $x^4 \equiv -1 \pmod{p}$ has solutions if and only if $-1$ is a quartic residue mod $p$. For that, since the multiplicative group is cyclic of order $p - 1$, then $-1$ is a quartic residue if and only if there exists an integer $k$ such that $(-1) \equiv g^{4k} \pmod{p}$, where $g$ is a primitive root mod $p$. Therefore, the order of $-1$ in the multiplicative group must divide $4k$. But since the order of $-1$ is 2 (since $(-1)^2 \equiv 1 \pmod{p}$), then 2 divides $4k$, which it does. But maybe more importantly, the exponent 4 must divide the order of the group? Wait, perhaps I need to think in terms of exponents.

Let me recall that in a cyclic group of order $m$, the number of solutions to $x^k = a$ is equal to the number of solutions if $a$ is in the subgroup generated by the $k$th powers. So, for the equation $x^4 = -1$ to have solutions, $-1$ must be in the subgroup generated by the 4th powers. Therefore, the order of $-1$ must divide the exponent of the group. Wait, maybe I need to think about the exponents.

Alternatively, since the multiplicative group $\pmod{p}$ is cyclic of order $p - 1$, then the equation $x^4 \equiv -1 \pmod{p}$ has solutions if and only if $-1$ is in the image of the map $x \mapsto x^4$. So, equivalently, the equation $x^4 \equiv -1 \pmod{p}$ has solutions if and only if there exists an integer $k$ such that $(-1) \equiv g^{4k} \pmod{p}$, where $g$ is a primitive root. Therefore, the order of $-1$ must divide $4k$. But since the order of $-1$ is 2, then 2 divides $4k$, which is always true. However, for there to be solutions, the exponent 4 must divide the order of the group? Wait, maybe not. Let me think again.

Suppose that the multiplicative group $\pmod{p}$ is cyclic of order $p - 1$. Let me denote the generator as $g$. Then, every element can be written as $g^m$ for some $m$. So, $-1$ is equal to $g^{(p-1)/2}$. Therefore, for $-1$ to be a quartic residue, there must exist some integer $k$ such that $g^{(p-1)/2} = (g^k)^4 = g^{4k}$. Therefore, we have that $g^{(p-1)/2} \equiv g^{4k} \pmod{p}$. Therefore, exponents must be congruent modulo $p - 1$. Therefore, $(p - 1)/2 \equiv 4k \pmod{p - 1}$. Therefore, $4k \equiv (p - 1)/2 \pmod{p - 1}$. Let me write that as $4k \equiv (p - 1)/2 \pmod{p - 1}$. Therefore, multiplying both sides by 2: $8k \equiv p - 1 \pmod{p - 1}$. Therefore, $8k \equiv 0 \pmod{p - 1}$. Therefore, $p - 1$ divides $8k$. But since $k$ is an integer, this is possible if and only if there exists some integer $k$ such that $8k \equiv 0 \pmod{p - 1}$. Which is equivalent to $p - 1$ divides $8k$. However, since $k$ can be any integer, this is always possible. Wait, that seems contradictory. Maybe I need to think differently.

Alternatively, since the equation $x^4 \equiv -1 \pmod{p}$ has solutions if and only if $-1$ is a quartic residue $\pmod{p}$. For that, since the multiplicative group is cyclic of order $p - 1$, then the quartic residues are the elements of the form $g^{4m}$ where $m$ is an integer. Therefore, $-1$ is a quartic residue if and only if there exists an integer $m$ such that $g^{(p-1)/2} \equiv g^{4m} \pmod{p}$.

Therefore, $(p-1)/2 \equiv 4m \pmod{p-1}$. Let me write this as $4m \equiv (p-1)/2 \pmod{p-1}$. Let me denote $d = \gcd(4, p-1)$. Then, for the equation $4m \equiv c \pmod{p-1}$ to have solutions, $c$ must be divisible by $d$. Here, $c$ is $(p-1)/2$. Therefore, we need that $(p-1)/2$ is divisible by $d = \gcd(4, p-1)$. Let me compute $d$.

So, $d = \gcd(4, p-1)$. Let me note that $p$ is a prime. So, $p-1$ is even, since $p$ is odd (except $p = 2$). Wait, $p = 2$ is a prime. Let me check $p = 2$ first. If $p = 2$, then $p^2 = 4$. So, $n^4 + 1 \equiv 0 \pmod 4$. Let me check $n = 1 : 1 + 1 = 2 \pmod 4$. Not 0. $n = 3 : 81 + 1 = 82 \equiv 2 \pmod 4$. Similarly, $n = 2 : 16 + 1 = 17 \equiv 1 \pmod 4$. So, no solution. Therefore, $p = 2$ is not possible. So $p$ must be an odd prime. Therefore, $p$ is odd, so $p-1$ is even. Therefore, $d = \gcd(4, p-1)$. Since $p-1$ is even, $d$ is either 2 or 4. If $p \equiv 1 \pmod 4$, then $p-1$ is divisible by 4, so $d = 4$. If $p \equiv 3 \pmod 4$, then $p-1$ is $2 \pmod 4$, so $d = 2$. Therefore, for the equation $4m \equiv (p-1)/2 \pmod{p-1}$ to have solutions, we need that $(p-1)/2$ is divisible by $d$. Let me check:

Case 1: $p \equiv 1 \pmod 4$. Then $p-1$ is divisible by 4, so $(p-1)/2$ is divisible by 2. Therefore, $d = 4$. Therefore, $(p-1)/2$ must be divisible by $d = 4$? Wait, no. Wait, the equation is $4m \equiv (p-1)/2 \pmod{p-1}$. Let me think again. Let me write the equation as:

We have $4m \equiv (p-1)/2 \pmod{p-1}$. Let me denote that equation. Let me note that $(p-1)/2$ is an integer because $p$ is odd. Let me denote $k = (p-1)/2$. Then, the equation becomes $4m \equiv k \pmod{p-1}$. But since $k = (p-1)/2$, then $k$ is equal to $(p-1)/2$. Therefore, the equation is $4m \equiv (p-1)/2 \pmod{p-1}$. Let me divide both sides by 2: $2m \equiv (p-1)/4 \pmod{(p-1)/2}$. Wait, maybe not. Let me think again. Let me write the equation as:

Let me consider the congruence $4m \equiv (p-1)/2 \pmod{p-1}$. Let me note that $p-1$ divides $4m - (p-1)/2$. Therefore, $4m - (p-1)/2 = (p-1) \cdot t$ for some integer $t$. Therefore, $4m = (p-1)/2 + (p-1) \cdot t$. Therefore, $4m = (p-1)(1/2 + t)$. Therefore, $m = (p-1)(1/2 + t)/4$. Therefore, $m$ must be an integer. Therefore, $(p-1)(1/2 + t)$ must be divisible by 4. Therefore, $(p-1)(1/2 + t) \equiv 0 \pmod 4$. Let me multiply both sides by 2: $(p-1)(1 + 2t) \equiv 0 \pmod 8$. Therefore, since $p-1$ is even, let me write $p-1 = 2k$. Then, the equation becomes $2k(1 + 2t) \equiv 0 \pmod 8$. Therefore, $k(1 + 2t) \equiv 0 \pmod 4$. Therefore, since $k = (p-1)/2$, which is an integer. Therefore, for this to hold, $k(1 + 2t)$ must be divisible by 4. Let me think about this.

But maybe this is getting too abstract. Let me try specific primes. Let me check small primes and see if there exists $n$ such that $n^4 + 1$ is divisible by $p^2$.

First, check $p = 2$: already checked, no solution.

$p = 3$: Check if there exists $n$ such that $n^4 + 1 \equiv 0 \pmod 9$. Let me compute $n^4 \pmod 9$ for $n = 1, 2, \ldots, 8$.

$n = 1 : 1^4 = 1 \pmod 9$. $1 + 1 = 2 \pmod 9 \neq 0$.

$n = 2 : 16 \pmod 9 = 7$. $7 + 1 = 8 \neq 0$.

$n = 3 : 81 \pmod 9 = 0$. $0 + 1 = 1 \neq 0$.

$n = 4 : 256 \pmod 9$. $256/9 = 28 \cdot 9 = 252$, $256 - 252 = 4$. So $4 + 1 = 5 \neq 0$.

$n = 5 : 5^4 = 625$. $625 \pmod 9$: $625/9 = 69 \cdot 9 = 621$, $625 - 621 = 4$. So $4 + 1 = 5 \neq 0$.

$n = 6 : 6^4 = 1296$. $1296 \pmod 9$ is $0$. $0 + 1 = 1 \neq 0$.

$n = 7 : 7^4 = 2401$. $2401 \pmod 9$: Let's compute $7^4 \pmod 9$. $7 \pmod 9$ is $7$. $7^2 = 49 \equiv 4 \pmod 9$. $7^4 = (7^2)^2 \equiv 4^2 = 16 \equiv 7 \pmod 9$. So $7 + 1 = 8 \neq 0$.

$n = 8 : 8^4 \pmod 9$. $8 \equiv -1 \pmod 9$. So $(-1)^4 = 1 \pmod 9$. $1 + 1 = 2 \neq 0$. So no solution for $p = 3$.

$p = 5$: Check if there exists $n$ such that $n^4 + 1 \equiv 0 \pmod{25}$. Let me check $n$ from 1 to 24.

But maybe there's a smarter way. Let me check if $-1$ is a quartic residue $\pmod 5$. Since 5 is prime, the multiplicative group $\pmod 5$ has order 4. So, the quartic residues are the elements raised to the 4th power. Since the group is cyclic of order 4, the 4th power map is the identity map. Therefore, every element raised to the 4th power is 1 $\pmod 5$. Therefore, the equation $x^4 \equiv -1 \pmod 5$ would be $x^4 \equiv 4 \pmod 5$. But since $x^4 \equiv 1 \pmod 5$ for all $x \not\equiv 0$, so $1 \equiv 4 \pmod 5$? No. Therefore, no solution. Therefore, $p = 5$ is not possible.

Wait, but maybe I made a mistake here. Let me check. For modulus 5, the multiplicative group has order 4. So, for any $x \not\equiv 0 \pmod 5$, $x^4 \equiv 1 \pmod 5$. Therefore, $x^4 \equiv -1 \pmod 5$ would imply $1 \equiv -1 \pmod 5$, which is $1 \equiv 4 \pmod 5$, which is false. Therefore, no solution. Therefore, $p = 5$ is not possible.

$p = 7$: Check if there exists $n$ such that $n^4 + 1 \equiv 0 \pmod{49}$. Let me check if $-1$ is a quartic residue $\pmod 7$. First, check $\pmod 7$. The multiplicative group $\pmod 7$ has order 6. So, quartic residues are elements raised to the 4th power. Let me compute the quartic residues $\pmod 7$. Let me compute $1^4 = 1$, $2^4 = 16 \equiv 2$, $3^4 = 81 \equiv 4$. $4^4 = 256 \equiv 4 \pmod 7$, since $4^2 = 16 \equiv 2$, so $4^4 \equiv 2^2 = 4$. Similarly, $5^4$: $5^2 = 25 \equiv 4$, $5^4 \equiv 4^2 = 16 \equiv 2 \pmod 7$. $6^4$: $6^2 = 36 \equiv 1$, $6^4 \equiv 1^2 = 1 \pmod 7$. Therefore, quartic residues $\pmod 7$ are $\{1, 2, 4\}$. Therefore, $-1 \pmod 7$ is 6. Is 6 a quartic residue? The quartic residues are $\{1, 2, 4\}$. So no. Therefore, no solution $\pmod 7$. Therefore, $p = 7$ is not possible.

$p = 11$: Let me check if $-1$ is a quartic residue $\pmod{11}$. The multiplicative group $\pmod{11}$ has order 10. So, quartic residues are elements raised to the 4th power. Let me note that the multiplicative group is cyclic of order 10. So, the quartic residues are the elements of the form $g^{4k}$ where $g$ is a generator. The number of quartic residues is $10/\gcd(4, 10) = 10/2 = 5$. Therefore, there are 5 quartic residues $\pmod{11}$. Let me find them. Let me take a generator, say 2 $\pmod{11}$. The powers of 2 are: $2^1 = 2, 2^2 = 4, 2^3 = 8, 2^4 = 16 \equiv 5, 2^5 = 10, 2^6 \equiv 9, 2^7 \equiv 7, 2^8 \equiv 3, 2^9 \equiv 6, 2^{10} \equiv 1 \pmod{11}$. The quartic residues are $\{2^{4k} \mid k \in \mathbb{Z}\}$. For $k = 0 : 2^0 = 1$. For $k = 1 : 2^4 = 5$. For $k = 2 : 2^8 = 3$. For $k = 3 : 2^{12} \equiv 2^2 = 4$. For $k = 4 : 2^{16} \equiv 2^6 = 9$. The set of quartic residues $\pmod{11}$ is $\{1, 3, 4, 5, 9\}$. We need to check if $-1 \equiv 10 \pmod{11}$ is in this set. It is not. Therefore, $p = 11$ is not possible.

$p = 13$: Let me check if $-1$ is a quartic residue $\pmod{13}$. The multiplicative group $\pmod{13}$ has order 12. So, quartic residues are elements raised to the 4th power. Let me check if $-1$ is a quartic residue. Since the multiplicative group is cyclic of order 12, the quartic residues are the elements of the form $g^{4k}$ where $g$ is a generator. The number of quartic residues is $12/\gcd(4, 12) = 12/4 = 3$. Therefore, there are 3 quartic residues. Let me check if $-1$ is among them. Let me take a generator, say 2 $\pmod{13}$. Let me compute the powers of 2:
$2^1 = 2, 2^2 = 4, 2^3 = 8, 2^4 = 16 \equiv 3, 2^5 = 6, 2^6 = 12, 2^7 \equiv 11, 2^8 \equiv 9, 2^9 \equiv 5, 2^{10} \equiv 10, 2^{11} \equiv 7, 2^{12} \equiv 1 \pmod{13}$.
So, the quartic residues would be the elements $2^{4k} \pmod{13}$. Let me compute: For $k = 0 : 2^0 = 1$. For $k = 1 : 2^4 = 3$. For $k = 2 : 2^8 = 9$. For $k = 3 : 2^{12} \equiv 1$. So the quartic residues are $\{1, 3, 9\}$. We need to check if $-1 \pmod{13}$ which is 12, is in the set. It is not. Therefore, $p = 13$ is not possible.

$p = 17$: Let me check. The multiplicative group $\pmod{17}$ has order 16. So, quartic residues are elements raised to the 4th power. Let me check if $-1$ is a quartic residue $\pmod{17}$. Since the multiplicative group is cyclic of order 16, the quartic residues are the elements of the form $g^{4k}$. The number of quartic residues is $16/\gcd(4, 16) = 16/4 = 4$. Therefore, there are 4 quartic residues. Let me check if $-1$ is among them. Let me take a generator, say 3 $\pmod{17}$. Let me compute the powers of 3:
$3^1 = 3, 3^2 = 9, 3^3 = 27 \equiv 10, 3^4 = 81 \equiv 13, 3^5 = 39 \equiv 5, 3^6 = 15, 3^7 = 45 \equiv 12, 3^8 = 36 \equiv 2$, $3^9 = 6, 3^{10} = 18 \equiv 1$, etc. Wait, maybe I need to check the order of 3 $\pmod{17}$. Since $3^{16} \equiv 1 \pmod{17}$. Let me check $3^8 \pmod{17}$: $3^8 = (3^4)^2 = 13^2 = 169 \equiv 169 - 17 \cdot 10 = 169 - 170 = -1 \pmod{17}$. Therefore, $3^8 \equiv -1 \pmod{17}$. Therefore, $3^8 \equiv -1 \pmod{17}$. Therefore, $-1$ is a quartic residue mod 17? Wait, because $3^8 \equiv -1 \pmod{17}$. Therefore, if we take $x = 3^2$, then $x^4 = (3^2)^4 = 3^8 \equiv -1 \pmod{17}$. Therefore, $x = 3^2 = 9$. Therefore, $9^4 \equiv -1 \pmod{17}$. Therefore, yes, $-1$ is a quartic residue mod 17. Therefore, $p = 17$ is possible. Therefore, $p = 17$ is a candidate.

**[The logic goes wrong here: The model correctly finds p=17 is the smallest prime candidate by systematically checking. However, instead of proceeding to the second part of the problem (finding m), it enters a repetitive loop, endlessly questioning its own correct conclusion.]**

Let me check if there exists $n$ such that $n^4 + 1$ is divisible by $17^2$. Wait, but first, let me check if $p = 17$ is the minimal prime for which this is possible.

Wait, before that, let me check primes between 13 and 17. The primes are 17, but before that, $p = 17$ is the next. Wait, $p = 17$ is the first prime where $-1$ is a quartic residue? Let me check $p = 17$. So, for $p = 17$, since $-1$ is a quartic residue mod 17, then there exists $n$ such that $n^4 \equiv -1 \pmod{17}$. Therefore, $p = 17$ is a candidate. Let me check if there is a smaller prime. Wait, we checked $p = 2, 3, 5, 7, 11, 13$. None worked. So $p = 17$ is the first candidate. Therefore, maybe $p = 17$ is the answer? Wait, but the problem says "the least prime number for which there exists a positive integer $n$ such that $n^4 + 1$ is divisible by $p^2$".

So, if $p = 17$ is the first such prime, then the answer is $p = 17$. But then, the problem says "Find the least positive integer $m$ such that $m^4 + 1$ is divisible by $p^2$". So, once we find $p = 17$, then find the minimal $m$.

But before that, let me check if there is a prime smaller than 17 where $-1$ is a quartic residue. Wait, maybe I made a mistake with $p = 17$. Let me check $p = 17$ again. For $p = 17$, since $-1$ is a quartic residue mod 17, then there exists $n$ such that $n^4 \equiv -1 \pmod{17}$. Therefore, $p = 17$ is possible. Let me check if there is a smaller prime. Let me check $p = 17$ is the first one. Let me check $p = 17$, then check $p = 7$, $p = 5$, etc. Wait, maybe I need to check $p = 17$ is indeed the first one. Let me check $p = 17$. Let me check if there is a prime between 13 and 17. The primes are 17, so no. So $p = 17$ is the first candidate. Therefore, maybe $p = 17$ is the answer. But let me check if there is a prime $p$ where $-1$ is a quartic residue mod $p$, but $p$ is smaller than 17. Wait, maybe I made a mistake with $p = 17$. Let me check $p = 17$ again. For $p = 17$, since $-1$ is a quartic residue mod 17, then there exists $n$ such that $n^4 \equiv -1 \pmod{17}$. Therefore, $p = 17$ is possible. Let me check if there is a prime $p$ where $-1$ is a quartic residue mod $p$. For example, $p = 17$, then $p = \ldots$ Let me check $p = 17$. Let me check if there is a smaller prime. Let me check $p = 17$, then $p = 7$, $p = 5$, etc. Wait, maybe I need to check $p = 17$ is indeed the first one. Let me check $p = 17$. Let me check if there is a prime $p$ where $-1$ is a quartic residue mod $p$. For example, $p = 17$, then $p = \ldots$ Let me think. For $p = 17$, since the multiplicative group is cyclic of order 16, and $-1$ is an element of order 2. So, for $-1$ to be a quartic residue, there must exist some element $x$ such that $x^4 \equiv -1 \pmod{p}$. Which is equivalent to $x^8 \equiv 1 \pmod{p}$. Therefore, the order of $x$ divides 8. But since the multiplicative group has order 16, then the order of $x$ divides 16. Therefore, $x^8 \equiv 1 \pmod{p}$. Therefore, the order of $x$ divides 8. Therefore, $x$ is an element of order dividing 8. Therefore, since $-1$ is in the group, and $-1$ has order 2. Therefore, if there exists an element of order 8,

But before that, let me check if there is a prime smaller than 17 where this is possible. Wait, maybe I made a mistake with $p = 17$. Let me check $p = 17$. Let me check if there is a prime $p$ where $-1$ is a quartic residue $\pmod{p}$. For example, $p = 17$ is the first such prime? Let me check $p = 17$. Let me check if there is a smaller prime. Let me check $p = 17$. Wait, maybe I need to check $p = 17$ is indeed the first one. Let me check $p = 17$. Let me check $p = 17$. ... Let me check $p = 17$. Let me

To solve the problem, we are asked to find the **least prime number** $p$ such that there exists a positive integer $n$ for which $n^4 + 1$ is divisible by $p^2$, and then determine the **least positive integer** $m$ such that $m^4 + 1$ is divisible by $p^2$.

Step 1: Understanding the Problem

We are looking for the **smallest prime** $p$ such that:

$$n^4 + 1 \equiv 0 \pmod{p^2}$$

This is equivalent to:

$$n^4 \equiv -1 \pmod{p^2}$$

This means that $-1$ must be a **quartic residue modulo** $p^2$. That is, there must exist an integer $n$ such that $n^4 \equiv -1 \pmod{p^2}$.

Step 2: Strategy

We will check small primes $p$ in increasing order to see if $-1$ is a quartic residue modulo $p^2$. That is, we want to find the **smallest prime** $p$ such that the congruence $n^4 \equiv -1 \pmod{p^2}$ has a solution.

We can use the following approach:

- For each prime $p$, check whether $-1$ is a quartic residue modulo $p$. - If it is, then check whether $n^4 \equiv -1 \pmod{p^2}$ has a solution. - The first such $p$ is our answer.

Step 3: Checking Small Primes

Let's test small primes:

- $p = 2$: $n^4 + 1 \equiv 0 \pmod{4} \Rightarrow n^4 \equiv -1 \pmod{4} \Rightarrow n^4 \equiv 3 \pmod{4}$. But $n^4 \pmod{4}$ is always 0 or 1. So no solution.

- $p = 3$: $n^4 \equiv -1 \pmod{9} \Rightarrow n^4 \equiv 8 \pmod{9}$. Try $n = 1, 2, \ldots, 8$: none satisfy $n^4 \equiv 8 \pmod{9}$.

- $p = 5$: $n^4 \equiv -1 \pmod{25} \Rightarrow n^4 \equiv 24 \pmod{25}$. Try $n = 1, 2, \ldots, 24$: no solution.

- $p = 7$: $n^4 \equiv -1 \pmod{49} \Rightarrow n^4 \equiv 48 \pmod{49}$. Try $n = 1, 2, \ldots, 48$: no solution.
- $p = 11$: $n^4 \equiv -1 \pmod{121} \Rightarrow n^4 \equiv 120 \pmod{121}$. Try $n = 1, 2, \ldots, 120$: no solution.
- $p = 13$: $n^4 \equiv -1 \pmod{169} \Rightarrow n^4 \equiv 168 \pmod{169}$. Try $n = 1, 2, \ldots, 168$: no solution.
- $p = 17$: Try $n = 1, 2, \ldots, 16$: $n^4 \equiv -1 \pmod{17}$ has a solution. For example, $n = 9$: $9^4 = 6561 \equiv -1 \pmod{17}$.

So, $p = 17$ is the **smallest prime** for which $n^4 + 1 \equiv 0 \pmod{p^2}$ has a solution.

Step 4: Finding the Smallest $m$ such that $m^4 + 1 \equiv 0 \pmod{17^2}$

We now need to find the **smallest positive integer** $m$ such that:

$$m^4 + 1 \equiv 0 \pmod{17^2}$$

That is:

$$m^4 \equiv -1 \pmod{289}$$

We already know that $n = 9$ satisfies $n^4 \equiv -1 \pmod{17}$, so we can try to lift this solution to modulo $17^2$ using **Hensel's Lemma**.

Let's check whether $n = 9$ satisfies $n^4 \equiv -1 \pmod{289}$:

$$9^4 = 6561 \Rightarrow 6561 \mod 289 = 6561 - 289 \times 22 = 6561 - 6358 = 221$$

So $9^4 \equiv 221 \pmod{289}$. But $-1 \equiv 288 \pmod{289}$, so $221 \neq 288$. Therefore, $n = 9$ is **not** a solution modulo 289.

We now try to find the **smallest** $m$ such that $m^4 \equiv -1 \pmod{289}$.

Try $m = 17k + 9$ for small $k$, and compute $m^4 \mod 289$.

After testing, we find that:

$$m = 17 \Rightarrow m^4 = 83521 \Rightarrow 83521 \mod 289 = 169$$

Try $m = 17 + 9 = 26$:

$$26^4 = 456976 \Rightarrow 456976 \mod 289 = 169$$

Try $m = 17 + 16 = 33$:

$$33^4 = 1185921 \Rightarrow 1185921 \mod 289 = 169$$

Try $m = 17 + 1 = 18$:

$$18^4 = 104976 \Rightarrow 104976 \mod 289 = 169$$

Try $m = 17 + 2 = 19$:

$$19^4 = 130321 \Rightarrow 130321 \mod 289 = 169$$

...

Try $m = 17 + 492 = 512$:

$$512^4 = 1600000000 \Rightarrow 1600000000 \mod 289 = 169$$

Try $m = 17 + 493 = 513$:

$$513^4$$

**[Model continues to iterate till here, testing values for $m$ from 1 up to 493 until it exceeds the maximum token limit without finding a solution.]**

