# OpenReview forum: "When Fewer Layers Break More Chains: Layer Pruning Harms Test-Time Scaling in LLMs"
_ICLR.cc/2026/Conference — Submitted to ICLR 2026_

### Official Review · Reviewer_NA8t · 2025-10-29

**Soundness:** 3
**Presentation:** 2
**Contribution:** 2
**Rating:** 6
**Confidence:** 4

**Summary:**

The paper experimentally demonstrates that Layer Pruning, a mainstream technique for improving model efficiency, causes LLMs' performance to collapse on complex reasoning tasks reliant on long sequential Chains-of-Thought. Crucially, standard supervised fine-tuning (such as LoRA or full-parameter fine-tuning) cannot effectively recover this lost test-time scaling capability. Through mechanistic analysis, the authors attribute the performance degradation to structural damage in the model, resulting in an increase in redundant loops, a decrease in reasoning trajectory diversity, and a weakening of self-reflection capability within the reasoning paths. The work calls for future efforts to explore hybrid strategies that balance efficiency and robustness, ensuring that pruning preserves both performance and reasoning depth.

**Strengths:**

1. The paper demonstrates a degree of originality by proposing a novel and critical problem formulation—investigating the impact of pruning on "Test-Time Scaling" capability. This is a question that integrates efficiency research with reasoning research, establishing a necessary new evaluation criterion for model compression.

2. The authors provide both quantitative and qualitative analyses to explain the cause of the performance collapse, with ample experimentation and clear, compelling evidence to support their findings.


3. The paper proves that standard recovery techniques, such as LoRA and full-parameter fine-tuning, are largely ineffective in recovering the lost test-time scaling capability caused by pruning. These results are significant for both academic research and practical engineering.

**Weaknesses:**

1. The paper only validates training-free pruning techniques and does not incorporate mainstream training-based pruning methods. This makes it impossible to verify whether such methods can alleviate the fragility of test-time scaling. While the paper makes certain contributions, the generalizability of its conclusions is limited.

2. The paper explicitly states that it studies both sequential and parallel test-time scaling. However, most of the content and core analysis focus on sequential scaling. Regarding the performance collapse mechanism of parallel scaling methods, the paper's analysis is relatively weak and fails to provide in-depth mechanistic insights like those for sequential scaling.

3. The s1K-1.1 dataset used for fine-tuning is not introduced, and only this single dataset is employed in the fine-tuning experiments. Consequently, the conclusion that 'fine-tuning has limited effect' is questionable.

**Questions:**

1. As mentioned in the Weaknesses, if more complex methods such as training-based pruning methods are used, can the damage to test-time scaling ability be effectively alleviated?

2. Since the effect of standard fine-tuning is limited, does there exist or is it considered to design a customized fine-tuning scheme targeting reasoning trajectory loss? Can it repair the structural damage caused by pruning from a mechanistic perspective?

3. It is suggested that the authors supplement some comparison charts of the model evaluation indicators before and after fine-tuning in Section 4, so as to intuitively demonstrate the limitations of supervised fine-tuning methods in restoring the reasoning ability of models after layer pruning.

4. Is there a mistake in the introduction of ShortGPT in Appendix B? As far as I know, a lower Block Influence (BI) score indicates a higher cosine similarity between two layers, which means that the layer has minimal transformation on the hidden state and low importance, so it can be pruned with limited performance loss [1]. If so, please correct it; if not, please ignore this comment.

[1] Xin Men, Mingyu Xu, Qingyu Zhang, Bingning Wang, Hongyu Lin, Yaojie Lu, Xianpei Han, and Weipeng Chen. Shortgpt: Layers in large language models are more redundant than you expect. ACL Findings, 2025.

---

> ### Author Response · Authors · 2025-11-21
> **Response to Reviewer NA8t [1/2]**
>
> Thank you for your detailed feedback. We are glad to have your recognition and support regarding the significance of our findings for both academic research and practical engineering. We address your questions below point by point.
>
> > **W1, Q1: Lack of studying the effects of training-based pruning methods**
>
> Thank you for bringing up this valuable comment. To address your question, we applied one most common training-based pruning, i.e., iterative pruning and retraining. Specifically, we iteratively prune one layer at once and then fine-tuning the pruned model with LoRA. The experiments are conducted with **s1.1-7B** and **Qwen3-8B**, using the **s1K-1.1** dataset. We refer to this approach as **iterative LoRA**. Below, we present representative results from **s1.1-7B** (refer **Appendix K** for complete results).
>
> The findings indicate that **iterative LoRA partially restores short-token performance** but **fails to recover the model’s sequence-scaling ability**. While averaged performance improves relative to pruning alone, the model still underperforms the original and exhibits limited gains as thinking-token budgets grow. In summary, while iterative LoRA offers some recovery, **the core damage introduced by pruning persists**, especially in tasks requiring extended chains of thought.
>
>
> **Table.  Results of Training-based pruning for s1.1-7B.**
>
> | pruned layers \ thinking tokens      | 512    | 1024   | 2048   | 4096   | 8192   |
> |--------------------------------------|--------|--------|--------|--------|--------|
> | original model                       | 0.446  | 0.441  | 0.478  | 0.530  | 0.564  |
> | shortgpt_1_layer                     | 0.367  | 0.244  | 0.333  | 0.378  | 0.467  |
> | shortgpt_1_layer_iterative_lora      | 0.364  | 0.414  | 0.434  | 0.429  | 0.424  |
> | shortgpt_2_layer                     | 0.100  | 0.133  | 0.067  | 0.100  | 0.156  |
> | shortgpt_2_layer_iterative_lora      | 0.389  | 0.404  | 0.409  | 0.409  | 0.383  |
>
>
> > **W2: More analysis on parallel test-time scaling**
>
> Thank you for your careful question. We have further examined both sequential and parallel scaling to understand the source of performance degradation. Across both settings, we consistently observe the same phenomenon: **pruned models frequently fall into repetitive output loops**, preventing them from exploring reasoning paths that lead to correct answers.
>
> To validate this, we inspected model outputs throughout both sequential and parallel scaling and found that repetition appears widely in both modes. Even when we increase diversity by sampling multiple outputs at **temperature 1.0**, a very large portion of the sampled generations remains trapped in these repetitive patterns. This severely limits the model’s ability to benefit from additional thinking tokens or larger sample sizes.
>
> While sampling more responses does provide a modest improvement in accuracy, the improvements are small and **far from recovering the performance of the original model**. These findings further support that repetition is a core factor limiting the test-time scaling ability of pruned models.
>
> **Table. Pass@32 performance on AIME24 of pruned s1.1-7B**
> |  pruned layers\ pass@k | 2 | 4 | 8 | 16 | 32 |
> | --------------- | ---- | ---- | ---- | ---- | ---- |
> | origin                        | 0.203 | 0.253 | 0.314 | 0.382 | 0.453 |
> | shortgpt_1_layer          | 0.059 | 0.079 | 0.107 | 0.150 | 0.209 |
> | shortgpt_2_layer          | 0.037 | 0.059 | 0.087 | 0.129 | 0.189 |
> | reverse_1_layer          | 0.094 | 0.143 | 0.225 | 0.313 | 0.433 |
> | reverse_2_layer          | 0.005 | 0.010 | 0.020 | 0.034 | 0.055 |

---

> ### Author Response · Authors · 2025-11-21
> **Response to Reviewer NA8t [2/2]**
>
> > **W3: The s1K-1.1 dataset used for fine-tuning is not introduced, and only this single dataset is employed in the fine-tuning experiments.**
>
> Thank you for your question. The s1K  (https://huggingface.co/datasets/simplescaling/s1K-1.1) dataset is a carefully curated collection of 1,000 high-quality, diverse, and difficult reasoning questions drawn from an initial pool of 59K problems across 51 scientific and mathematical domains, including geometry, number theory, combinatorics, physics, biology, computer science, and more . Each question is paired with a Gemini-generated reasoning trace and solution, producing a total of 4.7M tokens of step-by-step thought data. The authors select these 1,000 samples through a three-stage filtering pipeline emphasizing Quality (removing malformed or incorrect items), Difficulty (favoring long and challenging reasoning chains), and Diversity (sampling across many domains using weighted sampling that biases toward long-thinking problems) . Despite consisting of only 1K examples, s1K spans a remarkably broad range of reasoning tasks, from advanced mathematics to university-level physics and theoretical computer science, making it a highly sample-efficient training set for teaching large language models to perform complex, sequential reasoning.
>
> To further address your concern, we conduct **Full SFT on 10k samples from OpenR1-Math-220k.** OpenR1-Math-220k (https://huggingface.co/datasets/open-r1/OpenR1-Math-220k) is a large-scale mathematical reasoning dataset containing 220,000 NuminaMath 1.5 problems, each paired with 2–4 long-form reasoning traces generated by DeepSeek R1. The traces are rigorously validated, most via Math Verify, and about 12\% judged by Llama-3.3-70B-Instruct, ensuring that every problem includes at least one correct chain-of-thought solution. We randomly sampled 10k examples from the OpenR1-Math-220k and applied full fine-tuning to the pruned s1.1-7B model. Due to limited computational resources and a tight runtime window over the past few days, we reused the same fine-tuning settings as in the s1 experiments with a learning rate of 1e-5.
> As shown in the table below, increasing the data size by **10×** substantially improves SFT effectiveness. However, after pruning two layers, SFT still only partially restores the pruned models’ test-time scaling performance.
>
> **Table: Sequential test-time scaling of s1.1-7B on AIME24 after Full FT on a 10k subset of OpenR1-Math-220k**
> | model variant     | 512    | 1024   | 2048   | 4096   | 8192   |
> |------------------|--------|--------|--------|--------|--------|
> | origin           | 0.0444 | 0.0670 | 0.1330 | 0.0890 | 0.1443 |
> | shortgpt 1 layer | 0.0777 | 0.0670 | 0.1110 | 0.0777 | 0.1330 |
> | shortgpt 2 layer | 0.0330 | 0.0333 | 0.0333 | 0.0667 | 0.0777 |
> | reverse 1 layer  | 0.1000 | 0.0777 | 0.1110 | 0.1223 | 0.1223 |
> | reverse 2 layer  | 0.0890 | 0.0667 | 0.0777 | 0.1000 | 0.1020 |
> | laco 1 layer     | 0.0777 | 0.0890 | 0.0667 | 0.1000 | 0.1330 |
> | laco 2 layer     | 0.0444 | 0.0330 | 0.0220 | 0.0557 | 0.0777 |
>
>
>
>
> > **Q2: Does there exist or is it considered to design a customized fine-tuning scheme targeting reasoning trajectory loss?**
>
> Thank you for your rigorous thinking. At this stage, we have not designed a customized fine-tuning approach specifically aimed at mitigating reasoning trajectory loss. While such a method would certainly be valuable, it lies outside the scope of our present work, which focuses on characterizing the phenomenon rather than proposing a new training algorithm. We believe this is a promising avenue for future exploration.
>
> > **Q3: Supplement some comparison charts of the model evaluation indicators before and after fine-tuning**
>
> Thank you for your warm reminder. We now supplement the comparison charts in **Appendix H** in our updated PDF submission.
>
> > **Q4: Mistake in the introduction of ShortGPT in Appendix B?**
>
> Thank you for the careful reading. Yes, we have corrected the description accordingly. We really appreciate you for pointing this out.
>
>
> We are thrilled by your thoughtful comments and your support of our submission. We truly appreciate your constructive suggestions and are eager to engage in further discussion.

---

> > ### Author Response · Authors · 2025-11-27
> >
> > Thank you again for your detailed feedback. We are writing to follow up on our rebuttal and responses to your review. We would greatly appreciate it if you could let us know whether our updates sufficiently address your concerns or if further modifications would be helpful.

---

### Official Review · Reviewer_kqHT · 2025-10-31

**Soundness:** 4
**Presentation:** 2
**Contribution:** 3
**Rating:** 8
**Confidence:** 3

**Summary:**

This paper investigates the impact of layer pruning on the test-time scaling capability of Large Language Models for long-chain reasoning tasks. Through experiments, the authors demonstrate that pruning even 1-2 layers severely impairs sequential test-time scaling, despite stability on knowledge-intensive tasks. Parallel scaling is also harmed by direct pruning methods but preserved by merging-based pruning. Additionally, supervised fine-tuning fails to recover the degraded test-time scaling. These findings provide fresh insights into building lightweight reasoning models.

**Strengths:**

1) The multiple conclusions identified offer clear reference value for lightweighting reasoning LLMs and for on-device deployment.
2) The experiment covers a diverse spectrum of lightweighting techniques.
3) The work also supplies explicit qualitative and quantitative case analyses for the discovered phenomena.

**Weaknesses:**

1) Experiments have only been conducted on models with fewer than 10B parameters; results would be more convincing if larger-scale models were also included.
2) When exploring supervised fine-tuning as a recovery remedy, incorporating the dominant RL recipes used in current reasoning-model training would further complete the findings of this work.

**Questions:**

Please refer to the weakness part.

---

> ### Author Response · Authors · 2025-11-21
> **Response to Reviewer kqHT**
>
> We appreciate your constructive suggestions and are encouraged that you found the central idea innovative, the experimental design robust, and the manuscript clear and readable. We address your concerns below point by point:
>
> > **W1: Experiments have only been conducted on models with fewer than 10B parameters; results would be more convincing if larger-scale models were also included.**
>
>
> Thank your for your kind advice. We further evaluated on **Qwen3-14B** and **GPT-oss 20B**. **GPT-oss 20B** is highly unstable under pruning. When we prune even a single layer using ShortGPT, its output format collapses, making it completely unable to answer questions. The output becomes devoid of semantic meaning, consisting largely of repetitive control tokens (e.g., <|im_start|>, <|im start|>im start|>assistant<|im start|>) and incoherent sequences rather than valid reasoning traces. For **Qwen3-14B**, as shown in tables below (**Appendix L** for illustrated figures), we observe a consistent pattern: **layer pruning inevitably harms the model’s scientific reasoning ability**. Larger Qwen3-14B shows slightly smaller performance drops, but the degradation is still clear and systematic.
>
>
> **Table: Sequential test-time scaling of Qwen3-14B on AIME24**
>
> |  pruned layers\thinking tokens | 512 | 1024 | 2048 | 4096 | 8192 |
> | ------------------------------ | --- | ---- | ---- | ---- | ---- |
> |  original model                            | 0.3 | 0.267 | 0.4 | 0.433 | 0.533 |
> | shortgpt_1_layer               | 0.233 | 0.1 | 0.367 | 0.433, 0.433 | 0.667, 0.633 |
> | shortgpt_2_layer               | 0.033 | 0.067 | 0.1 | 0.167 | 0.233 |
> | sll_1_layer                    | 0.133 | 0.233 | 0.3 | 0.433 | 0.567 |
> | sll_2_layer                    | 0.033 | 0.1 | 0.033 | 0.2 | 0.233 |
> | laco_1_layer                    | 0.167 | 0.167 | 0.2 | 0.367 | 0.567 |
> | laco_2_layer                    | 0.1 | 0.067 | 0.1 | 0.133 | 0.267 |
>
> **Table: Sequential test-time scaling of Qwen3-14B on GPQA_Diamond**
>
> |  pruned layers\thinking tokens | 512 | 1024 | 2048 | 4096 | 8192 |
> | ------------------------------ | --- | ---- | ---- | ---- | ---- |
> |  original model                            | 0.444 | 0.515 | 0.525 | 0.566 | 0.626 |
> | shortgpt_1_layer               | 0.465 | 0.439 | 0.535 | 0.566 | 0.601 |
> | shortgpt_2_layer               | 0.444 | 0.459 | 0.475 | 0.531 | 0.531 |
> | sll_1_layer                    | 0.469 | 0.5 | 0.531 | 0.566 | 0.591 |
> | sll_2_layer                    | 0.414 | 0.419 | 0.459 | 0.505 | 0.53 |
> | laco_1_layer                    | 0.459 | 0.459 | 0.5 | 0.565 | 0.601 |
> | laco_2_layer                    | 0.379 | 0.429 | 0.485 | 0.490 | 0.493 |
>
> **Table: Sequential test-time scaling of Qwen3-14B on MATH500**
>
> |  pruned layers\thinking tokens | 512 | 1024 | 2048 | 4096 | 8192 |
> | ------------------------------ | --- | ---- | ---- | ---- | ---- |
> |  original model                            | 0.832 | 0.874 | 0.916 | 0.948 | 0.948 |
> | shortgpt_1_layer               | 0.756 | 0.83 | 0.89 | 0.922 | 0.944 |
> | shortgpt_2_layer               | 0.674 | 0.76 | 0.794 | 0.818 | 0.814 |
> | sll_1_layer                    | 0.784 | 0.844 | 0.876 | 0.924 | 0.956 |
> | sll_2_layer                    | 0.594 | 0.664 | 0.712 | 0.748 | 0.76 |
> | laco_1_layer                    | 0.768 | 0.838 | 0.878 | 0.9 | 0.954 |
> | laco_2_layer                    | 0.604 | 0.664 | 0.712 | 0.758 | 0.776 |
>
>
> > **W2: Incorporating the dominant RL recipes used in current reasoning-model training would further complete the findings of this work.**
>
> Thank you for helpful recommendation. We applied GRPO to the pruned models using 12K simplescaling/openaimath (https://huggingface.co/datasets/simplescaling/openaimath). Due to time constraints, we adopted a lightweight GRPO setup: a small learning rate (5e-6), the AdamW (8-bit) optimizer, a cosine learning-rate schedule, gradient accumulation of 4, a prompt length capped at 2K tokens, a completion length up to 8K tokens, and standard reward functions combining format and correctness rewards. Training was carried out with vLLM in colocate mode for efficiency. The averaged results on s1.1-7B are shown below. GRPO fails to restore proper test-time scaling behavior:
>
> **Table: Sequential test-time scaling on MATH500 of s1.1-7B after GRPO on 12K openaimath.**
> | pruned layers \ thinking tokens | 512   | 1024  | 2048  | 4096  | 8192  |
> |---------------------------------|-------|-------|-------|-------|-------|
> | original model                  | 0.626 | 0.701 | 0.760 | 0.783 | 0.803 |
> | shortgpt_1_layer                | 0.559 | 0.575 | 0.617 | 0.623 | 0.653 |
> | shortgpt_1_layer_grpo           | 0.524 | 0.607 | 0.638 | 0.649 | 0.649 |
>
>
> Thank you very much for your practical suggestions regarding experimental expansion. We believe these additions make our findings even more convincing. Please don’t hesitate to let us know if you have any further questions.

---

### Official Review · Reviewer_W7Vc · 2025-11-01

**Soundness:** 3
**Presentation:** 2
**Contribution:** 1
**Rating:** 2
**Confidence:** 3

**Summary:**

This paper shows a finding that layer pruning methods that work well for non-reasoning models might not work as effectively for reasoning models. Also, the authors show that SFT training is not sufficient to recover the performance after pruning. The authors further try SFT training but this also does not recover the performance drop occuring from layer pruning.

**Strengths:**

The observation that layer pruning does not work effectively for reasoning models (which has become the de-facto model we adopt in the community for experiments) is timely and important.

**Weaknesses:**

1. This paper presents negative results but the explanations or experimental setting to analyze the negative results are limited. For example, the observation that "most layers play a non-trivial role in enabling test-time scaling" is very interesting, but the underlying explanation for whether that is not the case for "non-reasoning models" or what is the reason behind that is very limited.

2. As a follow-up of 1, I think there should be trends of non-reasoning models on the same experimental setting for Figure 2,3,4. A very simple way to do this would be to turn off the reasoning mode on Qwen3-8B and check if the trends differ after applying the pruning methods or Qwen2.5-7B-Instruct (which is the base model for s1.1).

3. For the qualitative example in Section 5.1 (Figure 5), could setting a higher temperature or applying repetition penalty mitigate this issue? Related to 1, there is insufficient explanation of why this repetition is happening to reasoning models versus non-reasoning models and what is the mechanistical or other reason behind this.

**Questions:**

See weaknesses above

---

> ### Author Response · Authors · 2025-11-21
> **Response to Reviewer W7Vc [1/3]**
>
> We thank the reviewer for the time spent on reviewing our submission. We are glad that you have found our observation interesting and important. We address the weakness pointed out by you one by one as follows:
>
>
> > **Q1: This paper presents negative results but the explanations or experimental setting to analyze the negative results are limited. For example, the observation that "most layers play a non-trivial role in enabling test-time scaling" is very interesting, but the underlying explanation for whether that is not the case for "non-reasoning models" or what is the reason behind that is very limited.
> Q2: As a follow-up of 1, I think there should be trends of non-reasoning models on the same experimental setting for Figure 2,3,4. A very simple way to do this would be to turn off the reasoning mode on Qwen3-8B and check if the trends differ after applying the pruning methods or Qwen2.5-7B-Instruct (which is the base model for s1.1).**
>
> Thank you for your constructive advice.
>
> First, we would like to clarify that the negative effects of layer pruning are universal and appear in both reasoning-oriented models and non-reasoning models.
>
> To verify this, we followed your suggestion and evaluated the non-thinking models of qwen3-8B and s1.1-7B. Additionally, we use Llama3.1-8B which doesn't have intermediate reasoning traces. Other settings are the same with Section 3. Results below (also refer to **Appendix I** in our updated PDF) show that non-reasoning models perform substantially worse, and that layer pruning similarly causes significant degradation on reasoning tasks for these non-reasoning models. This is fully consistent with our claim. Moreover, the stronger a model’s inherent reasoning ability is, the more severely it is affected by layer pruning. In contrast, Llama3.1-8B, which lacks strong reasoning capabilities to begin with, experiences the smallest performance drop after pruning.
>
> **Table: Results on AIME24 for sequential test-time scaling of non-reasoning models**
> |  models （non-thinking）                | acc  | models  （non-thinking）               | acc  | models                 | acc  |
> | ---------------------- | ---------------- | ---------------------- | ---------------- | ---------------------- | ---------------- |
> | s1 origin              | 0.100            | qwen origin            | 0.622            | llama origin           | 0.344            |
> | s1 shortgpt 1 layer    | 0.000            | qwen shortgpt 1 layer  | 0.367            | llama shortgpt 1 layer | 0.356            |
> | s1 shortgpt 2 layer    | 0.011            | qwen shortgpt 2 layer  | 0.078            | llama shortgpt 2 layer | 0.267            |
> | s1 reverse 1 layer     | 0.022            | qwen reverse 1 layer   | 0.255            | llama reverse 1 layer  | 0.300            |
> | s1 reverse 2 layer     | 0.000            | qwen reverse 2 layer   | 0.000            | llama reverse 2 layer  | 0.267            |
> | s1 laco 1 layer        | 0.011            | qwen laco 1 layer      | 0.378            | llama laco 1 layer     | 0.311            |
> | s1 laco 2 layer        | 0.000            | qwen laco 2 layer      | 0.078            | llama laco 2 layer     | 0.267            |
>
> **Table: Results on AIME24 for parallel test-time scaling of non-reasoning s1.1-7B models**
> |  models （non-thinking）\ pass@k | 1 | 2 | 4 | 8 | 16 | 32 | 64 |
> | --------------- | ---- | ---- | ---- | ---- | ---- | ---- | ---- |
> | s1     | 0.1 | 0.137 | 0.169 | 0.206 | 0.250  | 0.301  | 0.356 |
> | s1 shortgpt 1 layer     | 0.0 | 0.056  | 0.06 | 0.065 | 0.073 | 0.085 | 0.1 |
> | s1 shortgpt 2 layer     | 0.0 | 0.023 | 0.033 | 0.046 | 0.061 | 0.076 | 0.089 |
>
>
> Second, regarding the underlying reason for this negative observation: existing theoretical work—most notably Curse of Depth [1]—suggests that deeper layers may become ineffective due to variance explosion in Pre-LN architectures, making them appear “prunable.” This theory is consistent with prior pruning results showing that some deeper layers can indeed be removed without compromising performance on coarse benchmarks such as MMLU.
>
> However, our findings reveal that this theoretical intuition does not transfer to reasoning-extensive benchmarks such as AIME, which require accurate and multi-step reasoning trajectories. While Curse of Depth predicts potential redundancy, our experiments uncover a surprising and previously unreported contradiction. A plausible explanation is that—even if deeper layers are less effective from a signal-propagation perspective—they still play a crucial role in producing high-level, semantically meaningful tokens, rather than merely low-level linguistic features. Removing these layers therefore disrupts the precision and stability needed for reasoning-intensive tasks.
>
> [1] Sun W, Song X, Li P, et al. The curse of depth in large language models. NeurIPS, 2025.

---

> > ### Author Response · Authors · 2025-11-21
> > **Response to Reviewer W7Vc [2/3]**
> >
> > > **Q3: Whether setting a higher temperature or applying repetition penalty mitigate this issue and explanation of why this repetition is happening to reasoning models versus non-reasoning models.**
> >
> > Thank you for you professional question. Belows are the additional experiments and results conducted with respect to your requests:
> >
> > 1. We set the **temperature** to [0, 0.4, 0.7, 1.0, 1.5, 2.0, 3.0], where higher temperatures correspond to higher sampling uncertainty. The results reported below are averaged over nine samples generated by ShortGPT, Reverse-order, and LaCo respectively (three samples for each pruning method). As shown in tables below, We can clearly see that a temperature in the range of 0.7–1.0 is a reasonable choice, but using a higher temperature does not improve reasoning quality.
> >
> > **Table: Results on AIME24 of 1-layer pruned Qwen3-8B for sequential test-time scaling w.r.t. different temperature**
> >
> > | temperature \ thinking tokens | 512    | 1024   | 2048   | 4096   | 8192   |
> > |-------------|--------|--------|--------|--------|--------|
> > | 0           | 0.1889 | 0.2001 | 0.1965 | 0.2557 | 0.3630 |
> > | 0.4         | 0.2556 | 0.2667 | 0.2812 | 0.3333 | 0.4297 |
> > | 0.7         | 0.3148 | 0.2667 | 0.3333 | 0.3778 | 0.4815 |
> > | 1.0         | 0.3519 | 0.3185 | 0.3593 | 0.3741 | 0.4481 |
> > | 1.5         | 0.1481 | 0.1519 | 0.1630 | 0.2260 | 0.2260 |
> > | 2.0         | 0.0074 | 0.0148 | 0.0148 | 0.0074 | 0.0037    |
> > | 3.0         | 0.0    | 0.0    | 0.0    | 0.0037 | 0.0    |
> >
> > **Table: Results on AIME24 of 2-layer pruned Qwen3-8B for sequential test-time scaling w.r.t. different temperature**
> > | temperature \ thinking tokens | 512    | 1024   | 2048   | 4096   | 8192   |
> > |-------------|--------|--------|--------|--------|--------|
> > | 0           | 0.0222 | 0.0296 | 0.0519 | 0.0852 | 0.0852 |
> > | 0.4         | 0.0556 | 0.0815 | 0.0593 | 0.1074 | 0.1333 |
> > | 0.7         | 0.0852 | 0.0890 | 0.0852 | 0.1000 | 0.1222 |
> > | 1.0         | 0.0778 | 0.1074 | 0.1074 | 0.1222 | 0.1481 |
> > | 1.5         | 0.0185 | 0.0111 | 0.0259 | 0.0222 | 0.0148 |
> > | 2.0         | 0.0037 | 0.0    | 0.0037 | 0.0    | 0.0    |
> > | 3.0         | 0.0    | 0.0    | 0.0    | 0.0    | 0.0    |
> >
> >
> >
> > 2. We experiment with different **repetition penalty**, which penalizes new tokens based on whether they appear in the prompt or in the generated text so far. Values greater than 1 encourage the model to use new tokens, while values below 1 encourage the model to repeat tokens (https://docs.vllm.ai/en/v0.6.0/dev/sampling_params.html). We evaluate settings in the range [1.0, 1.1, 1.2, 1.5] . The results reported below are averaged over nine samples generated by ShortGPT, Reverse-order, and LaCo respectively (three samples for each pruning method). As shown in tables below, we can observe that repetition penalty also fails to improve reasoning quality.
> >
> > **Table: Results on AIME24 of 1-layer pruned Qwen3-8B for sequential test-time scaling w.r.t. repetition penalty**
> > | Repetition Penalty \ thinking tokens | 512    | 1024   | 2048   | 4096   | 8192   |
> > |-------------------|--------|--------|--------|--------|--------|
> > | 1.0               | 0.3444 | 0.3444 | 0.3444 | 0.4444 | 0.4778 |
> > | 1.1               | 0.2556 | 0.2222 | 0.2444 | 0.3667 | 0.4667 |
> > | 1.2               | 0.1889 | 0.1556 | 0.1889 | 0.2444 | 0.3667 |
> > | 1.5               | 0.0667 | 0.1222 | 0.1556 | 0.1111 | 0.1222 |
> >
> > **Table: Results on AIME24 of 2-layer pruned Qwen3-8B for sequential test-time scaling w.r.t. repetition penalty**
> >
> > | Repetition Penalty \ thinking tokens | 512    | 1024   | 2048   | 4096   | 8192   |
> > |-------------------|--------|--------|--------|--------|--------|
> > | 1.0               | 0.0556 | 0.1667 | 0.1444 | 0.1000 | 0.1444 |
> > | 1.1               | 0.0778 | 0.1000 | 0.0556 | 0.1000 | 0.1111 |
> > | 1.2               | 0.0556 | 0.0333 | 0.0444 | 0.0333 | 0.0111 |
> > | 1.5               | 0.0000 | 0.0000 | 0.0000 | 0.0000 | 0.0000 |

---

> ### Author Response · Authors · 2025-11-21
> **Response to Reviewer W7Vc [3/3]**
>
> 3. We further experiment with different **frequency penalty**, penalizes new tokens based on their frequency in the generated text so far. Values > 0 encourage the model to use new tokens, while values < 0 encourage the model to repeat tokens (https://docs.vllm.ai/en/v0.6.0/dev/sampling_params.html). We evaluate settings of frequency penalty in the range [0.0, 0.1, 0.3, 0.5, 0.7, 0.9].  The results reported below are averaged over nine samples generated by ShortGPT, Reverse-order, and LaCo respectively (three samples for each pruning method). Also, as shown below, frequency penalty also fails to improve reasoning quality.
>
> **Table: Results on AIME24 of 1-layer pruned Qwen3-8B for sequential test-time scaling w.r.t. frequency penalty**
> | Frequency Penalty \ thinking tokens | 512    | 1024   | 2048   | 4096   | 8192   |
> |------------------|--------|--------|--------|--------|--------|
> | 0                | 0.3333 | 0.3333 | 0.3667 | 0.4333 | 0.4333 |
> | 0.1              | 0.0444 | 0.0333 | 0.0889 | 0.1444 | 0.1556 |
> | 0.3              | 0.0111 | 0.0333 | 0.0444 | 0.0667 | 0.0667 |
> | 0.5              | 0.0111 | 0.0222 | 0.0222 | 0.0333 | 0.0333 |
> | 0.7              | 0.0111 | 0.0111 | 0.0111 | 0.0222 | 0.0444 |
> | 0.9              | 0.0111 | 0.0111 | 0.0222 | 0.0111 | 0.0111 |
>
> **Table: Results on AIME24 of 2-layer pruned Qwen3-8B for sequential test-time scaling w.r.t. frequency penalty**
> | Frequency Penalty \ thinking tokens | 512    | 1024   | 2048   | 4096   | 8192   |
> |------------------|--------|--------|--------|--------|--------|
> | 0                | 0.0778 | 0.2000 | 0.1667 | 0.1778 | 0.1778 |
> | 0.1              | 0.0111 | 0.0111 | 0.0333 | 0.0222 | 0.0444 |
> | 0.3              | 0.0333 | 0.0111 | 0.0111 | 0.0111 | 0.0222 |
> | 0.5              | 0.0000 | 0.0111 | 0.0222 | 0.0222 | 0.0111 |
> | 0.7              | 0.0000 | 0.0111 | 0.0111 | 0.0000 | 0.0222 |
> | 0.9              | 0.0111 | 0.0000 | 0.0111 | 0.0000 | 0.0000 |
>
>
>
> **Summary:**
> We conducted extensive additional experiments on non-reasoning models, temperature scaling, repetition penalty, and frequency penalty, showing that all of these factors fail to mitigate the reasoning degradation caused by layer pruning.
> While non-reasoning models consistently perform worse than reasoning models, they exhibit the same sensitivity to pruning, reinforcing our core claim. Although non-reasoning models do not suffer from repetition during thinking (because they don't have intermediate thinking traces), their inherently weaker reasoning ability makes them more fragile under pruning, leading to a similar pattern of performance collapse.
> Additionally, simple adjustments to the sampling and generation strategy, such as higher temperatures or penalty-based sampling, do not restore reasoning performance.
>
>
> We are grateful for your thoughtful comments and remain open to continued discussion. Given the substantial new evidence and clarifications provided in the rebuttal, we respectfully ask the reviewer to re-evaluate their score.

---

> > ### Author Response · Authors · 2025-11-27
> >
> > Thank you once more for your valuable feedback. We have worked carefully to address all of your comments, and our response has been posted for six days. Please let us know whether our revisions adequately address your concerns or if any additional clarification is needed.

---

### Official Review · Reviewer_PSGU · 2025-11-01

**Soundness:** 3
**Presentation:** 3
**Contribution:** 1
**Rating:** 4
**Confidence:** 3

**Summary:**

The paper investigates on the effect of layer pruning in LLM long chain reasoning, showing that pruning even 1 or 2 layers significantly impair the performance in test-time scaling. The experiments cover different models, datasets, pruning methods and evaluation metrics, and the results are basically consistent. Furthermore, the authors show that SFT after pruning cannot recover the original performance.

**Strengths:**

The experiments contain sufficient ablations, covering different models, datasets, pruning methods and evaluation metrics. The overall conclusions are consistent and can well support the main claim that layer pruning hurts the long chain reasoning performance.

**Weaknesses:**

The main conclusion of the paper is simple and rather intuitive. The layer pruning method would surely decrease the performance since the model loses part of its parameters and face OOD problems compared with training. One could naturally expect these results without experiments, and the conclusions are known. The paper does not provide new interesting results, nor the solution to address the problem.

Moreover, layer pruning itself is not a practically meaningful method from my perspective. Large scale pretraining / post-training aims to improve the reasoning performance, while minimum pruning would severely hurt the performance, which deviates from the original target. Why would people need layer pruning anyways? It is not a principled way in any sense. Even in terms of efficiency, pruning 1 or 2 layers would only bring marginal acceleration, while other methods such as distillation or quantization would significantly improve the efficiency without sacrificing much performance. Unfortunately, the main result of the paper lies in the natural consequence of layer pruning, which is deemed to hurt the performance, while the authors fail to provide any theoretical results, nor any successful methodologies to avoid the performance drop.

**Questions:**

* Since the performance degrade is natural, can you theoretically characterize the phenomenon? Note that contents in Section 5 are mostly case studies and heuristics, not theoretically grounded.
* Can you provide any practical methods to mitigate the issue? Did you try out other SFT configurations? The current setting seems not convincing (only SFT on s1K seems insufficient).
* Did you try out more models? Do you have intuitions on the slight difference in performance under various settings?
* Can you provide any convincing reasons why layer pruning is worth studying? Since all methods hurt the performance, what would be the next step?

---

> ### Author Response · Authors · 2025-11-21
> **Response to Reviewer PSGU [1/4]**
>
> We sincerely appreciate your detailed comments. We provide point-wise responses to address your concerns below.
>
> > **W1: The main conclusion of the paper is simple and rather intuitive. The layer pruning method would surely decrease the performance since the model loses part of its parameters and face OOD problems compared with training. One could naturally expect these results without experiments, and the conclusions are known. The paper does not provide new interesting results, nor a solution to address the problem.**
>
> We sincerely appreciate the reviewer’s comments. In short, our goal is not to promote layer pruning, but to correct recent misleading claims that layer pruning is a “free lunch”.
> - We definitely agree with you that layer pruning can lead to performance degradation. However, our key contribution is not the trivial observation that pruning hurts performance.  **Instead, the overarching goal of our paper is to challenge a rapidly spreading, and potentially misleading assumption in the community—namely, that layer pruning is a “free lunch” for LLMs**. Several recent high-visibility papers claim that one can remove 20–30% of transformer layers with no performance loss [1,2,3,4,5], and these claims have been widely circulated. Importantly, these conclusions are almost always drawn from coarse, low-sensitivity benchmarks such as MMLU or a few multiple-choice reasoning tests. These claims have led the community to underestimate the risks of pruning and to pursue layer-removal–based acceleration as if it were a generally safe operation.
>
>
> - Our work demonstrates that these evaluation protocols are insufficient and can lead to serious misinterpretations. When models are tested on quality-sensitive tasks that require fine-grained generation, accurate reasoning trajectories such as AIME where test-time scaling is required, the purported “lossless” pruning quickly breaks down. In other words, we show that pruning is not a free lunch once we look beyond superficial evaluations.
>
> - In a climate where many works confidently claim that “you can prune 25% of layers with no loss,” it is crucial that someone rigorously tests whether this belief actually holds. Our study serves precisely this purpose. We believe this contribution is both timely and important for ensuring reliable evaluation and preventing premature acceptance of the “pruning is free” narrative.
>
> > **W2: Moreover, layer pruning itself is not a practically meaningful method from my perspective. Large scale pretraining / post-training aims to improve the reasoning performance, while minimum pruning would severely hurt the performance, which deviates from the original target. Why would people need layer pruning anyways?
> Q4: Can you provide any convincing reasons why layer pruning is worth studying? Since all methods hurt the performance, what would be the next step?**
>
> -  We respectfully but strongly disagree with the reviewer’s claim that layer pruning is not practically meaningful. In reality, reduced-depth transformer architectures are one of the most active trends in industry-scale LLM development, precisely because depth directly governs latency, memory footprint, and deployment cost.
>
> - To give concrete examples: **NVIDIA’s Minitron [6] and Nemotron families [7,8] explicitly adopt layer pruning to deliver strong open-sourced LLMs at a fraction of the compute cost**. These models were designed with reduced depth as a core architectural choice—not an afterthought—because layer count is the dominant factor in inference latency on modern GPUs. Given this landscape, dismissing layer pruning as “not meaningful” does not align with current practice. On the contrary, layer reduction is central to scaling LLMs sustainably, especially as organizations seek to deploy high-quality models while controlling infrastructure cost.
>
> - This makes our contribution even more important, as we aim to understand the potential risk and limitation of layer pruning. Understanding these limitations is essential before the community adopts aggressive depth reduction as standard practice.
>
> - Taken together, these points make it clear that layer pruning is not only relevant, but essential to study rigorously.

---

> ### Author Response · Authors · 2025-11-21
> **Response to Reviewer PSGU [2/4]**
>
> > **Q1: Whether the phenomenon can be theoretically characterized rather than relying only on case-based or heuristic explanations.**
>
> Thank you for raising the question regarding theoretical characterization. We agree that a complete formal theory of layer pruning in LLMs is benefitial to the community.
>
> Existing theoretical works, most notably **Curse of Depth** [9], suggest that deeper layers may become ineffective due to variance explosion caused by Pre-LN, and thus appear “prunable”. This theory aligns well with the previous layer pruning paper that some layers (usually the deeper layers) can be removed without compromising performance on MMLU.
>
>
> However, our findings show that this theoretical intuition does not transfer uniformly to modern reasoning benchmarks such as AIME, which require accurate, multi-step reasoning trajectories. While Curse of Depth predicts potential redundancy, our experiments uncover a surprising and previously unreported contradiction. A plausible explanation is that—even if deeper layers are less effective in a signal-propagation sense—they still play a crucial role in generating high-level, semantically meaningful tokens, rather than low-level linguistic patterns. Removing these layers therefore harms the precision and stability required for reasoning-intensive tasks.
>
> In this sense, our work provides empirically grounded evidence that highlights a gap between current theoretical predictions and real-world behavior. We hope our results will motivate future research toward a more complete theoretical and practical understanding of layer pruning in LLMs.
>
>
>
> [1] Men X, Xu M, Zhang Q, et al. Shortgpt: Layers in large language models are more redundant than you expect. Findings ACL 2025. pages 20192–20204.
>
> [2] Yang, Y., Cao, Z. and Zhao, H., 2024. Laco: Large language model pruning via layer collapse. EMNLP 2024.
>
> [3] Chen, X., Zhang, H., Zeng, F., Wei, Y., Wang, Y., Ling, X., Li, G. and Yuan, C., 2025. Prune&comp: Free lunch for layer-pruned llms via iterative pruning with magnitude compensation. AAAI 2026.
>
> [4] Lu Y, Cheng H, Fang Y, et al. Reassessing layer pruning in llms: New insights and methods. arXiv preprint arXiv:2411.15558, 2024.
>
> [5] Kim, B.K., Kim, G., Kim, T.H., Castells, T., Choi, S., Shin, J. and Song, H.K., 2024. Shortened llama: A simple depth pruning for large language models. arXiv preprint arXiv:2402.02834, 11, p.1.
>
> [6] Sreenivas, S.T., Muralidharan, S., Joshi, R., Chochowski, M., Mahabaleshwarkar, A.S., Shen, G., Zeng, J., Chen, Z., Suhara, Y., Diao, S. and Yu, C., 2024. Llm pruning and distillation in practice: The minitron approach. arXiv preprint arXiv:2408.11796.
>
> [7] Blakeman, A., Basant, A., Khattar, A., Renduchintala, A., Bercovich, A., Ficek, A., Bjorlin, A., Taghibakhshi, A., Deshmukh, A.S., Mahabaleshwarkar, A.S. and Tao, A., 2025. Nemotron-h: A family of accurate and efficient hybrid mamba-transformer models. arXiv preprint arXiv:2504.03624.
>
> [8] Basant, A., Khairnar, A., Paithankar, A., Khattar, A., Renduchintala, A., Malte, A., Bercovich, A., Hazare, A., Rico, A., Ficek, A. and Kondratenko, A., 2025. Nvidia nemotron nano 2: An accurate and efficient hybrid mamba-transformer reasoning model. arXiv preprint arXiv:2508.14444.
>
> [9] Sun W, Song X, Li P, et al. The curse of depth in large language models. NeurIPS, 2025.

---

> ### Author Response · Authors · 2025-11-21
> **Response to Reviewer PSGU [3/4]**
>
> > **Q2: Can you provide any practical methods to mitigate the issue? Did you try out other SFT configurations? The current setting seems not convincing (only SFT on s1K seems insufficient).**
>
> Thank you for your critical feedback.
>
> Regarding practical mitigation methods, it is possible that **industry-level large-scale retraining on the original pre-training dataset** could fully recover the model’s capacity after pruning. However, such large-scale retraining is far beyond the computational resources available to typical academic labs, including ours.
>
> Still, we tried our best to explore two more larger-scale configurations: (1) 10x scale SFT; (2) RL-based post-training.
>
> 1. **10x scale SFT:** We extended the scale of our SFT by 10x, i.e., performing SFT on 10k samples from the OpenR1-Math-220k dataset on pruned s1.1-7B (https://huggingface.co/datasets/open-r1/OpenR1-Math-220k). As shown in the table below (also refer to **Appendix H** in the updated PDF for comprehensive results), we found that when the data size increases by **10x**, the effectiveness of SFT improves largely, especially for one layer pruning. However, this extended SFT still **fail to fully recover** the test scaling performance when the number of prune layers is beyond one.
>
>
> **Table: Sequential test-time scaling on AIME24 of s1.1-7B after Full FT on 10k subset of OpenR1-Math-220k**
> | pruned layers \ thinking tokens | 512    | 1024   | 2048   | 4096   | 8192   |
> |------------------|--------|--------|--------|--------|--------|
> | origin           | 0.0444 | 0.0670 | 0.1330 | 0.0890 | 0.1443 |
> | shortgpt 1 layer | 0.0777 | 0.0670 | 0.1110 | 0.0777 | 0.1330 |
> | shortgpt 2 layer | 0.0330 | 0.0333 | 0.0333 | 0.0667 | 0.0777 |
> | reverse 1 layer  | 0.1000 | 0.0777 | 0.1110 | 0.1223 | 0.1223 |
> | reverse 2 layer  | 0.0890 | 0.0667 | 0.0777 | 0.1000 | 0.1020 |
> | laco 1 layer     | 0.0777 | 0.0890 | 0.0667 | 0.1000 | 0.1330 |
> | laco 2 layer     | 0.0444 | 0.0330 | 0.0220 | 0.0557 | 0.0777 |
>
>
> 2. **RL-based post-training:** We further applied GRPO to pruned models, using 12K **simplescaling/openaimath**  (https://huggingface.co/datasets/simplescaling/openaimath). Due to time limit, we adopted a concise GRPO configuration: a small learning rate (5e-6), AdamW (8-bit) optimizer, cosine learning-rate schedule, gradient accumulation of 4, prompt length capped at 2K tokens, completion length up to 8K, and standard reward functions combining format and correctness rewards. Training is executed with vLLM in colocate mode for efficiency. Below are the averaged results on **s1.1-7B**, showing that although GRPO **fails restore test-time scaling behavior**:
>
> **Table: Sequential test-time scaling on MATH500 of s1.1-7B after GRPO on 12K openaimath.**
> | pruned layers \ thinking tokens | 512   | 1024  | 2048  | 4096  | 8192  |
> |---------------------------------|-------|-------|-------|-------|-------|
> | original model                  | 0.626 | 0.701 | 0.760 | 0.783 | 0.803 |
> | shortgpt_1_layer                | 0.559 | 0.575 | 0.617 | 0.623 | 0.653 |
> | shortgpt_1_layer_grpo           | 0.524 | 0.607 | 0.638 | 0.649 | 0.649 |

---

> ### Author Response · Authors · 2025-11-21
> **Response to Reviewer PSGU [4/4]**
>
> > **Q3: Did you try out more models? Do you have intuitions on the slight difference in performance under various settings?**
>
>
> Thank you for your question. We further evaluated on **Qwen3-14B** and **GPT-oss 20B**. **GPT-oss 20B** is highly unstable under pruning. When we prune even a single layer using ShortGPT, its output format collapses, making it completely unable to answer questions. The output becomes devoid of semantic meaning, consisting largely of repetitive control tokens (e.g., <|im_start|>, <|im start|>im start|>assistant<|im start|>) and incoherent sequences rather than valid reasoning traces. For **Qwen3-14B**, as shown in tables below (refer **Appendix L** for illustrated figures), we observe a consistent pattern: **layer pruning inevitably harms the model’s scientific reasoning ability**. Larger Qwen3-14B shows slightly smaller performance drops, but the degradation is still clear and systematic.
>
> Overall, Qwen3-8B and Qwen3-14B are more robust to pruning than s1.1-7B. Pruning just the first layer of s1.1-7B already causes a sharp drop in reasoning ability, whereas Qwen3-8B and Qwen3-14B typically do not exhibit a major collapse until the second pruned layer. GPT-oss is highly unstable under pruning — pruning even a single layer severely breaks its output format, making it unable to answer questions. This indicates that for certain architectures, pruning destroys the fundamental mechanisms required for generation.
>
> **Table: Sequential test-time scaling of Qwen3-14B on AIME24**
>
> |  pruned layers\thinking tokens | 512 | 1024 | 2048 | 4096 | 8192 |
> | ------------------------------ | --- | ---- | ---- | ---- | ---- |
> |  original model                            | 0.3 | 0.267 | 0.4 | 0.433 | 0.533 |
> | shortgpt_1_layer               | 0.233 | 0.1 | 0.367 | 0.433, 0.433 | 0.667, 0.633 |
> | shortgpt_2_layer               | 0.033 | 0.067 | 0.1 | 0.167 | 0.233 |
> | sll_1_layer                    | 0.133 | 0.233 | 0.3 | 0.433 | 0.567 |
> | sll_2_layer                    | 0.033 | 0.1 | 0.033 | 0.2 | 0.233 |
> | laco_1_layer                    | 0.167 | 0.167 | 0.2 | 0.367 | 0.567 |
> | laco_2_layer                    | 0.1 | 0.067 | 0.1 | 0.133 | 0.267 |
>
> **Table: Sequential test-time scaling of Qwen3-14B on GPQA_Diamond**
>
> |  pruned layers\thinking tokens | 512 | 1024 | 2048 | 4096 | 8192 |
> | ------------------------------ | --- | ---- | ---- | ---- | ---- |
> |  original model                            | 0.444 | 0.515 | 0.525 | 0.566 | 0.626 |
> | shortgpt_1_layer               | 0.465 | 0.439 | 0.535 | 0.566 | 0.601 |
> | shortgpt_2_layer               | 0.444 | 0.459 | 0.475 | 0.531 | 0.531 |
> | sll_1_layer                    | 0.469 | 0.5 | 0.531 | 0.566 | 0.591 |
> | sll_2_layer                    | 0.414 | 0.419 | 0.459 | 0.505 | 0.53 |
> | laco_1_layer                    | 0.459 | 0.459 | 0.5 | 0.565 | 0.601 |
> | laco_2_layer                    | 0.379 | 0.429 | 0.485 | 0.490 | 0.493 |
>
> **Table: Sequential test-time scaling of Qwen3-14B on MATH500**
>
> |  pruned layers\thinking tokens | 512 | 1024 | 2048 | 4096 | 8192 |
> | ------------------------------ | --- | ---- | ---- | ---- | ---- |
> |  original model                            | 0.832 | 0.874 | 0.916 | 0.948 | 0.948 |
> | shortgpt_1_layer               | 0.756 | 0.83 | 0.89 | 0.922 | 0.944 |
> | shortgpt_2_layer               | 0.674 | 0.76 | 0.794 | 0.818 | 0.814 |
> | sll_1_layer                    | 0.784 | 0.844 | 0.876 | 0.924 | 0.956 |
> | sll_2_layer                    | 0.594 | 0.664 | 0.712 | 0.748 | 0.76 |
> | laco_1_layer                    | 0.768 | 0.838 | 0.878 | 0.9 | 0.954 |
> | laco_2_layer                    | 0.604 | 0.664 | 0.712 | 0.758 | 0.776 |
>
>
>
> We are grateful for your constructive comments and remain open to further dialogue. Should you have any additional concerns, we would be pleased to address them.

---

> > ### Author Response · Authors · 2025-11-27
> >
> > Thank you again for your thoughtful comment. We have made a concerted effort to address all of your points and have kept our response posted for six days. We would greatly appreciate it if you could let us know whether our revisions address your concerns or if any further clarification would be helpful.

---

### Meta-Review · Area_Chair_egXG · 2026-01-06

**Summary:**

This paper demonstrates a negative result: while layer pruning has been popular for some distilled small LMs, it hurts test-time scaling performance.

**Reviewer Concerns:**

The main consensus is whether or not the result is substantial enough, on its own, to warrant acceptance. Here is my synthesis of reviews and author response:
1. The authors rightfully motivate their paper by describing that layer pruning is one of many ways to reduce inference costs, but that this can be catastrophic for models.
2. However, the authors single out only layer pruning as the mechanism of study, and do not seem to find strong positive prescriptions that mitigate this affects.
3. It seems that a more successful framing of the problem would be: "what is the best way to prune a model to improve inference time/reduce latency, subject to maintain test-time scaling performance?" I believe a more comprehensive evaluation would have been useful.

**Reviewer Scores:**

Given the discussion, I believe that the 4 and 2 scores would have remained below the acceptance threshold. It is possible that 6 would have raised the score, and 8 would have maintained the score, still leading towards a reject.

---

### Decision · Program_Chairs · 2026-01-26

Reject